

# Aerosol optical properties and instantaneous radiative forcing based on high temporospatial resolution CARSNET ground-based measurements over eastern China

Huizheng Che[1*], Bing Qi[2], Hujia Zhao[1], Xiangao Xia[3,4], Philippe Goloub[5], Oleg Dubovik [5],

Victor Estelles [6], Emilio Cuevas-Agulló[7], Luc Blarel[3], Yunfei Wu[8], Jun Zhu[9], Rongguang Du[2],

Yaqiang WANG[1], Hong Wang[1], Ke Gui[1], Jie Yu[1], Yu Zheng[9], Tianze Sun[1], Quanliang Chen[10],

Guangyu Shi[11], Xiaoye Zhang[1*]

State Key Laboratory of Severe Weather (LASW) and Institute of Atmospheric Composition, Chinese Academy of Meteorological Sciences, CMA, Beijing, 100081, China
Hangzhou Meteorological Bureau, Hangzhou, 310051, China
Laboratory for Middle Atmosphere and Global Environment Observation (LAGEO), Institute of Atmospheric Physics, Chinese Academy of Sciences, Beijing, 100029, China
School of Geoscience University of Chinese Academy of Science, Beijing, 100049, China
Laboratoire d'Optique Amosphérique, Université des Sciences et Technologies de Lille, 59655, Villeneuve d'Ascq, France
Dept. Fisica de la Terra i Termodinamica, Universitat de Valencia, C/ Dr. Moliner 50, 46100 Burjassot, Spain
Centro de Investigación Atmosférica de Izaña, AEMET, 38001 Santa Cruz de Tenerife , Spain
Key Laboratory of Regional Climate-Environment for Temperate East Asia, Institute of Atmospheric Physics, Chinese Academy of Sciences, Beijing 100029, China
Collaborative Innovation Center on Forecast and Evaluation of Meteorological Disasters, Nanjing University of Information Science & Technology, Nanjing 210044, China
Plateau Atmospheric and Environment Key Laboratory of Sichuan Province, College of Atmospheric Sciences, Chengdu University of Information Technology, Chengdu, 610225, China
State Key Laboratory of Numerical Modeling for Atmospheric Sciences and Geophysical Fluid Dynamics (LASG), Institute of Atmospheric Physics, Chinese Academy of Sciences, Beijing, 100029, China

Corresponding author: chehz@camscma.cn & xiaoye@camscma.cn



## Abstract

Variations in the optical properties of aerosols and their radiative forcing were investigated
based on long-term synchronous observations made at three-minute intervals from 2011 to
2015 over seven adjacent CARSNET (China Aerosol Remote Sensing NETwork) urban
(Hangzhou), suburban (Xiaoshan, Fuyang, LinAn, Tonglu, Jiande) and rural (ChunAn) stations
in the Yangtze River Delta region, eastern China. The aerosol optical depth (AOD) varied from
0.68 to 0.76, with two peaks in June and September, and decreased from the eastern coast to
western inland areas. The ratio of the AOD of fine-mode particles to the total AOD was >0.90
and the extinction Angström exponent was >1.20 throughout the year at all seven sites. The
Moderate Resolution Imaging Spectroradiometer (MODIS) C6 retrieval AOD was validated by
comparison with ground-based observations. The correlation coefficients ($R^2$) between the
MODIS C6 AOD data and the values measured on the ground were ~0.73−0.89. The
single-scattering albedo varied from 0.91 to 0.94, indicating that scattering aerosol particles
are dominant in this region. The real parts of the refractive index were ~1.41–1.43, with no
significant difference among the seven urban, suburban and rural sites. Large imaginary parts
of the refractive index were seen in August at all urban, suburban and rural sites. The
fine-mode radii in the Yangtze River Delta region were ~0.2–0.3 μm with a volume of 0.10–
0.12 μm$^3$ and the coarse-mode radii were ~2.0 μm with a volume close to 0.07 μm$^3$. The
fine-mode aerosols were obviously larger in June and September than in other months at
almost the sites. The absorption AOD was low in the winter. The absorption Angström
exponent and the extinction Angström exponent were used to classify the different types of
aerosol and the components of mixtures. The aerosols caused negative radiative forcing both
at the Earth's surface and at the top of the atmosphere all year round in the Yangtze River
Delta region of eastern China.





## 1. Introduction

Aerosols have important effects on the Earth's climate at both global and regional scales, although there are still great uncertainties in assessing their impact (Hansen et al. 2000; Solomon et al., 2007; Schwartz and Andreae, 1996). Aerosols affect not only the radiative balance of the Earth–atmosphere system by directly scattering and absorbing solar radiation (Charlson et al., 1992; Ackerman and Toon, 1981), but also indirectly affect the climate through aerosol–cloud interactions (Twomey et al., 1984; Albrecht et al., 1989; Li et al., 2016).

The optical properties of aerosols influence the aerosol radiative balance and can be used to predict and assess global and regional changes in the Earth's climate (Eck et al., 2005; Myhre et al., 2009; IPCC, 2013; Panicker et al., 2013). Long-term, ground-based observations are crucial to our understanding of the global and regional variations in the optical properties of aerosols and their effects on the Earth's climate (Holben et al., 2001; Kaufman et al., 2002; Sanap and Pandithurai, 2014; Li et al., 2016). Ground-based monitoring networks have been established worldwide—for instance, AERONET (Holben et al., 1998; Goloub et al., 2007), SKYNET (Takamura et al., 2004), EARLINET (Pappalardo et al., 2014) and the GAW-PFR Network (Wehrli, 2002; Estelles et al., 2012), which includes several automated sites in China. CARSNET (the China Aerosol Remote Sensing NETwork) (Che et al., 2009a, 2015b) and CSHNET (the Chinese Sun Hazemeter Network) were established to obtain data on aerosol optical characteristics in China (Xin et al., 2007).

Most of the ground-based studies of the optical properties of aerosols in China have been concentrated in urban regions undergoing rapid economic development, which have high aerosol loadings and serious environmental problems (Cheng et al., 2015; Pan et al., 2010; Xia et al., 2013; Wang et al., 2015; Che et al., 2015a). Analyses of the aerosol optical depth (AOD), the types of aerosol present and the classification of ambient aerosol populations based on their size and absorption properties (Giles et al., 2011) are needed to understand their effects on the Earth's climate and environment (Che et al., 2009b; Wang et al., 2010; Zhu et al., 2014).



The Yangtze River Delta (YRD) region in eastern China has undergone rapid economic
growth and has high emissions of aerosols (Fu et al., 2008; Zhang et al., 2009). There have
been many studies of the optical properties of aerosols in eastern China and these are
important in our understanding of both the local air quality and regional climate change (Duan
and Mao, 2007; Pan et al., 2010; Ding et al., 2016). Basic investigations of the variation in the
optical characteristics of aerosols over the YRD region have been carried out at Nanjing, Hefei,
Shanghai, Shouxian and Taihu (Zhuang et al. 2014; Li et al., 2015; Wang et al., 2015; He et al.,
2012; Lee et al., 2010; Cheng et al., 2015; Xia et al., 2007). These studies in the YRD region
have mostly been single-site and/or short-period investigations. The study sites are ~100 km
apart from each other, which makes high spatial resolution satellite and modeling validations
difficult. Thus there is still a lack of long-term, continuous and synchronous observations of the
optical characteristics of aerosols, especially over adjacent urban, suburban and rural areas in
the YRD region.
High-frequency ground-based observations of the variations in the optical characteristics
of aerosols are necessary to our understanding of the processes involved in air pollution (e.g.
the source, transport and diurnal variations of the pollution) and their effect on the regional
climate. Ground-based observations are also important in the validation and improvement of
satellite retrieval data (Holben et al., 2017; Xie et al., 2011). A high density of ground-based
sun- and sky-scanning spectral radiometers within a local or meso-scale region is required to
capture small-scale variations in aerosols for the accurate validation of satellite observations
and to compare in situ versus remote sensing observations (Xiao et al., 2016; Holben et al.,
2017). The MODIS (Moderate Resolution Imaging Spectroradiometer) retrieval AOD has a
high accuracy with a wide spectral coverage (Tanré et al., 1997; Kaufman, et al. 1997) and the
algorithm has been validated and improved based on AERONET data (Chu et al., 2002;
Ichoku et al., 2002; Remer et al., 2005; Levy et al., 2010;). Levy et al. (2013) refined the
MODIS Collection 6 (C6) aerosol retrieval process to provide better AOD retrievals. Some
validations of satellite aerosol retrievals have been carried out in China with ground-based
observations from CSHNET (Li, et al., 2007; Wang, et al., 2007; Xin, et al., 2007) and
CARSNET (Che et al., 2009a, Che et al., 2011a; Tao et al., 2015).





We investigated the variation in the optical properties of aerosols and aerosol radiative
forcing (ARF) using three-minute intervals of sun photometer measurements from 2011 to
2015 at seven adjacent CARSNET (~10–40 km) urban, suburban and rural sites over eastern
China. The aims of this study were: (1) to investigate the synchronous variations and
differences in the optical properties of aerosols over urban, suburban and rural areas of the
YRD megacity, eastern China; (2) to analyze the type and dominant distribution pattern of
aerosols in the YRD via the extinction and absorption properties of aerosols; (3) to understand
the difference in the ARF calculated from ground-based measurements of the optical
properties of aerosols over urban, suburban and rural areas in eastern China; and (4) to
evaluate the MODIS AOD retrieval data using the CARSNET AOD for the YRD. The results of
this study will help the satellite and modeling communities to improve future aerosol retrieval
data and simulations.
**2. Site descriptions, measurements and data**
Fig. 1 shows the geographical locations of the seven CARSNET sites in the YRD; these
locations are described in Table 1.

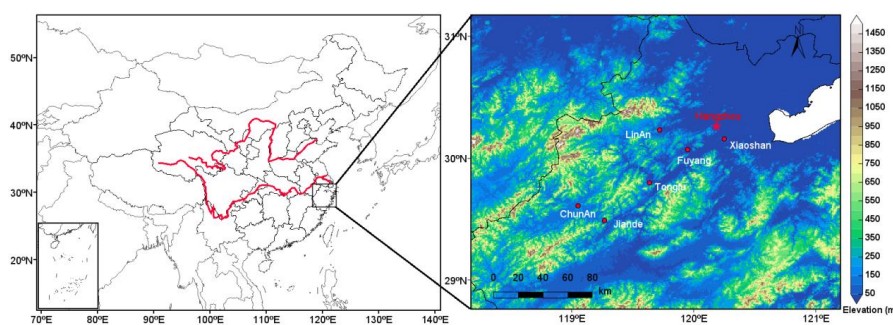

Fig. 1. Geographical location and elevation map for the seven CARSNET sites in the YRD.
The rural site of ChunAn can be regarded as a representative background location
unaffected by local and regional pollution. The site has a small population and a good
ecological environment, although there is some agricultural activity and burning of biomass
from crop residues. Hangzhou is a densely populated urban site with a large volume of


vehicular traffic and is therefore more affected by anthropogenic activity. LinAn, Fuyang,
Jiande, Xiaoshan Tonglu and Xiaoshan are suburban sites and are all affected by both
anthropogenic activity and pollution from industrial and agricultural production.
CE-318 sun photometers (Cimel Electronique, Paris, France) were installed at these
seven sites in the YRD from 2011 to 2015. The instruments were standardized and calibrated
annually according to the protocols reported by Che et al. (2009a). The instruments in this
study were made inter-comparison calibration by the CARSNET reference instruments, which
were periodically calibrated at Izaña in Spain. The cloud-screened AOD at different
wavelengths was obtained using ASTPwin software (Cimel Electronique) (Smirnov et al.,
2000). Instantaneous direct data for the AOD were selected at least ten times each day at a
temporal resolution of about three minutes and the corresponding values of Angström
exponent (α) were calculated by instantaneous AOD values at 440 and 870 nm.
The aerosol optical properties—including the single-scattering albedo (SSA), the complex
refractive index, the volume size distribution, the absorption AOD (AAOD), the absorption
Angström exponent (AAE) and the fraction of spherical particles—were retrieved from the
almucantar irradiance measurements according to the methods of Dubovik and King (2000)
and Dubovik et al. (2002, 2006). The SSA was retrieved using only $AOD_{440nm} > 0.40$
measurements to avoid the large uncertainties inherent in a low AOD. The complex refractive
index was also retrieved by sky irradiance measurements in the range 1.33–1.60 for the real
part and in the range 0.0005–0.50 for the imaginary part (Dubovik and King, 2000; Che et al.,
2015b). In the volume size distribution, the radius range is selected from 0.05–15μm. The
AAOD and the AAE were calculated as described in equations (1) and (2):
$$AAOD(\lambda) = [1 - SSA(\lambda)] \times EAOD(\lambda) \qquad (1)$$
$$AAE = -d\ln[AAOD(\lambda)]/d\ln(\lambda) \qquad (2)$$
The ARF data were calculated by the radiative transfer module used by the AERONET
inversion (García et al., 2012). The broadband fluxes from 0.20 to 4.0 μm were calculated
according to the radiative transfer model GAME (Global Atmospheric ModEl) (Dubuisson et al.,





1996, 2006; Roger et al., 2006).
The MODIS C6 aerosol optical thickness products refined by Levy et al. (2013) were used
to compare the MODIS AOD retrievals with our ground-based observations. The MODIS C6
AOD retrievals were formed into a merged dataset combining the Deep Blue and Dark Target
methods. This version of MODIS includes some important changes from earlier
versions—such as the central wavelength assumptions, Rayleigh scattering and the gas
absorption performance (Levy et al., 2013)—and improvements in the radiometric calibration
(Lyapustin et al., 2014). All cloud- and snow-free land surfaces have been expanded in the
MODIS C6 aerosol products (Hsu et al., 2013). The AOD data from Terra-MODIS were
validated by matching the CARSNET AODs within 30 minutes of the MODIS overpass within
the 3×3 pixels surrounding the CARSNET site. The AOD at 550 nm was interpolated between
two wavelengths of the ground-based AOD measurements at 440 and 675 nm.
**3. Results and discussion**
**3.1 Aerosol optical depth and Angström exponent**
The AOD over the seven urban, suburban and rural sites in this study varied from 0.68 to
0.76 (Table 1). The annual values of the AOD at Hangzhou, Xiaoshan, Fuyang, LinAn, Tonglu,
Jiande and ChunAn were about 0.76±0.42, 0.76±0.43, 0.76±0.45, 0.73±0.44, 0.71±0.41,
0.73±0.40 and 0.68±0.38, respectively, which suggests that aerosol loading is at a high level at
all seven urban, suburban and rural sites in the YRD. This suggests that aerosol pollution is on
the regional rather than the local scale in the YRD region. The AOD at the urban site of
Hangzhou was the highest of all the study sites as a result of high local anthropogenic activity
in this urban area compared with the other suburban and rural sites. The AOD at the rural site
of ChunAn was lower than at the urban and suburban sites due to lower levels of
anthropogenic activity. The AOD decreased from the eastern coast to the inland areas towards
the west (from ~0.76±0.42 at Hangzhou to ~0.68±0.38 at ChunAn). This is due to the high
aerosol loading from economic development and anthropogenic influences. There is more
industrial activity and high resident density in the eastern part of the Hangzhou metropolis



region, resulting in higher aerosol emissions. The AOD in Hangzhou in urban eastern China
was similar to that in Shenyang (0.75) in urban northeast China (Zhao et al., 2013), and in
Beijing (0.76) and Tianjin (0.74) in urban north China (Che et al., 2015b), indicating that
aerosol pollution is both common and at a similar level throughout most urban areas of China.
The AOD values at the urban and suburban sites of Hangzhou were slightly higher than at
Pudong (0.70) and Hefei (0.69), other urban areas in eastern China, suggesting that higher
aerosol loadings were emitted here (He et al., 2012; Liu et al., 2017). However, the AOD at all
seven sites was lower than that obtained at Wuhan (1.05), Nanjing (0.88), Dongtan (0.85),
Taihu (0.77) and Xuzhou (0.92) in previous studies in eastern China (Wang et al., 2015; Li et
al., 2015; Pan et al., 2010; Xia et al., 2007; Wu et al., 2016). This indicates that the aerosol
loading caused by anthropogenic activities is very high in both urban and suburban areas in
eastern China. The site at LinAn is regarded as the regional background site in eastern China
and is representative of the background atmospheric characteristics of this region (Che et al.,
2009c). The average AOD at LinAn was about 0.73±0.44, which is higher than that at the other
regional background stations of China, such as Longfengshan (0.35; northeastern China), Mt
Waliguan (0.14, inland Asia), Xinglong (0.28, northern China), Akedala (0.20, northwestern
China) and Shangri-La (0.11, southwestern China) (Wang et al., 2010; Che et al., 2011; Zhu et
al., 2014; Che et al., 2015b). The aerosol loading in eastern China (especially in the YRD
region) is at least twice as high as in other regions of China.
Table 1. Geographical location and annual mean optical parameters of aerosols at the seven
observation sites in the YRD.

|  | Hangzhou | Xiaoshan | Fuyang | LinAn | Tonglu | Jiande | ChunAn |
|---|---|---|---|---|---|---|---|
| Site type | Urban | Suburban | Suburban | Suburban | Suburban | Suburban | Rural |
| Longitude (° E) | 120.19 | 120.25 | 119.95 | 119.72 | 119.64 | 119.27 | 119.05 |
| Latitude (° N) | 30.26 | 30.16 | 30.07 | 30.23 | 29.80 | 29.49 | 29.61 |
| Altitude (m) | 41.9 | 14.0 | 17.0 | 139 | 46.1 | 88.9 | 171.4 |
| [a]$N_{day}$ | 485 | 180 | 217 | 562 | 498 | 480 | 439 |
| [b]$N_{inst.}$ | 2052 | 752 | 906 | 2410 | 2255 | 1952 | 1731 |
| [c]AOD | 0.76±0.42 | 0.76±0.43 | 0.76±0.45 | 0.73±0.44 | 0.71±0.41 | 0.73±0.40 | 0.68±0.38 |
| [c]$AOD_{fine}$ | 0.68±0.42 | 0.69±0.41 | 0.69±0.44 | 0.66±0.43 | 0.64±0.41 | 0.66±0.40 | 0.61±0.38 |
| [c]$AOD_{coarse}$ | 0.08±0.06 | 0.07±0.06 | 0.07±0.06 | 0.07±0.07 | 0.07±0.06 | 0.07±0.07 | 0.06±0.05 |
| [d]EAE | 1.29±0.26 | 1.37±0.24 | 1.32±0.24 | 1.29±0.27 | 1.30±0.26 | 1.32±0.28 | 1.22±0.25 |





| [c]SSA | 0.91±0.06 | 0.93±0.04 | 0.94±0.04 | 0.93±0.05 | 0.92±0.04 | 0.92±0.05 | 0.94±0.03 |
|---|---|---|---|---|---|---|---|
| [c]$SSA_{fine}$ | 0.93±0.05 | 0.95±0.04 | 0.95±0.04 | 0.94±0.04 | 0.94±0.04 | 0.94±0.05 | 0.95±0.03 |
| [c]$SSA_{coarse}$ | 0.82±0.09 | 0.83±0.08 | 0.84±0.08 | 0.81±0.08 | 0.81±0.08 | 0.82±0.09 | 0.81±0.07 |
| [c]Real | 1.43±0.07 | 1.41±0.06 | 1.41±0.06 | 1.42±0.06 | 1.43±0.06 | 1.41±0.05 | 1.41±0.05 |
| [c]Imaginary | 0.011±0.010 | 0.008±0.006 | 0.007±0.006 | 0.009±0.007 | 0.009±0.007 | 0.010±0.009 | 0.007±0.004 |
| [c]AAOD | 0.06±0.05 | 0.05±0.04 | 0.04±0.04 | 0.05±0.04 | 0.05±0.04 | 0.06±0.04 | 0.04±0.03 |
| [d]AAE | 1.13±0.46 | 0.88±0.42 | 0.85±0.43 | 0.98±0.35 | 1.11±0.49 | 1.16±0.44 | 0.93±0.31 |
| [c]$Rmeas_t$ (µm) | 0.70±0.34 | 0.65±0.31 | 0.66±0.33 | 0.66±0.33 | 0.65±0.33 | 0.62±0.24 | 0.65±0.30 |
| [c]$Rmea_{fine}$ (µm) | 0.18±0.05 | 0.18±0.04 | 0.19±0.05 | 0.19±0.05 | 0.19±0.05 | 0.19±0.05 | 0.20±0.05 |
| [c]$Rmea_{coarse}$ (µm) | 2.67±0.47 | 2.73±0.42 | 2.75±0.45 | 2.71±0.52 | 2.66±0.48 | 2.63±0.47 | 2.74±0.49 |
| [c]Reff (µm) | 0.30±0.10 | 0.29±0.09 | 0.30±0.09 | 0.29±0.10 | 0.29±0.10 | 0.29±0.09 | 0.30±0.10 |
| [c]$Reff_{fine}$ (µm) | 0.16±0.04 | 0.16±0.03 | 0.17±0.04 | 0.16±0.04 | 0.16±0.04 | 0.17±0.04 | 0.17±0.04 |
| [c]$Reff_{coarse}$ (µm) | 2.21±0.40 | 2.26±0.35 | 2.30±0.39 | 2.24±0.44 | 2.19±0.41 | 2.16±0.39 | 2.27±0.42 |
| [c]Volume ($µm^3$) | 0.19±0.09 | 0.19±0.09 | 0.19±0.09 | 0.18±0.09 | 0.17±0.09 | 0.18±0.09 | 0.17±0.07 |
| [c]$Volume_{fine}$ ($µm^3$) | 0.10±0.06 | 0.11±0.06 | 0.11±0.07 | 0.10±0.06 | 0.10±0.06 | 0.10±0.06 | 0.10±0.06 |
| [c]$Volume_{coarse}$ ($µm^3$) | 0.09±0.06 | 0.08±0.05 | 0.08±0.06 | 0.08±0.05 | 0.08±0.06 | 0.08±0.07 | 0.07±0.05 |
| [c]ARF-BOT ($W/m^2$) | −93±44 | −84±41 | −80±40 | −81±39 | −79±39 | −82±40 | −74±34 |
| [c]ARF-TOA ($W/m^2$) | −35±20 | −36±21 | −37±21 | −36±21 | −35±20 | −35±21 | −40±19 |

[a] Number of available observation days.
[b] Number of instantaneous observations.
[c] Optical parameters at a wavelength of 440 nm.
[d] Angström exponents between 440 and 870 nm.

Ding et al. (2013a, b) showed that plumes from agricultural burning in June may
significantly and seriously affect the radiation balance and air quality of the YRD region. In this
study, the monthly averaged AODs at most sites showed two peaks in June and September
(Fig. 2) with values of ~1.26±0.50 and ~1.03±0.57, respectively. This may be attributed to the
accumulation of fine-mode particles via hygroscopic growth in the summer season and the
burning of crop residue biomass under a continental high-pressure system with good
atmospheric stability and frequent temperature inversions. These conditions lead to the poor
diffusion of pollutants (Xia et al., 2007).





The annual fine-mode AOD values at Hangzhou, Xiaoshan, Fuyang, LinAn, Tonglu,
Jiande and ChunAn were about 0.68±0.42, 0.69±0.41, 0.69±0.44, 0.66±0.43, 0.64±0.41,
0.66±0.40 and 0.61±0.38, respectively (Fig. 2). The seasonal variation in the AOD was similar
to the total AOD at these urban, suburban and rural sites. The ratio $AOD_f/AOD_t$ consistently
exceeded 0.90 at all sites, which indicates that fine-mode particles make a major contribution
to the total AOD in the YRD. The annual coarse-mode AOD values at Hangzhou, Xiaoshan,
Fuyang, LinAn, Tonglu, Jiande and ChunAn were between about 0.06 and 0.08. The ratio
$AOD_c/AOD_t$ was about 0.10, which indicates that about 10% of the contribution to the AOD in
the YRD region is from coarse particles. The variation in the coarse-mode AOD (Fig. 2) also
showed a significant increase in March at all seven sites of about 0.14±0.08, 0.08±0.04,
0.09±0.09, 0.13±0.11, 0.13±0.11, 0.14±0.08 and 0.11±0.07 at Hangzhou, Xiaoshan, Fuyang,
LinAn, Tonglu, Jiande and ChunAn, respectively. This was mainly caused by dust episodes
from north/northwest China, which contributed to the optical properties of aerosols in this
region (Zhang et al., 2012).





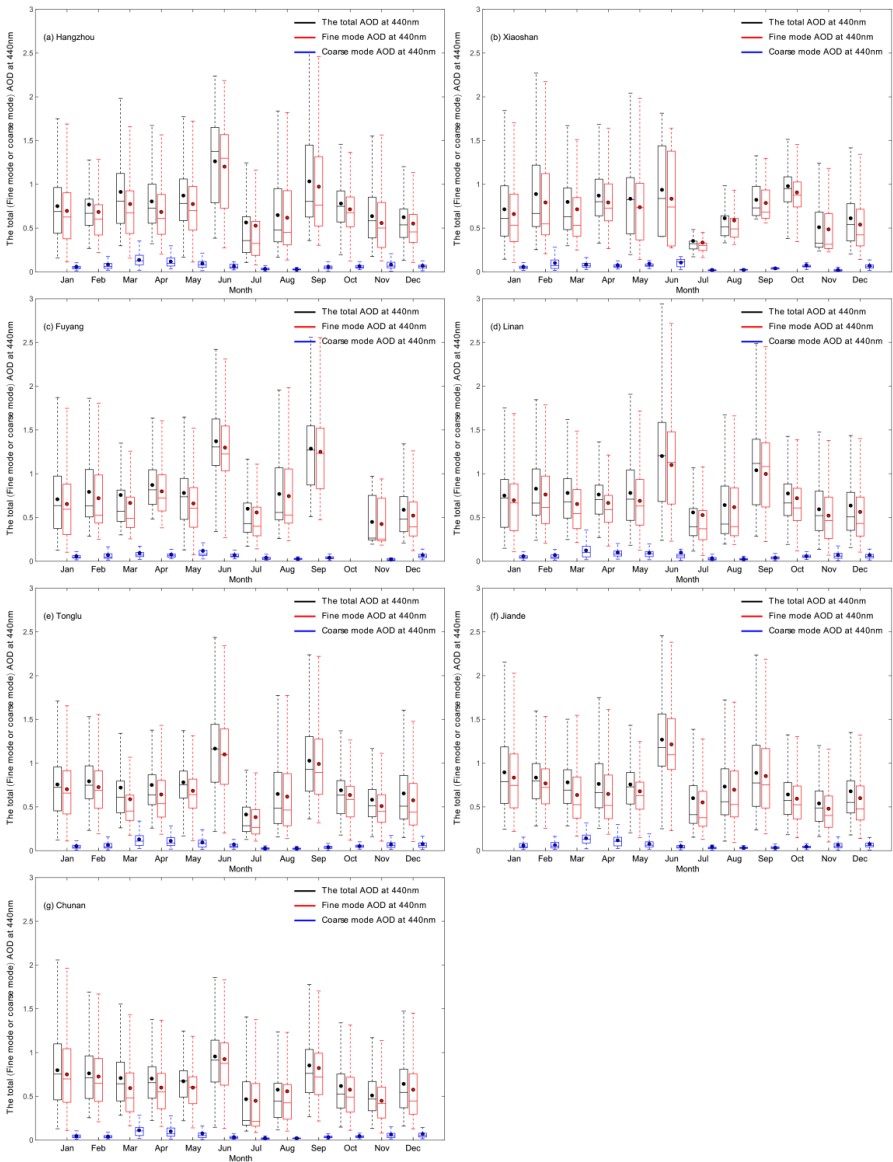


Fig. 2. Variation in the total, fine- and coarse-mode $AOD_{440\,nm}$ over (a) Hangzhou, (b) Xiaoshan,
(c) Fuyang, (d) LinAn, (e) Tonglu, (f) Jiande and (g) ChunAn. The boxes represent the 25th to
75th percentile distribution, while the dots and solid lines within each box represent the mean
and median, respectively.

Figure 3 shows that the annual extinction Angström exponent (EAE) at Hangzhou,




Xiaoshan, Fuyang, LinAn, Tonglu, Jiande and ChunAn was about 1.29±0.26, 1.37±0.24,
1.32±0.24, 1.29±0.27, 1.30±0.26, 1.32±0.28 and 1.22±0.25, respectively. Values of EAE >1.20
were found in all months throughout the year, indicating that small particle size distributions
were favored in the YRD region. The monthly average value of the EAE in Hangzhou was
higher in January (~1.40±0.23) and September (~1.43±0.24). This indicated the dominance of
small particles from anthropogenic emissions and agricultural activity in autumn and winter
(Tan et al., 2009). The EAE was lower in March (~1.16±0.24) and April (~1.13±0.22), which
reflects the effect of mineral dust aerosols (Gong et al., 2003). However, this effect is not as
obvious in the YRD region as other regions in north or northeast China.

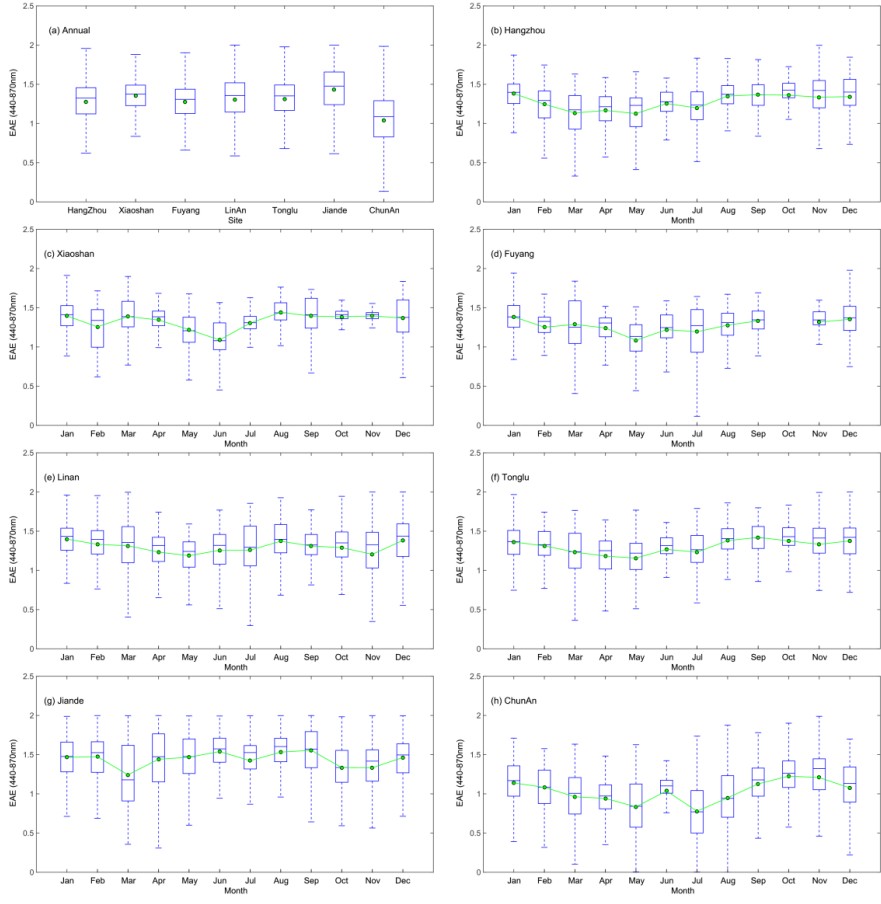


Fig. 3. (a) Annual variation in the EAE at 440–870 nm. Variation in the EAE at 440–870 nm



over (b) Hangzhou, (c) Xiaoshan, (d) Fuyang, (e) LinAn, (f) Tonglu, (g) Jiande and (h) ChunAn.
The boxes represent the 25th to 75th percentile distribution, while the dots and solid lines
within each box represent the mean and median, respectively.
Validation of the MODIS C6 retrieval AOD values was carried out by comparison with
ground-based observations (Figure 4). The systematic performance of the MODIS C6 retrieval
AOD values was generally stable in the YRD region, with most of the plots scattered around
the 1:1 regression line. The correlation coefficients ($R^2$) between the Terra-MODIS and sun
photometer AOD (550 nm) values were about 0.73, 0.83, 0.77, 0.89, 0.85, 0.81 and 0.86 at
Hangzhou, Xiaoshan, Fuyang, LinAn, Tonglu, Jiande and ChunAn, respectively. The linear
regression fitting performed better at the suburban sites of LinAn and Jiande. The fitting curve
was almost consistent with the 1:1 reference line, which suggests that the aerosol properties
were well defined for the MODIS C6 products. A large part of the MODIS retrieval AOD value
was outside the expected error envelope of $\pm (0.05 + 20\%\tau_{CARSNET})$, especially for AOD values
<0.80 in Hangzhou and Xiaoshan. This indicates that the MODIS retrieval algorithm could still
be improved, especially in urban areas. The MODIS retrieval AOD performed better at the
other five sites (Fuyang, LinAn, Tonglu, Jiande and ChunAn) in the YRD; most of the retrieved
AOD values for these sites fell within the expected error envelope. The MODIS retrievals were
overestimated at Hangzhou, Xiaoshan and ChunAn. This could be because the MODIS SSA
was underestimated at and near to urban sites (Tao et al., 2015). The small deviation at the
suburban sites suggested that the MODIS C6 retrieval method was suitable for capturing the
optical properties of aerosols in suburban areas of the YRD.





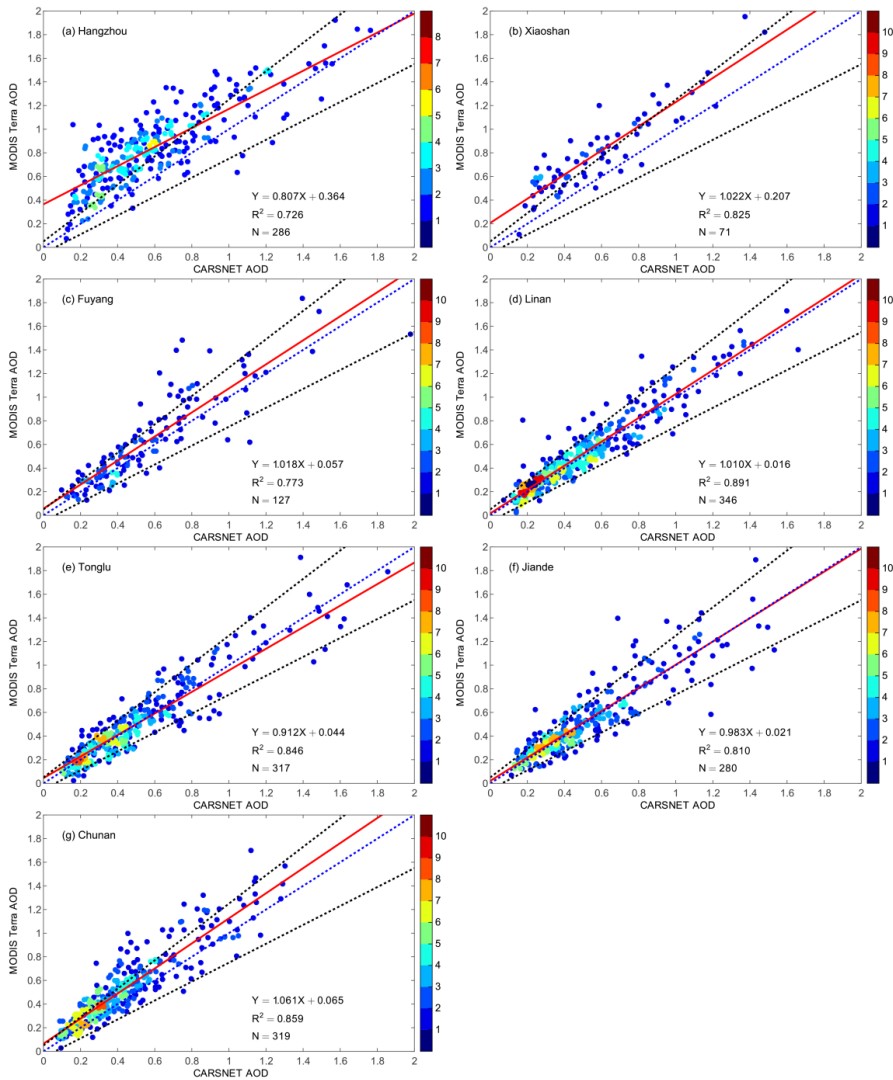


Fig. 4. Comparison of C6 MODIS AOD at 550 nm with the CARSNET AOD in (a)

Hangzhou, (b) Xiaoshan, (c) Fuyang, (d) LinAn, (e) Tonglu, (f) Jiande and (g) ChunAn. The red
solid line represents the linear regression. The two black dotted lines represent the expected
errors in the MODIS retrievals.

The relationship between the EAE and the spectral difference in the EAE

($\delta EAE = EAE_{440-675nm} - EAE_{675-870nm}$) was analyzed to investigate the contribution of fine



particles ($R_f$) and their fraction ($\eta$) to the total extinction (EAOD) at 440 nm (Gobbi et al., 2007).
In this framework, values of AOD>0.15 are represented by different colors to avoid errors in
the δEAE. The lines indicate contribution of the fixed radius ($R_f$) and fraction ($\eta$) of the
fine-mode particles to the total extinction. Gobbi et al. (2007) used the difference in the EAE
and AOD data to determine the growth of fine-mode particles or contamination by
coarse-mode particles at eight AERONET stations: Beijing (China), Rome (Italy), Kanpur
(India), Ispra (Italy), Mexico City (Mexico), NASA Goddard Space Flight Center (GSFC, USA),
Mongu (Zambia) and Alta Floresta (Brazil).
Fig. 5 shows that the high EAOD values (>1.00) cluster in the plots for all seven urban,
suburban and rural sites, which is attributed to fine-mode particles with δEAE <0 and $\eta$ ~50–
90%. This variation in the fine-mode particles is similar to the results from Beijing and Kanpur
($\eta$ ~70–90%). However, there were very few coarse-mode particles (δEAE~0, $\eta$~0–10%) in
this study, suggesting that the dominance of dust is not significant in eastern China. These
results showed a different pattern from that of other regions in north/northeast China (Wang et
al., 2010; Zhu et al., 2014). For δEAE ~0 and 10%<$\eta$<30%, high extinction was associated
with a mixture dominated by fine-mode particles and less persistent coarse-mode particles.
Clustering concentrated in the region α~1.5, δα ~−0.5 with high AOD values at all sites, which
may be linked to an increase in size of the fine-mode particles by coagulation as they aged
and hygroscopic events, as seen at other locations (e.g. Ispra, Italy; Mexico City, Mexico;
GSFC, USA).





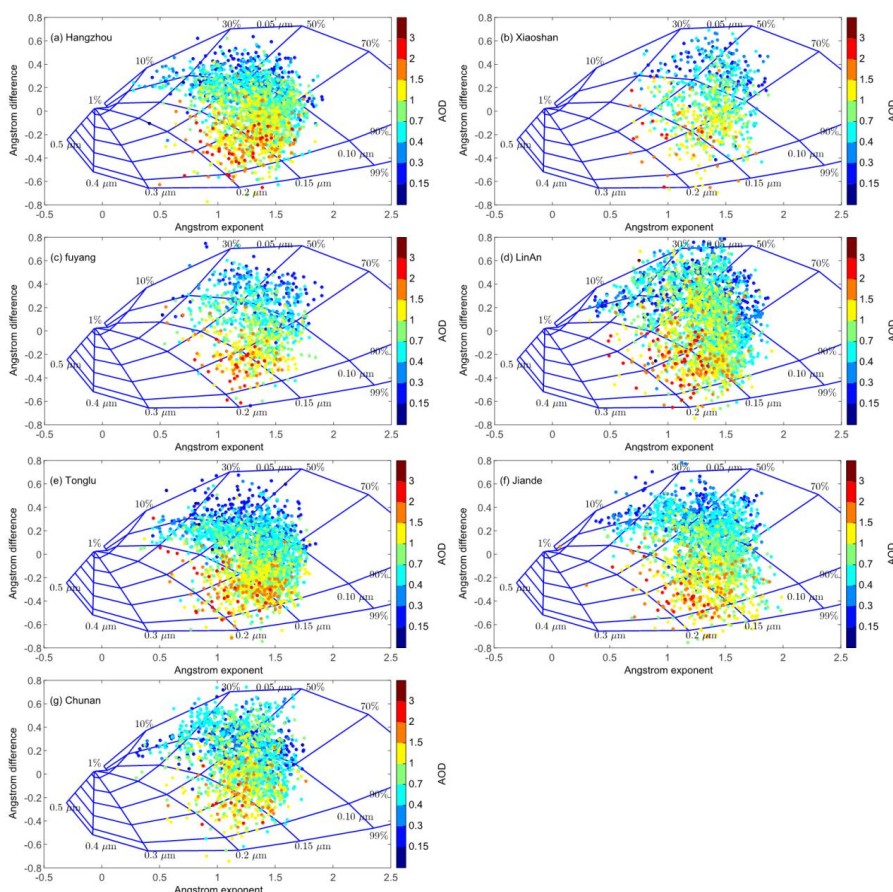


Fig. 5. Angström exponent difference as a function of $\alpha_{440-870\,nm}$ and the $AOD_{440\,nm}$ over (a)
Hangzhou, (b) Xiaoshan, (c) Fuyang, (d), LinAn, (e) Tonglu, (f) Jiande and (g) ChunAn.
**3.2 Single-scattering albedo and aerosol complex refractive index**
The distribution of the total, fine- and coarse-mode SSAs at seven sites in the YRD are
shown in Fig. 6. The total SSA varied from 0.91 to 0.94, which is similar to the range seen in
other regions of China, such as Wuhan (0.92), Beijing (0.89) and Xinglong (0.92) (Wang et al.,
2015; Xin et al., 2014; Zhu et al., 2014). This indicated that scattering aerosol particles in
eastern China resulting from high levels of industrial and anthropogenic activity were dominant.
The characteristics of the SSA at these seven sites gradually increased from the east coast
(0.91±0.06 at Hangzhou) inland toward the west (0.94±0.03 at ChunAn). These results



indicate the emissions caused by human activity affect the absorption of aerosols in urban
areas. The SSA was higher at LinAn and ChunAn than at the other sites, which may reflect the
presence of a larger number of scattering aerosols (e.g. particles from urban/industrial
activities) over the regional background/rural sites than over urban or suburban sites. The SSA
over urban and suburban sites showed the largest monthly variation. The monthly average
values of $SSA_t$ were high in February (~0.94±0.05) and June (~0.92±0.06), but low in March
(~0.90±0.06) and August (~0.89±0.09) in Hangzhou. However, the monthly SSA values at the
rural site of ChunAn only varied from 0.92 to 0.95. We concluded that the type of aerosol at
urban/suburban sites was more complex than at rural sites.

The range of variation in the SSA of fine particles ($SSA_f$) was 0.93–0.95, whereas the SSA

for coarse-mode particles ($SSA_c$) was 0.81–0.84 at the seven sites (Fig. 6). The fine- and
coarse-mode particles displayed significant scattering and absorption abilities in the urban,
suburban and rural areas of the YRD region. Fig. 6 shows a significant decrease in the
fine-mode SSA in July/August and in the coarse-mode SSA in March/April. At Hangzhou, the
lower fine-mode SSA values in July/August (~0.92±0.08/~0.90±0.08) were probably a result of
aerosols from biomass burning and the lower coarse-mode SSA values in March/April
(~0.79±0.08/~0.81±0.07) may reflect the existence of light-absorbing dust aerosols (Yang et al.,
2009). The SSA depends on the wavelength and dust particles absorb strongly at short
wavelengths, resulting in a lower SSA at 440 nm (Eck et al., 2010). The absorption/scattering
properties of fine- and coarse-mode particles determine the total SSA in the YRD. These
differences in the SSA were mostly dependent on the type of aerosol and the ratio of absorbing
and non-absorbing components in the aerosols.





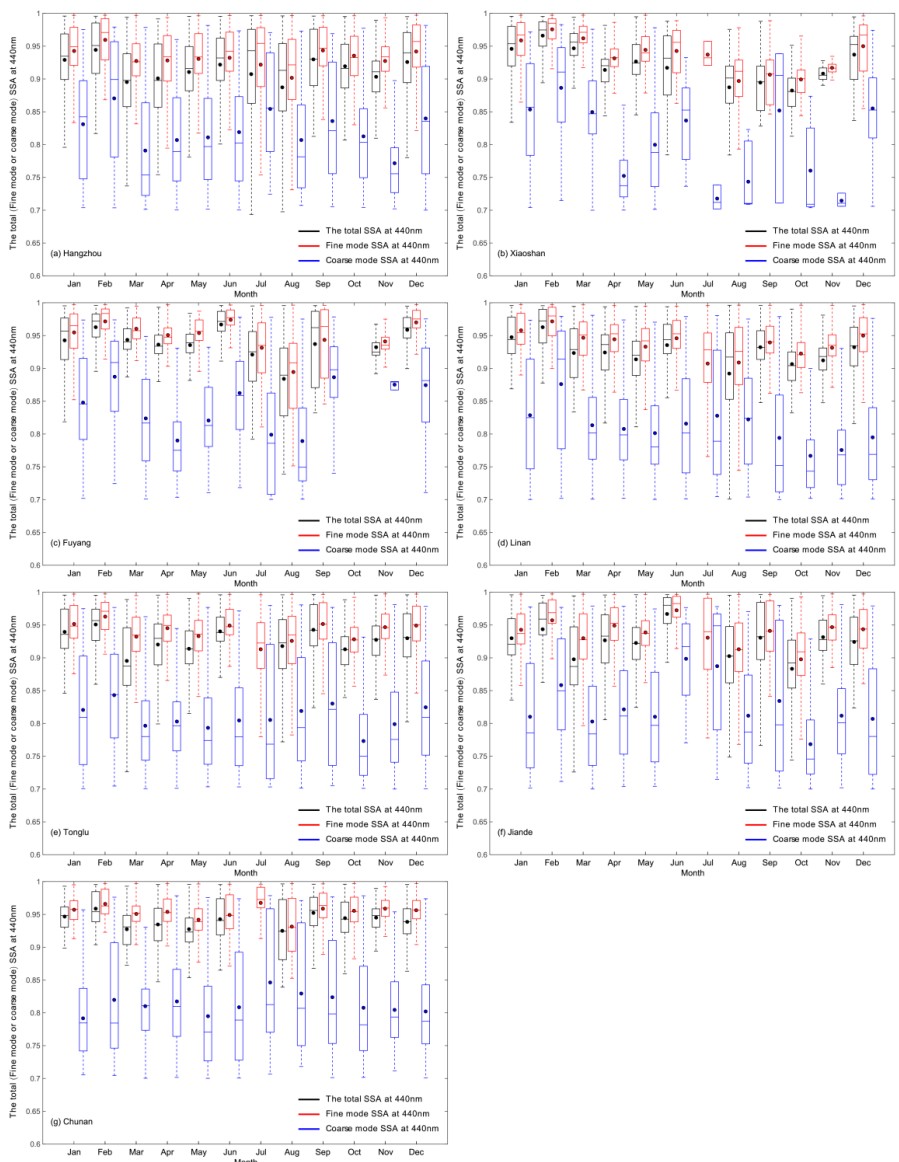


Fig. 6. Variation in the total, fine- and coarse-mode $SSA_{440\ nm}$ over (a) Hangzhou, (b) Xiaoshan,
(c) Fuyang, (d) LinAn, (e) Tonglu, (f) Jiande and (g) ChunAn. The boxes represent the 25th to
75th percentile distribution, while the dots and solid lines within each box represent the mean
and median, respectively.

The real and imaginary parts of the refractive index represent the scattering and



absorption capacity of particles, respectively. The refractive index is determined by the
hygroscopic conditions and the chemical composition of the aerosols (Dubovik and King,
2000). There was no significant difference between the real parts of the refractive index among
the seven urban, suburban and rural sites in this study (range 1.41–1.43). The real parts of the
refractive index in this study were smaller than the real parts of ammonium sulfate and
ammonium nitrate (1.55), which may be due to the hygroscopic conditions or the mixture of
dust particles. The real part of the refractive index was highest in March (~1.46±0.06) and
November (~1.45±0.06) and lowest in July (~1.42±0.06) and August (~1.41±0.07) at the urban
sites. A higher level of dust aerosols with weak scattering in spring and autumn could
contribute to a higher value of the real part of the refractive index; this was reduced or
eliminated by rainfall during the summer months.

The imaginary part of the refractive index was higher at the urban site of Hangzhou

(~0.0112 ± 0.0104) as a result of the high loading of absorption aerosols in this region and was
consistent with the lower SSA. High imaginary parts of the refractive index occurred in August
at all urban, suburban and rural sites in the YRD, which may be due to the higher emission of
absorptive particles by the post-harvest burning of crop residues. The burning of crop residues
may cause a large deterioration in the regional air quality in the YRD region.
**3.3 Radius and aerosol volume size distributions**

Fig. 7 shows the monthly aerosol size distribution (d$V$/dln$r$) in the YRD for all sites. The

volumes of fine-mode aerosols were obviously higher than those of coarse-mode aerosols
over all sites. The fine-mode radii were ~0.2–0.3 μm in the YRD with a volume of 0.10–0.12
μm$^3$ and the coarse-mode radii were ~2.0 μm with a volume close to 0.07 μm$^3$. The amount of
fine-mode aerosols was higher in June and September than in other months at almost sites,
except for Xiaoshan. This could be caused by aerosol humidification (Eck et al., 2012; Li et al.,
2010, 2014; Huang et al., 2016). This phenomenon is also found over Bejing and Shenyang in
north/northeast China, suggesting that hygroscopic growth occurs over many regions of China
(Li et al., 2011; Che et al., 2015c).




The coarse-mode radius in spring at all sites was smaller than in other cities in north and
northeast China affected by frequent dust transport events in spring (Kong et al., 2011; Zhao et
al., 2015). The coarse-mode particles showed a larger effective radius at all seven urban,
suburban and rural sites in the summer, which may due to the adhesion of new particles onto
larger particles (such as fly ash).

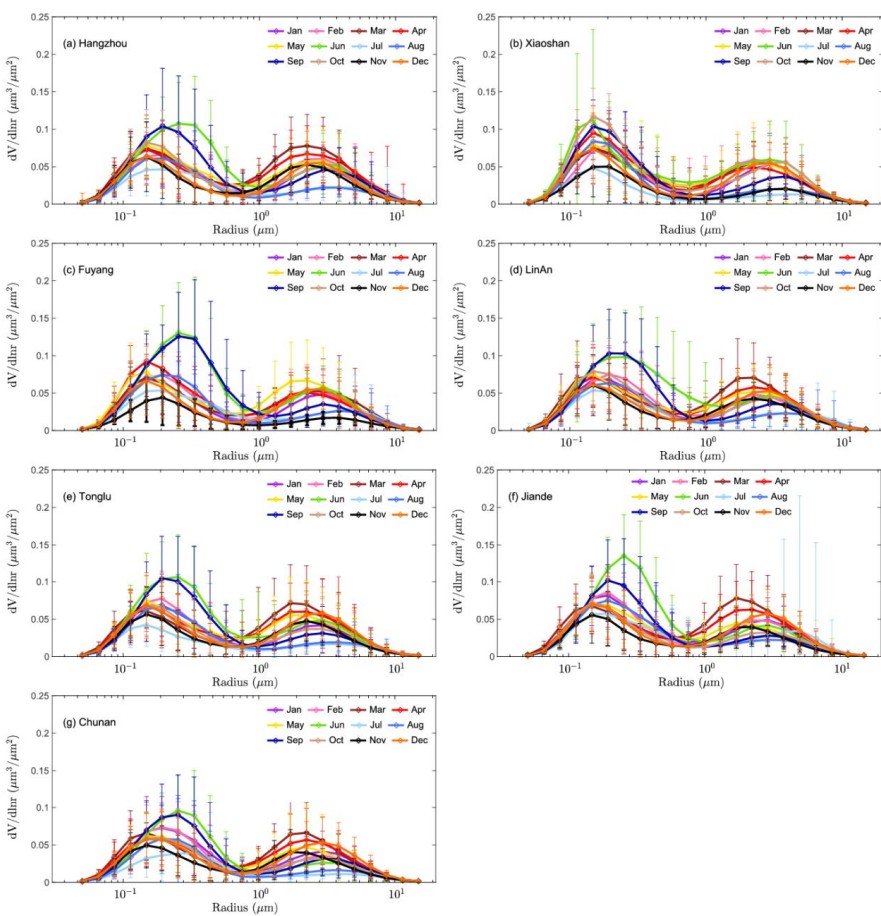


Fig. 7. Variation in the annual volume size distribution over (a) Hangzhou, (b) Xiaoshan, (c)
Fuyang, (d) LinAn, (e) Tonglu, (f) Jiande and (g) ChunAn.
**3.4 Absorption aerosol optical depth and absorption Angström exponent**
The annual AAODs at Hangzhou, Xiaoshan, Fuyang, LinAn, Tonglu, Jiande and ChunAn


were about 0.06±0.05, 0.05±0.04, 0.04±0.04, 0.05±0.04, 0.05±0.04, 0.06±0.04 and 0.04±0.03,
respectively (Fig. 8). The higher annual values of the AAOD in Hangzhou and Jiande indicate
that there are more absorbing aerosol particles at these sites. The similar AAOD level at the
seven sites suggests that absorbing aerosols are distributed homogeneously in the YRD
region. The monthly AAOD at the urban site of Hangzhou was 0.09±0.06 in March as a result
of the presence of absorbing dust particles. The AAOD of about 0.07±0.04 in August is related
to the burning of crop residues. The AAODs in the winter season at all the sites in the YRD
region were <0.05, which suggests that absorbing aerosol emissions did not frequently occur
at these sites, unlike in the northern regions of China.

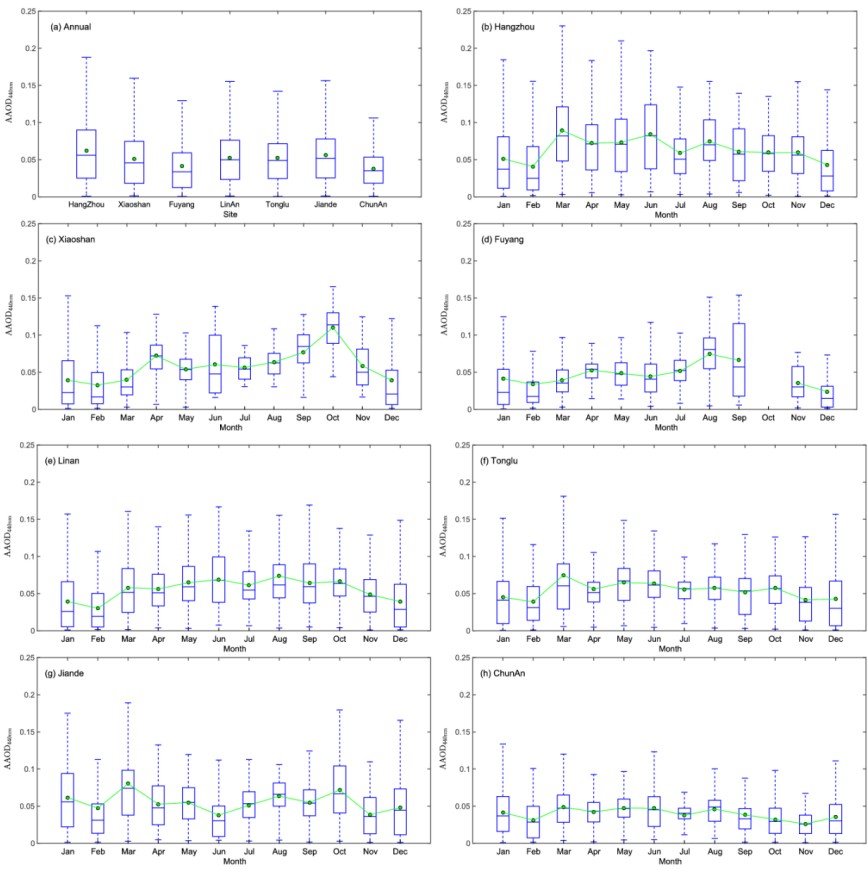


Fig. 8. (a) Annual variation in the absorption aerosol optical depth at 440 nm ($AAOD_{440\ nm}$) over





(b) Hangzhou, (c) Xiaoshan, (d) Fuyang, (e) LinAn, (f) Tonglu, (g) Jiande and (h) ChunAn. The
boxes represent the 25th to 75th percentile distribution, while the dots and solid lines within
each box represent the mean and median, respectively.
The annual mean values for the AAE at Hangzhou, Xiaoshan, Fuyang, LinAn, Tonglu,
Jiande and ChunAn were about 1.13±0.46, 0.88±0.42, 0.85±0.43, 0.98±0.35, 1.11±0.49,
1.16±0.44 and 0.93±0.31, respectively (Fig. 9). The mean values of the AAE at Xiaoshan and
Fuyang were <1.00, suggesting the presence of absorbing or non-absorbing materials coating
black carbon at these suburban and rural sites (Bergstrom et al., 2007; Lack and Cappa et al.,
2010; Gyawali et al., 2009). The AAE values were close to 1.00 at LinAn and ChunAn,
indicating that the absorptive aerosols were dominated by particles of black carbon (Zhang et
al., 2012; Li et al., 2016). By contrast, the AAE values at Hangzhou, Tonglu and Jiande
were >1.00, indicating the presence of absorptive aerosols from the burning of biomass. This
difference in the AAE distribution indicates the absorbing aerosols have different
characteristics resulting from the different emission sources at urban, suburban and rural sites
in the YRD. The AAE was <1.00 in June – August at all urban, suburban and rural sites of the
YRD, which suggested the presence aerosols coated with absorbing or non-absorbing
material in summer season. This process is favored by high temperatures and high humidity
under conditions of strong solar radiation (Shen et al., 2015, Zhang et al., 2015). The particles
coagulate and grow rapidly in the presence of sufficient water vapor (Li et al., 2016). The AAE
became increasingly close to, or larger than, 1.00 at all seven sites from September, which is
consistent with decreasing amounts of precipitation. This increase in the AAE was related to
the emission of black carbon from biomass burning (Soni et al., 2010; Russell et al., 2010).

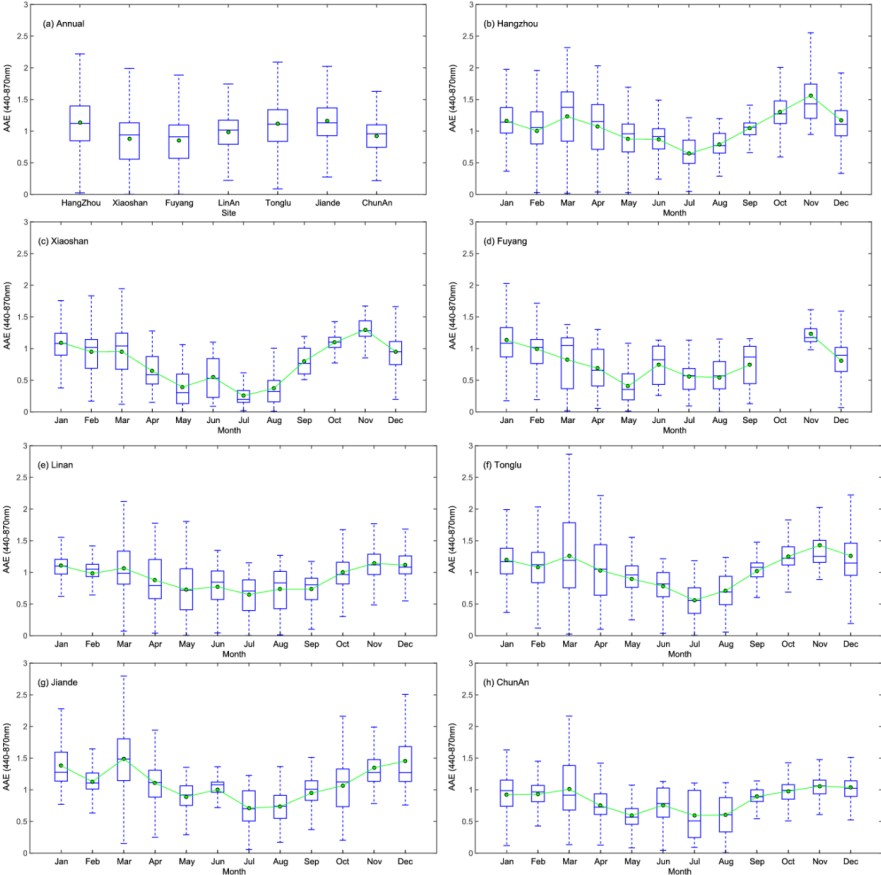


Fig. 9. (a) Annual variation in the absorption Angström exponent at 440 nm (AAE$_{440\,nm}$) over (b)

Hangzhou, (c) Xiaoshan, (d) Fuyang, (e) LinAn, (f) Tonglu, (g) Jiande and (h) ChunAn. The

boxes represent the 25th to 75th percentile distribution, while the dots and solid lines within

each box represent the mean and median, respectively.

The AAE can be used to indicate the major types (urban/industrial, biomass burning,

dust/mixed dust) or optical mixtures of absorbing aerosol particles (Schnaiter et al., 2006;

Russell et al., 2010; Giles et al., 2011; 2012; Mishra and Shibata, 2012). Giles et al., (2011)

examined AAE/EAE data from Kanpur to classify the categories of absorbing aerosols. The

"mostly dust" category has been defined as having an EAE value ≤0.50 and sphericity fraction

<0.20 with an AAE value >2.00. The "mostly black carbon" category has been defined as





having an EAE value >0.80 and a sphericity fraction ≥0.20 with 1.00<AAE≤2.00. Values of
EAE >0.80 and AAE >2.00 indicate a concentration of organic carbon (Arola et al., 2011). The
"mixed black carbon and dust" category was centered at EAE ~0.50 with AAE ~1.50 and used
to represent an optical mixture with black carbon and mineral dust particles as the dominant
absorbers.

We used the instantaneous AAE and EAE values to classify the dominant absorbing

aerosol types in urban, suburban and rural areas of the YRD (Fig. 10; Table 2). Table 2 shows
that the "mostly dust" category was very low at both suburban and rural sites (<0.01%) and just
~0.24% at the urban site of Hangzhou. This indicates that dust does not dominate the
absorbing aerosol particles in the YRD region of eastern China, which is completely different
from other regions of north/northeast China. The "mostly black carbon" category dominates the
absorbing aerosols in the urban, suburban and rural areas in the YRD region. The percentage
"mostly black carbon" varied from ~20 to 40% depending on each site, indicating the mixing of
black carbon as well as brown and soot carbon species from biomass burning and
urban/industrial activities. Because of the long-distance transportation and local fugitive dust
effect, the "mixed black carbon and dust" category contributed ~5% of the absorbing aerosol
particles in the YRD region. There were also ~1–4% of the "organic carbon" category
identified as absorbing aerosol particles in this region. Particles with EAE values of ~0.40 and
~1.25 could be regarded as "mixed large particles" greater than microns in size and submicron
"mixed small particles", respectively (Giles et al. 2012). The frequency of "mixed large particles"
was <0.5% at the urban, suburban and rural sites (Table 2). By contrast, the frequency of
"mixed small particles" was ~18–36%.

The EAE ($\alpha_{ext}$) and AAE ($\alpha_{abs}$) values at all the urban, suburban and rural sites were

distributed mainly around 1.25 and 1.00–1.50 (Fig. 10), respectively. In contrast with the
results of Giles et al. (2011), the sphericity fraction did not show an obvious transition from
non-spherical to spherical particles from the urban, suburban and rural sites in YRD. The
sphericity fraction showed a dispersed distribution of spherical particles, indicating a mixture of
fine-mode particles derived from anthropogenic sources and coarse-mode particles, such as




dust events transported from north/northwest China or local fugitive dust emissions.

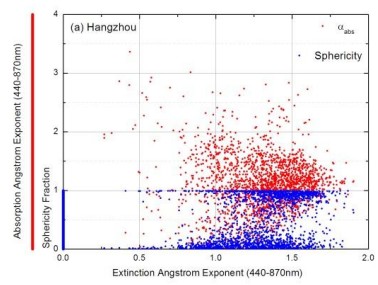 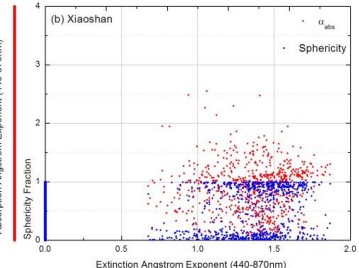


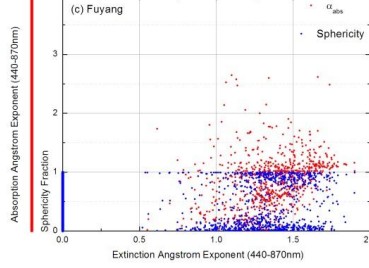 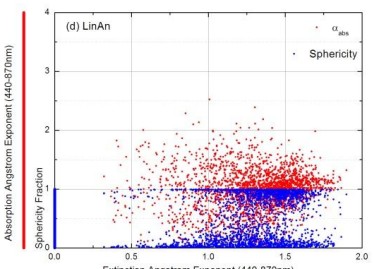


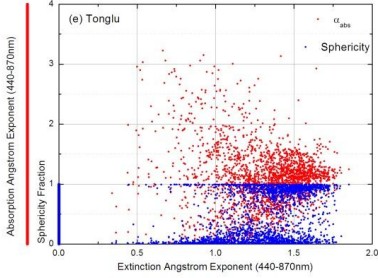 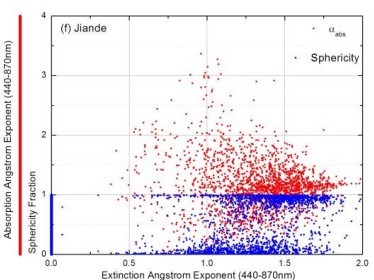


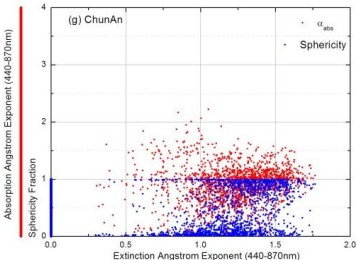






Fig. 10. The AAE and the sphericity fraction as a function of the EAE at 440–870 nm over (a)
Hangzhou, (b) Xiaoshan, (c) Fuyang, (d), LinAn, (e) Tonglu, (f) Jiande and (g) ChunAn.
Table 2. Types of aerosol at the seven sites in the Yangtze River Delta.

|  | Mostly dust (%) | Mixed black carbon and dust (%) | Mostly black carbon (%) | Organic carbon (%) | Mixed large particles (%) | Mixed small particles (%) |
|---|---|---|---|---|---|---|
| Hangzhou | 0.24 | 6.14 | 34.68 | 2.58 | 0.19 | 36.34 |
| Xiaoshan | <0.01 | 2.93 | 27.00 | 0.80 | <0.01 | 23.40 |
| Fuyang | <0.01 | 1.21 | 19.51 | 1.10 | <0.01 | 18.63 |
| LinAn | <0.01 | 6.18 | 28.91 | 0.50 | 0.37 | 28.04 |
| Tonglu | <0.01 | 4.92 | 34.26 | 3.55 | 0.18 | 33.33 |
| Jiande | <0.01 | 6.71 | 40.04 | 3.23 | 0.26 | 35.28 |
| ChunAn | <0.01 | 7.16 | 24.15 | 0.23 | 0.12 | 26.75 |


**3.5 Aerosol radiative forcing at the Earth's surface and top of the atmosphere**
Figures 11 and 12 show the variations in ARF at the surface (ARF-BOA) and at the top of
the atmosphere (ARF-TOA) at the urban, suburban and rural sites in the YRD region.
The annual ARF-BOA values for Hangzhou, Xiaoshan, Fuyang, LinAn, Tonglu, Jiande and
ChunAn were about −93±44, −84±40, −80±40, −81±39, −79±39, −82±40 and −74±34 W/m$^2$,
respectively. The higher ARF-BOA values in Hangzhou indicate that there was high aerosol
loading at this site, which scattered and absorbed more radiation and caused a significant
cooling effect at the surface. The monthly value of the ARF-BOA in Hangzhou was higher in
June (about −132±48 W/m$^2$) and September (about −106±48 W/m$^2$), which is consistent with
the timing of burning biomass from crop residues. Ding et al. (2016) found that black carbon
emitted from biomass burning can modify the meteorology of the planetary boundary layer and
substantially decrease the surface heat flux. Hygroscopic growth at the same time enhances



the aerosol optical extinction (Yan et al., 2009; Zhang et al., 2015); this was also an important
factor in the large ARF-BOA values in June and September at the urban, suburban and rural
sites in the YRD.

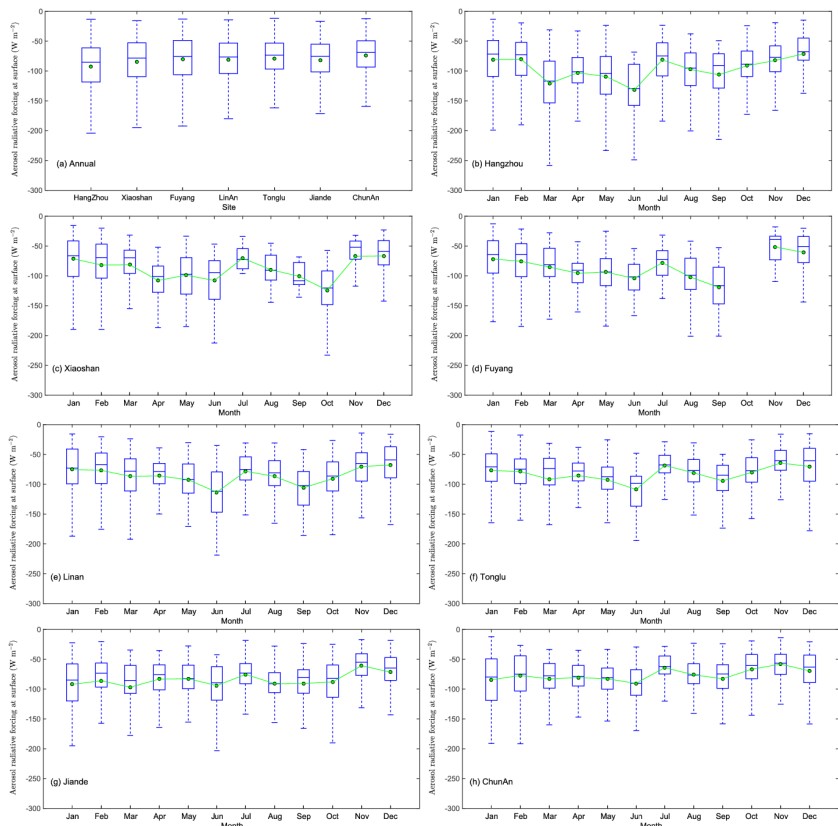


Fig. 11. (a) Annual variation of the ARF at the surface over (b) Hangzhou, (c) Xiaoshan, (d)
Fuyang, (e) LinAn, (f) Tonglu, (g) Jiande and (h) ChunAn. The boxes represent the 25th to 75th
percentile distribution, while the dots and solid lines within each box represent the mean and
median, respectively.
The ARF-TOA values were less than −40 W/m$^2$ at the urban, suburban and rural sites in
the YRD. The AFR-TOA values were negative all year, which suggests that the aerosols
caused a cooling effect at the TOA as well as at surface in the YRD. This is different from the





north/northeast regions of China, where the instantaneous AFR-TOA value can be positive in
the winter season as a result of the large surface area reflecting short wavelength radiation
and heating caused by absorbing aerosols (Che et al., 2014). The surface albedo in the YRD
region is lower than in north/northeast China as a result of better vegetation. At the same time,
there is also a low level of absorbing aerosol emissions in winter. This caused obvious
negative AFR at the TOA at the urban, suburban and rural sites in the YRD.

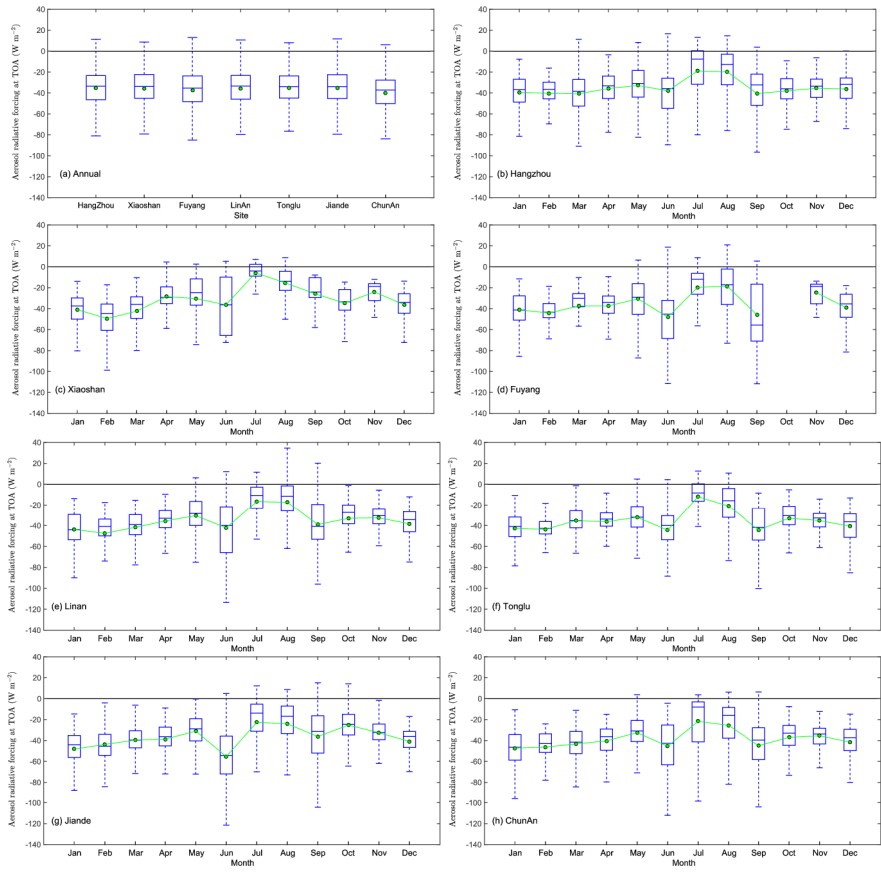


Fig. 12. (a) Annual variation in the aerosol radiative forcing at the top of the atmosphere (TOA)
in (b) Hangzhou, (c) Xiaoshan, (d) Fuyang, (e) LinAn, (f) Tonglu, (g) Jiande and (h) ChunAn.
The boxes represent the 25th to 75th percentile distribution, while the dots and solid lines
within each box represent the mean and median, respectively.



**4. Summary and discussion**

The aerosol optical properties, including the AOD, EAE, SSA, complex refractive index, volume size distribution, and the absorption properties of the AAOD and AAE were retrieved from satellite data over the YRD in eastern China for the period 2011–2015.

Aerosol loading was at a high level over both urban and suburban sites and even over the rural sites in the YRD, which suggests that pollution from aerosols is not just local, but has occurred at a regional scale over eastern China in recent years. The AOD showed a decreasing trend from the east coast inland to the west as a result of contributions from anthropogenic activity. Hygroscopic growth and the burning of biomass from crop residues in the summer season could cause this obvious increase in the AOD. The ratio of $AOD_f/AOD_t$ was consistently >0.90, indicating that fine-mode particles made a major contribution to the total AOD in the YRD. The relationship between the EAE and the spectral difference in the EAE suggested that the dominance of dust is not important in eastern China. The MODIS C6 AOD retrievals performed better in suburban than in urban and rural areas, but were systematically overestimated in rural and urban areas and their immediate surroundings. A large part of the MODIS retrieval AOD was outside the expected error, especially at AOD values <0.80 in urban areas and their immediate surroundings.

The range of variation of the total, fine- and coarse-mode SSA values was 0.91–0.94, 0.93–0.95 and 0.81–0.84, respectively, in the YRD region, suggesting the presence of mainly scattering aerosol particles in eastern China as a result of high industrial and anthropogenic activity. The fine- and coarse-mode particles showed significant scattering and absorption in the urban, suburban and rural areas of the YRD region. The imaginary part of the refractive index was larger at urban sites as a result of the high loading of absorption aerosols. The large imaginary parts occurring in August may be due to the higher emission of absorptive particles from the post-harvest burning of biomass.

The similar AAOD levels at the seven sites indicated that absorbing aerosols were homogeneously distributed in the YRD region. The low AAODs in the winter season suggest



fewer absorbing aerosol emissions at the urban, suburban and rural sites. The difference in
the distribution of the AAE suggests that the absorbing aerosols have different characteristics
depending on the emission source. Hygroscopic growth not only contributed to the high
aerosol extinction values, but also increased the size of the fine-mode particles in the summer
in the YRD region. The "mostly black carbon" category was the dominant contributor of
absorbing aerosols at the urban, suburban and rural sites in the YRD region. The submicron
"mixed small particle" category had a significant effect on the aerosol optical properties over
the YRD region. The sphericity fraction showed a dispersed distribution of spherical particles,
indicating a mixture of both fine- and coarse-mode particles from anthropogenic and natural
sources.
The large ARF-BOA indicated a high aerosol loading that scattered and absorbed more
radiation. It also showed that the cooling effect of the aerosols at the surface was stronger in
the YRD region. Both the burning of biomass from crop residues and the hygroscopic growth
of particles could make important contributions to the ARF-BOA in summer over the YRD
region. The AFR-TOA values were negative all year, suggesting that the aerosols had a
cooling effect at the TOA.
The column aerosol optical properties over urban, suburban and rural areas of YRD
region of China were investigated and the results will increase our understanding of the
characteristics and sources of aerosol emissions over eastern China. Future research should
consider the vertical distribution of aerosols by Lidar, the validation of the aerosol optical
results of other satellite products such as VIIRS and GOCI, and a comprehensive analysis of
the physical and chemical properties of aerosols and meteorological factors.
**Acknowledgments**
This work was supported by grant from National Key R & D Program Pilot Projects of
China (2016YFA0601901), National Natural Science Foundation of China (41590874
&41375153), Natural Science Foundation of Zhejiang Province (LY16010006), the CAMS
Basis Research Project (2016Z001 & 2014R17), the Climate Change Special Fund of CMA




(CCSF201504), CAMS Basic Research Project (2014R17), the Special Project of Doctoral
Research supported by Liaoning Provincial Meteorological Bureau (D201501), Hangzhou
Science and Technology Innovative project (20150533B17) and the European Union Seventh
Framework Programme (FP7/2007-2013) under grant agreement no. 262254.

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
