# Peer review of "Aerosol optical properties and instantaneous radiative forcing based on high temporospatial resolution CARSNET ground-based measurements over eastern China"

_Atmospheric Chemistry and Physics, 2017_

## Referee Comment (RC1) · Anonymous Referee #1 · 28 Jul 2017

General comments:

Chen et al. reported sunphotometer measurements of aerosol microphysical and optical properties and modeling of aerosol surface and top-of-the-atmosphere (TOA) radiative forcing (RF) at several ground stations of eastern China. This manuscript is poorly written and needs a major overhaul. In many places, discussions on the results are groundless without providing any backing evidence or appropriate references to the literature. Most of the discussions is too superficial to provide any value or interest to the aerosol-climate-change or air-pollution communities. The logical flow is confusing

and unnatural, with potential for improvement in the organization of sections. Some sections need to be rewritten and reorganized to remove redundancy, while additional information must be provided in a few places.

Detailed comments (Line number: L):

Title of the manuscript needs to be changed. Title implies that the study reports aerosol instantaneous radiative forcing, while monthly and annual RF are shown instead. Remove "instantaneous". "Temporospatial" is awkward. Consider using "spatiotemporal" instead. Even worse, the author mentioned nothing whatsoever in the manuscript about the purpose or the advantage of such high temporospatial data (more comments on this later on). "Seven sites" should not be treated as "high spatial resolution". Consider writing the full name of CARSNET in the title.

Abstract needs to be rewritten. The common flow of logic is to discuss the aerosol microphysical properties (size, refractive index) first, then optical properties (Kext, FMF, SSA, etc), and finally RF. This comment also applies to the main body text. I understand that the author is trying to follow the way how these variables are derived from the AERONET inversion algorithm. I do not think this is necessary. The result subsections need to be reorganized, combined, and shortened to make the paper more concise and less confusing.

L38: what wavelength is the AOD value? If it is 440 nm, why not use 500 nm or 550 nm to make the result more useful for the community and more easily comparable to past studies? Optical properties are wavelength dependent. Provide this information when introducing any variables at the first time.

L40: use the term "fine mode fraction" if that is what you mean here.

L43: The Pearson correlation coefficient is noted as R, not R^2 (which is called coefficient of determination).

L49-50, use volume fraction instead.

L52-54: reads like a method description (remove it), and what are the key findings?

L54-56: this statement is too general to be included in an abstract. Abstract is to present the most original and important findings of the study. Do not simply report the results as general statements. Writing an interesting abstract is a critical step to pass a peer review.

Section 1

Introduction needs to polished. Always write out the acronyms at their first appearance. This comment applies to the rest of the manuscript.

L92: it looks like the author wants to emphasize the advantage of using multi-site, multi-year, as well as three-minute-interval (L35) or high-frequency (L97) data; however, not in a single place of the manuscript the author discusses the advantage of such data. After all, all sites seemed to have similar aerosol characteristics. And all results are presented as monthly or annual means. Consider adding analysis of diurnal cycle and interannual variations, or exploring other unique aspects of the data.

Section 2.

L130: Given the high AOD (comparable to urban site), ChunAn is nothing like a background site. (L177)

L146-157: describe the assumptions of the inversion algorithms, some of which may not apply for eastern China. Also describe the accuracies of the inversion variables (which are essentially a remote sensing product).

L159-161: Need more descriptions on the technical details of radiative transfer calculations. For example, how is the atmospheric profile treated and where are the profile data coming from? How is cloud scattering treated? How are clouds and aerosols vertically distributed? How is the surface and TOA radiative forcing defined?

L162-172: Why not use Aqua? I understand that MODIS/Aqua is more stable and

has better calibration than Terra (corrects me if I am wrong). It seems that the author calculated AOD at 550 nm in order to validate MODIS. If this is the case, why not show AOD at this wavelength for the entire manuscript (e.g. Table 1)?

Section 3.

L177: from Table 1, Xiaoshan and Fuyang have far smaller observation samples than other sites. Why? Does this affect the comparability between the sites? ChunAn AOD is only 10% less than other sites. I would not say it is a representative background site.

L180-188: this part has serious redundancy. Rewrite it to be more concise.

L188-206: It is dangerous to use AOD solely as a metric for pollution (PM) or emission level. Keep in mind that AOD is an optical quantity (at a specific wavelength) and measures the light extinction of a vertical atmospheric column from the ground to TOA. The AOD difference between sites may not reflect the ambient pollution severity if some sites are affected by transported events. The AOD difference between cities may not reflect the emission sources if the meteorological conditions are so different to affect the dispersion and lifetime...The author needs a more solid analysis here.

L225-230: Another example of redundancy. Rewrite to be more concise.

L231-235: Another example of unconvincing analysis. Is the transported dust event verified? Please describe the specific events (2012-2015) rather than citing a climatological study. I suspect that fugitive dust from road traffic or construction activity is a more persistent and significant source for China's cities.

L241-250: The findings based on EAE are very similar to those from FMF (L222-235). Consider merging them to be more concise.

L322-334: SSA depends on two factors - particle size and composition. It is expected that coarse mode SSA is less than fine mode SSA. I do not see the need for two SSAs for fine and coarse modes, separately. This paragraph presents no new findings and can be removed.

L349-351: The real part of refractive index is related to scattering, while the imaginary part to absorption. Does the author mean to say spring dust absorption increases the imaginary part (not real part)?

L377-378: AAOD values have very large uncertainties (standard deviation is comparable to mean). Why? Given these uncertain estimates, it would be difficult to make a robust comparison between sites.

L392, this paragraph seems redundant as the later part from L416 discusses EAE and AAE links to aerosol types.

L430-432: explain why the YRD region is "completely different from north/northeast China"

Figure 10: really bad visualization. Consider other options, like histograms.

L483: do you intend to say "large surface reflectance (or albedo)"

Section 4 should be shortened. There is no need to repeat all results. There should be one or wo paragraphs discussing the implications of the most important findings.

––––––––––––––––––––––––––––––––

---

## Referee Comment (RC2) · Anonymous Referee #2 · 31 Jul 2017

In this paper, the authors analyzed aerosol optical and physical properties derived from CARSNET ground-based measurements for 7 locations in the RYD region. Ground-based observations are definitely welcomed in the community. However, this paper needs major revisions. Firstly, also suggested by the first reviewer, this is a poorly written paper. Also, there are several major technical issues with the paper. The first reviewer has done a great job highlighting issues and I have added several more comments below.

Aerosol radiative forcing values are computed using a radiative transfer model (Global

Atmospheric ModEl, GAME). However, no details are provided on their radiative trans-
fer modeling efforts. For example, how do they define surface (broadband) character-
istics? What are the required aerosol properties such as vertical distributions? The
CARSNET observations are in discrete channels, how the authors perform a narrow-
band to broad band conversion? A detailed uncertainty analysis is also needed but is
lacking.

The authors compared CARSNET measurements with a merged MODIS Deep Blue
and Dark Target product. Given that the MODIS Deep Blue and Dark Target methods
are fundamentally different in their retrieval processes, I would recommend the authors
evaluating each product separately.

Also, in lines 248-249, it states "The EAE was lower in March ($\sim$1.16$\pm$0.24) and April
($\sim$1.13$\pm$0.22), which reflects the effect of mineral dust aerosols (Gong et al., 2003)."
This seems to contradict to a later conclusion (Table 2) that the dust aerosol presence
is insignificant for the region. This actually brings up an issue, as the authors try to
compare mean properties of the 7 sites and try to provide explanations for the differ-
ences. Some explanations are weak with little or no supporting evidence. In addition,
the differences in some of the mean properties are actually way smaller than variations
(numbers after $\pm$ sign) of the data, and thus some statistical methods are needed to
back up the authors' comments with consideration of data spreads.

Line 108-109, I am not sure what the authors mean by "Levy et al. (2013) refined the
MODIS Collection 6 (C6) aerosol retrieval process to provide better AOD retrievals".
What is "better AOD retrieval"? May be the authors referring to more accurate AOD
retrievals?

Line 135 "Jiande, Xiaoshan Tonglu and Xiaoshan" should be "Jiande, Xiaoshan, Tonglu
and Xiaoshan"?

Lines 143-144, "Instantaneous direct data for the AOD were selected at least ten times
each day at temporal resolution of about three minutes" Define "direct data". Also, what

are the section criteria? Details need to be provided.

Line 156, define EAOD.

Lines 169-171, "The AOD data from Terra-MODIS were validated by matching the CARSNET AODs within 30 minutes of the MODIS overpass within the 3×3 pixels surrounding the CARSNET site." Are both CARSNET and MODIS data averaged for the process? Need some details.

Lines 180-182, "The AOD at the urban site of Hangzhou was the highest of all the study sites as a result of high local anthropogenic activity in this urban area compared with the other suburban and rural sites." Results do not support this comment, as the three sites, including Handzou, Xiaoshan and Fuyang, have mean AOD of 0.76. Also, as I mentioned before, variations in data are larger than differences in mean values. Some statistical analyses are needed to back up their conclusions.

Lines 189-197, the authors compared AOD values from the 7 sites to other regions in China as reported from other papers. However, the authors should also take the temporal variation into consideration (e.g., mean values change from year to year, right?).

Line 241, what is extinction Angström exponent?

Lines 311-314, "The characteristics of the SSA at these seven sites gradually increased from the east coast (0.91±0.06 at Hangzhou) inland toward the west (0.94±0.03 at ChunAn). These results indicate the emissions caused by human activity affect the absorption of aerosols in urban areas." Again, the authors need to consider other possibilities and worry about data varibility. How about meteorological conditions? What about hygroscopic growth? Again, the authors need to back up their comments with evidence.

Figure 10, what do the colors represent?

[Figure]

---

## Referee Comment (RC3) · Anonymous Referee #3 · 6 Aug 2017

This paper presented the aerosol optical properties as observed over seven CARSNET sites over eastern China. Aerosol loading, together with aerosol SSA, refractive index, and particle size were analyzed in a monthly scale. Authors also evaluated MODIS AOD retrievals, analyzed aerosol type, and calculated radiative forcing using those ground-based measurements. Their study covered many aspect of aerosol properties over the studied domain, providing a comprehensive analysis.

However, the paper lacks focus, by simply redundantly piling data analysis variable by variable. As indicated by the other two reviewers, the paper is poorly structured. Sub-

stantial revisions are needed to improve the organization of data analysis on various parameters and make the paper more concise and focused. Some aerosol variables (i.e, AE and aerosol size, and SSA and refractive index) are highly related and should to be presented interactively. The paper also needs to be re-organzied in three different section, i.e., (1) analysis of aerosol properties; (2) validation of MODIS (this part indeed does not sever the objectives of this paper, should consider to remove); (3) radiative forcing estimate. In addition, a discussion section may be added to discuss the results and how they can be interpreted in perspective of previous studies, as well as the strength and limitation of the present study (see my below comments).

Specific comments:

1.Page 3, L18-19: Many networks are listed here. But is not clear "which" network "includes several automated sites in China". Please revise this in a more accurate way.

2.Page 6, L14: The SSA was retrieved using only -> The retrieved SSA was used only when Also, please provide reference for selecting AOD440 of 0.4 as the threshold.

3.Page 6, L15-19: These two sentence are very confusing and need rewords like: Real and imaginary parts of refractive index at 4 wavelengths (440, 675, 870, and 1020 nm) were retrieved from sky radiance and were confined in the range of . . ., respectively. Also retrieved were aerosol volumes of xx size bins within the 0.05 - 15 um radius range.

4.Page 7, L10: Justification is needed for using 3x3 pixel averaging.

5.In the first paragraph of the result section (and in many places in the following sections), authors included a lot of comparisons of AOD between YRD area to other regions of China. Such comparisons may be interested but would distract readers. The result section should focus on presenting the findings, and such extensive discussion should be placed in a discussion section.

6.Figure 2: The font size of the figure labels and legends should be increased.

7.Page 14, L2-3: Could authors explain in more detail on the reasoning of "method for estimating the surface reflectance was suitable for this region"? What about surface reflectance estimation in Hangzhou site?

8.Section 3.2: Please note that AERONET inversion algorithm assume refractive index does not vary with aerosol particle size [Dubovik et al. 2000]. In other words, refractive indices are same for retrieved fine mode and coarse mode. As a results, mode-specific SSA were not recommend to use due to large uncertainty. Furthermore, the coast-mode aerosol loading is too small to offer sufficient information on absorption of the coarse-mode particles. Therefore, only total SSA should be used for the analysis to avoid misleading (even AERONET total SSA has error of 0.03). In addition, SSA on a longer wavelength could be included to examine the absorbing aerosol type, as different absorbing particles (dust and smoke) appear different spectral contrast of SSA.

9. Figure 5: As in comments, please consider removing SSA of each mode and retaining the total SSA. Font size of labels should be increased.

10.Section 3.3: Again, AERONET refractive index retrievals are not size dependent.

11. Section 3.6: Is Figure 13 based on monthly averaged variables? If true, monthly data may cause problem in classifying aerosol type. Those would simply represent the mean values of those parameters rather the mean states of aerosol types. The information of actual aerosol types may fade out during averaging process.

12.Table 2: Pie chart may be a better option to present the aerosol type category.

Corrections:

Page 3, L25: aerosols present -> aerosols

Page 6, L 21: Do you intend to say "from 0.2 to 4.0 um"?

Page 6, L26-27: I believe the reference for AERONET inversion algorithm (sphericity

fraction) should be Dubovik et al. [2006], please verify.

Page 7, L1: "were used to provide an evaluation of MODIS AOD retrieval with" -> were evaluated against

Page 7, L16: the variation range of AOD at 440 nm is -> the annual mean of AOD at 440 nm ranges

Page 7, L24: The word "trend" often refers to change with time. "pattern" could be better option.

Page 8, L3: a little bit higher -> slightly higher

Page 9, L16: Please use "fine-mode fraction of AOD"; consistent higher -> consistently higher

Page 9, L18: variation for -> variation of

––––––––––––––––––––––––––––

---

## Author Comment (AC1) · 30 Aug 2017

General comments:

Che et al. reported sunphotometer measurements of aerosol microphysical and optical properties and modeling of aerosol surface and top-of-the-atmosphere (TOA) radiative forcing (RF) at several ground stations of eastern China. This manuscript is poorly written and needs a major overhaul. In many places, discussions on the results are groundless without providing any backing evidence or appropriate references to the literature. Most of the discussions are too superficial to provide any value or interest to the aerosol-climate-change or air-pollution communities. The logical flow is confusing and unnatural, with potential for improvement in the organization of sections. Some sections need to be rewritten and reorganized to remove redundancy, while additional information must be provided in a few places.

Response: Thanks for the reviewer's important and constructive comments and suggestions, the major overhaul has been done in some sections which need to be rewritten and reorganized according to the helpful suggestions. The discussions and the logical flow improvement in the organization in the revised manuscript have been rewritten and reorganized and some redundancy section has been removed and concise.

Reviewer 1

1. Title of the manuscript needs to be changed. Title implies that the study reports aerosol instantaneous radiative forcing, while monthly and annual RF are shown instead. Remove "instantaneous". "Temporospatial" is awkward. Consider using "spatiotemporal" instead. Even worse, the author mentioned nothing whatsoever in the manuscript about the purpose or the advantage of such high temporospatial data (more comments on this later on). "Seven sites" should not be treated as "high spatial resolution". Consider writing the full name of CARSNET in the title.

Response: Thanks for the suggestions. The title of the manuscript has been changed as "Aerosol optical properties and radiative forcing based on synchronous measurements of China Aerosol Remote Sensing Network (CARSNET) over eastern China" according to the reviewer's important comments.

2. Abstract needs to be rewritten. The common flow of logic is to discuss the aerosol microphysical properties (size, refractive index) first, then optical properties (Kext, FMF, SSA, etc), and finally RF. This comment also applies to the main body text. I understand that the author is trying to follow the way how these variables are derived from the AERONET inversion algorithm. I do not think this is necessary. The result subsections need to be reorganized, combined, and shortened to make the paper more concise and less confusing.

Response: According to the helpful suggestions of reviewer, the section of Abstract has been rewritten in the revised manuscript. In the Abstract, the aerosol microphysical properties of volume size distribution has been described firstly, then the aerosol optical properties including AOD, SSA, refractive index, AAOD and finally RF has been written. This order has also applies to the main body text to make the paper more concise and less confusing.

3. L38: what wavelength is the AOD value? If it is 440 nm, why not use 500 nm or 550nm to make the result more useful for the community and more easily comparable to past studies? Optical properties are wavelength dependent. Provide this information when introducing any variables at the first time.

Response: The wavelength of AOD is at 440nm in the sentences of "*The aerosol optical depth (AOD440nm) varied from 0.68 to 0.76,*" in line 45-46 in Abstract. Consider of the suggestions of reviewers, we added the section of monthly and diurnal AOD at 500 nm in section 3.2 from line 364 to 384 as well as figure 4 and 5 as follows: "*The monthly and diurnal cycle of AOD at 500nm has also been discussed in Fig.4 and Fig.5. The annual values of AOD500nm over the seven urban, suburban and rural sites in this study varied from 0.53 (ChunAn) to 0.68 (Hangzhou).The results show that two peaks of AOD at 500nm occurs in June and September with values of 1.25±0.59 and 1.00±0.42 in the urban site of Hangzhou, respectively which has the similar pattern as the other sites. The increase of AOD at 500nm in June is not corresponding to the same increase pattern of EAE (about 1.5) which indicates the aerosols types may be relatively constant in this region. The Fig.5 depicts the diurnal patterns of AOD at 500nm in this megacity area of eastern China. We can see that there are two types of diurnal patterns in this region. The daily AOD has been found increased in early morning (08:00 hr to 09:00 hr) and afternoon (12:00 hr to 14:00 hr) about the value of 0.60 to 0.70in Hangzhou,*

*Xiaoshan, Fuyang and Linan, while the decreasing of daily AOD has been observed from 0.70 to 0.50 during the daytime (from 07:00 hr to 16:00 hr) in Tonglu, Jiande and ChunAn. The high AOD during 07:00～09:00 in the urban area may be due to the anthropogenic activities and aerosol emissions from the morning rush hour. The decreased AOD with the value of 0.37±0.36 occurred in the suburban cities of Tonglu, Jiande and ChunAn may be due to the meteorological conditions more than anthropogenic effects. During the day, the aerosols in the near-surface may spread into vertical as a result of turbulence due to the more and more unstable atmosphere by the continuous strengthening of solar radiation.".*

[Figure]

*Fig.4.Variation in the AOD at 500nm & EAE at 440–870 nm over (a) Hangzhou, (b) Xiaoshan,*

*(c) Fuyang, (d) LinAn, (e) Tonglu, (f) Jiande and (g) ChunAn. The boxes represent the 25th to 75th percentile distribution, while the dots and solid lines within each box represent the mean and median, respectively.*

[Figure]

*Fig.5. Variation of diurnal cycle in the AOD at 500 nm over (a) Hangzhou, (b) Xiaoshan, (c) Fuyang, (d) LinAn, (e) Tonglu, (f) Jiande and (g) ChunAn. The boxes represent the 25th to 75th percentile distribution, while the dots and solid lines within each box represent the mean and median, respectively.*

4. L40: use the term "fine mode fraction" if that is what you mean here.

Response: The words "The ratio of the AOD of fine-mode particles" has been changed as "The fine mode fraction" in Abstract in line 47 and this change applied to the rest of the manuscript.

5. L43: The Pearson correlation coefficient is noted as R, not Rˆ2 (which is called coefficient of determination).

Response: Thanks for the suggestions of the reviewer. The sentences in line 43 of "The correlation coefficients (R2) between the MODIS C6 AOD data and the values measured on the ground were ~0.73–0.89." has been modified as "The MODIS/Terra C6 retrieval AOD values was generally more stable in the YRD region compared with the MODIS/Aqua product with the two Deep Blue (10km) and Dark Target (3km and 10km) methods against ground-based observations." in the revised manuscript in Abstract on line 50-55. In the section 3.2, the corresponding correlation coefficients (R) values has been revised as "*The correlation coefficients (R) between the MODIS/Aqua and MODIS/Terra between by the Dark Target methods at 3km and sun photometer AOD (550 nm) values were about 0.84 to 0.92 and 0.85 to 0.94 in the YRD region, respectively.*" in line 411-415, "*The correlation coefficients (R) of the MODIS/Aqua and MODIS/Terra between sun photometer AOD (550 nm) values by the Deep Blue and Dark Target methods at 10km were about 0.81 to 0.90, 0.85 to 0.90, 0.69 to 0.91 and 0.85 to 0.93 in the YRD region, respectively.*" in line 432-line 435, "*In particular, the biases of the correlation coefficients (R) occurred in LinAn and Jiande has decreased from 0.94 and 0.90 to 0.87 and 0.88.*" in line 437-439.

6. L49-50, use volume fraction instead.

Response: According the reviewer's suggestions, the sentences in line 49-50 "The fine-mode radii in the Yangtze River Delta region were ~0.2–0.3 µm with a volume of 0.10–0.12 µm3 and the coarse-mode radii were ~2.0 µm with a volume close to 0.07 µm3." has been modified as "The fine-mode radii in the Yangtze River Delta region were ~0.2–0.3 µm with a volume fraction of 0.10–0.12 µm3 and the coarse-mode radii were ~2.0 µm with a volume fraction close to 0.07 µm3." in line 39-41. This changes has also been revised in section 3.1 line 229-231 as "*The fine-mode radii were ~0.2–0.3 µm in the YRD with a volume fraction of 0.10–0.12 µm3 and the coarse-mode radii were ~2.0 µm with a volume fraction close to 0.07 µm3.*".

7. L52-54: reads like a method description (remove it), and what are the key findings?

Response: Thanks for the important suggestions. The sentences "*The absorption Angström exponent and the extinction Angström exponent were used to classify the different types of aerosol and the components of mixtures.*" in line 52-54 has been modified in the revised manuscript as "*The absorption Angström exponent and the extinction Angström exponent shows that the "mostly dust" category was very low in the suburban and rural sites (<0.01%) and also less in the urban site (~0.24%)*" to highlight the key findings in line 63-66.

8. L54-56: this statement is too general to be included in an abstract. Abstract is to present the most original and important findings of the study. Do not simply report the results as general statements. Writing an interesting abstract is a critical step to pass a peer review.

Response: Thanks for the helpful suggestions. This statement in the abstract has been revised to present the most original and important findings of the study. The sentences in L54-56 of "*The aerosols caused negative radiative forcing both at the Earth's surface and at the top of the atmosphere all year round in the Yangtze River Delta region of eastern China.*" has been changed as "*The aerosols caused negative radiative forcing both at the Earth's surface and at the top of the atmosphere all year round in the Yangtze River Delta region with the lower surface albedo in an unique geographical climate conditions of better vegetation in the YRD region than in north/northeast China.*" to show the special surface features in this distinct in line 66-70.

Section 1

9. Introduction needs to polish. Always write out the acronyms at their first appearance. This comment applies to the rest of the manuscript.

Response: According to the reviewers suggestion, the introduction part has been revised. The acronyms such as "AERONET"," SKYNET", "EARLINET" and "GAW-PFR Network" has been written out the full name at their first appearance as" AERONET (Aerosol Robotic Network)", "SKYNET (SKYrad Network)", "EARLINET (European aerosol Lidar network)" and "GAW-PFR Network (Global Atmosphere Watch Programmer-Precision Filter Radiometers)" in Introduction and throughout the manuscript.

10. L92: it looks like the author wants to emphasize the advantage of using multi-site, multiyear, as well as three-minute-interval (L35) or high-frequency (L97) data; however, not in a single place of the manuscript the author discusses the advantage of such data. After all, all sites seemed to have similar aerosol characteristics. And all results are presented as monthly or annual means. Consider adding analysis of diurnal cycle and interannual variations, or exploring other unique aspects of the data.

Response: Thanks for the constructive comments and suggestions. We added the analysis of monthly and diurnal cycle of AOD at 500nm in section 3.2 to display a unique aspect of the data in this region. The result is as follows "*The monthly and diurnal cycle of AOD at 500nm has also been discussed in Fig.4 and Fig.5. The annual values of $AOD_{500nm}$ over the seven urban, suburban and rural sites in this study varied from 0.53 (ChunAn) to 0.68 (Hangzhou).The results show that two peaks of AOD at 500nm occurs in June and September with values of 1.25±0.59 and 1.00±0.42 in the urban site of Hangzhou, respectively which has the similar pattern as the other sites. The increase of AOD at 500nm in June is not corresponding to the same increase pattern of EAE (about 1.5) which indicates the aerosols types may be relatively constant in this region. The Fig.5 depicts the diurnal patterns of AOD at 500nm in this megacity area of eastern China. We can see that there are two types of diurnal patterns in this region. The daily AOD has been found increased in early morning (08:00 hr to 09:00 hr) and afternoon (12:00 hr to 14:00 hr) about the value of 0.60 to 0.70in Hangzhou, Xiaoshan, Fuyang and Linan, while the decreasing of daily AOD has been observed from 0.70 to 0.50 during the daytime (from 07:00 hr to 16:00 hr) in Tonglu, Jiande and ChunAn. The high AOD during 07:00～09:00 in the urban area may be due to the anthropogenic activities and aerosol emissions from the morning rush hour. The decreased AOD with the value of 0.37±0.36 occurred in the suburban cities of Tonglu, Jiande and ChunAn may be due to the meteorological conditions more than anthropogenic effects. During the day, the aerosols in the near-surface may spread into vertical as a result of turbulence due to the more and more unstable atmosphere by the continuous strengthening of solar radiation.*".*

Section 2.

11. L130: Given the high AOD (comparable to urban site), ChunAn is nothing like a background site. (L177)

Response: Thanks for the suggestions. Considering to the AOD level of ChunAn comparable to urban site, we changed the description of ChunAn as "rural site" in Table 1 and "*clean site*" as "*The rural site of ChunAn can be regarded as a representative clean site less affected by local and regional pollution.*" In section 2 line 145 and "*The site at LinAn is regarded as the clean suburban site in eastern China with an average AOD about 0.73±0.44,*" in section 4 on line 813-814 and "*e.g. particles from urban/industrial activities) over the clean rural sites than over urban or suburban sites.*" In Section 3.3 on lines 516-518 instead of the background site in this manuscript.

12. L146-157: describe the assumptions of the inversion algorithms, some of which may not apply for eastern China. Also describe the accuracies of the inversion variables (which are essentially a remote sensing product).

Response: Thanks for the important suggestions. The assumption of the inversion algorithm is "*The inversion algorithm is under an assumption of homogeneous nonsphericity aerosol particles distribution according to Dubovik (2006).*" and "*Real and imaginary parts of refractive index at 4 wavelengths (440, 675, 870, and 1020 nm) were retrieved from sky radiance and were confined in the range of 1.33–1.60 and 0.0005–0.50, respectively (Dubovik and King, 2000; Che et al., 2015b). Also retrieved were aerosol volumes of 22 size bins within the 0.05 - 15 um radius range.*" as mentioned in senction2 on line 171-173 and line 177-180. The accuracies of the inversion variables has been described in section 2 as "*The accuracies of SSA is~0.03, and the errors are about 30%–50%/0.04 for the imaginary/real part of the complex refractive index under the conditions of AOD440nm larger than 0.4 with the solar zenith anglemore than 50°.*" in line 173-175. The assumption of the inversion algorithms has also been used in Taihu, Nanjing, etc in eastern China in AERONET sites (Holben et al., 1998).

13. L159-161: Need more descriptions on the technical details of radiative transfer calculations. For example, how is the atmospheric profile treated and where are the profile data coming from? How is cloud scattering treated? How are clouds and aerosols vertically distributed? How is the surface and TOA radiative forcing defined?

Response: Thanks for the reviewer's constructive comments. More descriptions of atmospheric profile, cloud scattering treated, aerosols vertically distributed and the surface and TOA radiative forcing has been added in section 2 as follows "*The ARF (aerosol radiative forcing) data were calculated by the radiative transfer module used by the AERONET inversion (García et al., 2012) under the assumption of cloud-free consideration. In this code, the aerosol vertical properties have been considered into a homogeneous atmosphere layers because of the weak dependent of ground radiances on the whole atmospheric column with minor uncertainties (Dubovik et al., 2000).The broadband fluxes from 0.20 to 4.0µm were calculated according to the radiative transfer model GAME (Global Atmospheric ModEl) (Dubuisson et al., 1996, 2006; Roger et al., 2006). The size distribution, complex refractive index, and spherical particles fraction has been retrieved from the almucantar plane in the measurements. The SA (surface albedo) is obtained from the MODIS albedo product (MCD43C3) with the interpolation value of 440, 670, 870, and 1020 nm. The water vapor at 940 nm has been retrieved by the sun photometer. The ozone content was obtained from NASA Total Ozone Mapping Spectrometer measurements from 1978 to 2004. And other atmospheric gaseous data came from the US standard 1976 atmosphere model. In this study, the two parameters of ARF at the surface (ARF-BOA) and at the top of the atmosphere (ARF-TOA) have been calculated to describe the aerosol direct radiation effect to account for the changes of the solar radiation by calculating the difference energy between the aerosols presentation and absentation.*" In Line 188-211.

14. L162-172: Why not use Aqua? I understand that MODIS/Aqua is more stable and has better calibration than Terra (corrects me if I am wrong). It seems that the author calculated AOD at 550 nm in order to validate MODIS. If this is the case, why not show AOD at this wavelength for the entire manuscript (e.g. Table 1)?

Response: The reviewer's suggestions is very important. The Aqua product has been added to validate the MODIS AOD at 550nm according to the suggestions in section 3.2. The MODIS C6 aerosol optical thickness products refined by Levy et al. (2013) were evaluated against our ground-based observations by the Deep Blue and Dark Target methods at 3km and 10km separately in section 3.2 as follows in line 400-476:

[revised manuscript text omitted]

Moreover, the monthly and diurnal AOD at 500nm has been analyzed for the entire manuscript in Table 1 and section 3.2. In this study, all sites have AOD measurements at 340, 380, 440, 500, 670, 870, 1020, 1640nm, respectively. The direct measured AOD $_{500nm}$ is more accurate than $AOD_{550nm}$ transferred by the interpolated and extrapolated value from by AOD values at 440 and 675nm. To avoid uncertainty, the 550nm AOD were calculated for MODIS validation. The direct measured $AOD_{500nm}$ were used to analyze the aerosol optical characteristics and also is to consistent and comparable to previous studies We added the analysis of monthly and diurnal cycle of AOD at 500nm in section 3.2 to display a unique aspect of the data in this region.

15. L177: from Table 1, Xiaoshan and Fuyang have far smaller observation samples than other sites. Why? Does this affect the comparability between the sites? ChunAn AOD is only 10% less than other sites. I would not say it is a representative background site.

Response: Thanks for the suggestions. The problem of smaller observation samples in Xiaoshan and Fuyang has been explained in section 2 as follows in 246-251: "*Smaller observation sample has been found in Xiaoshan and Fuyang with 180 and 217 available observation days, respectively. The number of 180 observation days in Xiaoshan is less than half of the year may have less representative and need further observation accumulation, while the observation days of 217 in Fuyang was more than half of the year may not affect the comparability between the other sites.*". According to the AOD level of ChunAn comparable to urban site, we changed the description of ChunAn as "rural site" in Table 1 and "*clean site*" as "*The rural site of ChunAn can be regarded as a representative clean site less affected by local and regional pollution.*" In section 2 line 145 and "*The site at LinAn is regarded as the clean suburban site in eastern China with an average AOD about 0.73±0.44,*" in section 4 on line 813-814 and "*e.g. particles from urban/industrial activities) over the clean rural sites than over urban or suburban sites.*" In Section 3.3 on lines 516-518 instead of the background site in this manuscript. And this has been response in question 11 in the above.

16. L180-188: this part has serious redundancy. Rewrite it to be more concise.

Response: According to the reviewer's helpful suggestion, this part in line 180-188 has been rewritten concisely in section 3.2 line 251-265 as follows as "*The annual values of the AOD440nm at Hangzhou, Xiaoshan, Fuyang, LinAn, Tonglu, Jiande and ChunAn were about 0.76±0.42, 0.76±0.43, 0.76±0.45, 0.73±0.44, 0.71±0.41, 0.73±0.40 and 0.68±0.38, respectively, which suggests that column aerosol loading is at a high level at all seven urban, suburban and rural sites in the YRD on the regional rather than the local scale. The AOD440nm decreased from the eastern coast to the inland areas towards the west (from*

*~0.76±0.42 at Hangzhou to ~0.68±0.38 at ChunAn) due to the high aerosol loading from economic development and anthropogenic influences. The annual AOD440nm shows that the aerosol loading has similar level in Hangzhou, Xiaoshan and Fuyang, and with the 4%-10% decrease in LinAn, Tonglu, Jiande and ChunAn, respectively. The AOD440nm at the urban site of Hangzhou was the highest as a result of the more industrial activity and high resident density in the eastern part metropolis region resulting in higher aerosol emissions compared with the other suburban and rural sites.*".

17. L188-206: It is dangerous to use AOD solely as a metric for pollution (PM) or emission level. Keep in mind that AOD is an optical quantity (at a specific wavelength) and measures the light extinction of a vertical atmospheric column from the ground to TOA. The AOD difference between sites may not reflect the ambient pollution severity if some sites are affected by transported events. The AOD difference between cities may not reflect the emission sources if the meteorological conditions are so different to affect the dispersion and lifetime...The author needs a more solid analysis here.

Response: The reviewer's suggestions are important and constructive. In the revised paper, we used aerosol extinction ability instead of describing a metric for pollution (PM) or emission level based on the value of AOD. These modification has been modified in DISCUSSION as "*The AOD in Hangzhou in urban eastern China was similar to that in Shenyang (0.75) in urban northeast China (Zhao et al., 2013), and in Beijing (0.76) and Tianjin (0.74) in urban north China (Che et al., 2015b), indicating that the aerosol extinction is both common and at a similar level throughout most urban areas of China. The AOD values at the urban and suburban sites of Hangzhou were slightly higher than at Pudong (0.70) and Hefei (0.69), other urban areas in eastern China, suggesting that higher aerosol extinction ability were observed here (He et al., 2012; Liu et al., 2017). However, the AOD at all seven sites was lower than that obtained at Wuhan (1.05), Nanjing (0.88), Dongtan (0.85), Taihu(0.77) and Xuzhou (0.92) in previous studies in eastern China (Wang et al., 2015; Li et al., 2015; Pan et al., 2010; Xia et al., 2007; Wu et al., 2016). This indicates that the aerosol loading caused by anthropogenic activities is very high in both urban and suburban areas in eastern China. The site at LinAn is regarded as the clean suburban site in eastern China with an average AOD about 0.73±0.44,*

*which is higher than that at the other regional background stations of China, such as Longfengshan (0.35; northeastern China), Mt Waliguan (0.14, inland Asia), Xinglong (0.28, northern China), Akedala (0.20, northwestern China) and Shangri-La (0.11, southwestern China) (Wang et al., 2010; Che et al., 2011; Zhu et al., 2014; Che et al., 2015b). The aerosol loading in eastern China (especially in the YRD region) is at least twice as high as in other regions of China which indicate the strong aerosol extinction.*".

18. L225-230: Another example of redundancy. Rewrite to be more concise.

Response: Thanks for the helpful suggestion. A concise description has been re-written in the manuscript in section 3.2 line 314-342 as follows "*The annual fine-mode AOD values at Hangzhou, Xiaoshan, Fuyang, LinAn, Tonglu, Jiande and ChunAn were about 0.68±0.42, 0.69±0.41, 0.69±0.44, 0.66±0.43, 0.64±0.41, 0.66±0.40 and 0.61±0.38, respectively (Fig.3). The seasonal variation in the AOD was similar to the total AOD at these urban, suburban and rural sites. The fine-mode fraction of AOD consistently exceeded 0.90 which indicates a major contribution of fine mode fraction to the total AOD in the YRD. Moreover, the figure 3 shows that the EAE at Hangzhou, Xiaoshan, Fuyang, LinAn, Tonglu, Jiande and ChunAn was about 1.29±0.26, 1.37±0.24, 1.32±0.24, 1.29±0.27, 1.30±0.26, 1.32±0.28 and 1.22±0.25, respectively. Values of EAE >1.20 were found in all months throughout the year, indicating that small particle size distributions were favored in the YRD region. The annual coarse-mode AOD values at Hangzhou, Xiaoshan, Fuyang, LinAn, Tonglu, Jiande and ChunAn were between about 0.06 and 0.08 with the coarse mode fraction of AOD about 0.10 which indicates the 10% contribution of coarse mode fraction to the AOD in the YRD. The less coarse mode fraction indicated that there is no obvious effect of the coarse particles in the YRD region than that contributed to the higher aerosol loading in other north/northeast China (Zhang et al., 2012). Some dusts cases can be observed in YRD region that transported from north/northwest China during 2012-2015 reflect the effect of mineral dust aerosols (Gong et al., 2003). The fugitive dust from road traffic and construction activity is another more persistent and significant source for China's cities as well as these eastern megacities.*"

19. L231-235: Another example of unconvincing analysis. Is the transported dust event verified?

Please describe the specific events (2012-2015) rather than citing a climatological study. I suspect that fugitive dust from road traffic or construction activity is a more persistent and significant source for China's cities.

Response: Thanks for the important suggestions. The two reasons of fugitive dust from road traffic or construction activity has been added in line 340-342. According to the reviewer's suggestion, we rechecked the PM10 mass concentration observation data from 2012-2015 and describe a specific events on 5 March 2013. Moreover, the Lidar and CALIPSO data have also been investigated. As can be seen in the figure below, the PM10 mass concentration is higher in March to May than the value in February and June. This pattern clearly indicates the influence of dust in spring of Hangzhou. The Eastern China has been affected by a wide range of dust event in North China. These dust events across the northwest to the northeast area of China, then continue keep going south and east China. The higher PM10 mass concentration is obviously affected by the dust come from the northwest.

[Figure]

For example, there is a dust transportation event occurred in Hangzhou on 05-09 March, 2013. A series of transported aerosol masses over Hangzhou are monitored by MPL shown through time-height cross section of extinction coefficient at 527nm. Referring to CALIPSO L2 retrieval results of vertical feature mask and aerosol subtypes illustrated in the Figure bellowing, the northwestern upwind areas of Hangzhou exists an aerosol layer mixed of "dust" and "polluted dust" about 3km thick from the surface in 5 March 2013, since when a thin external layer is detected concurrently. Therefore, the higher AOD in spring described in this study is significantly affected by dust process.

[Figure]

[Figure]

20. L241-250: The findings based on EAE are very similar to those from FMF (L222-235). Consider merging them to be more concise.

Response: According to the reviewer's helpful suggestion, the findings based on EAE in line 241-250 and those from FMF in Line 222-235 has been merged and modified to be more concise. The revised content is as following "*The monthly and diurnal cycle of AOD at 500nm has also been discussed in Fig.4 and Fig.5. The annual values of AOD 500nm over the seven urban, suburban and rural sites in this study varied from 0.53 (ChunAn) to 0.68 (Hangzhou).The results show that two peaks of AOD at 500nm occurs in June and September with values of 1.25±0.59 and 1.00±0.42 in the urban site of Hangzhou, respectively which has the similar pattern as the other sites. The increase of AOD at 500nm in June is not corresponding to the same increase pattern of EAE (about 1.5) which indicates the aerosols types may be relatively constant in this region. The Fig.5 depicts the diurnal patterns of AOD at*

*500nm in this megacity area of eastern China. We can see that there are two types of diurnal patterns in this region. The daily AOD has been found increased in early morning (08:00 hr to 09:00 hr) and afternoon (12:00 hr to 14:00 hr) about the value of 0.60 to 0.70in Hangzhou, Xiaoshan, Fuyang and Linan, while the decreasing of daily AOD has been observed from 0.70 to 0.50 during the daytime (from 07:00 hr to 16:00 hr) in Tonglu, Jiande and ChunAn. The high AOD during 07:00~09:00 in the urban area may be due to the anthropogenic activities and aerosol emissions from the morning rush hour. The decreased AOD with the value of 0.37±0.36 occurred in the suburban cities of Tonglu, Jiande and ChunAn may be due to the meteorological conditions more than anthropogenic effects. During the day, the aerosols in the near-surface may spread into vertical as a result of turbulence due to the more and more unstable atmosphere by the continuous strengthening of solar radiation.*" in line 364-384.

21. L322-334: SSA depends on two factors - particle size and composition. It is expected that coarse mode SSA is less than fine mode SSA. I do not see the need for two SSAs for fine and coarse modes, separately. This paragraph presents no new findings and can be removed.

Response: Thanks for the important suggestions. Both reviewers mentioned this question. According to the two reviewer's suggestion, the section of coarse mode SSA and fine mode SSA has been removed in section 3.3 according to the suggestions of reviewer. Then we added the discussion of SSA depends on different wavelength in section 3.2 line 539-551 as "The wavelength dependence of SSA present specific absorption/scattering properties of different type aerosol (Sokolik and Toon, 1999; Eck et al., 2010). The SSA of dust in spring shown a dependence on the spectrum from 440nm to 1020nm in general (Cheng et al., 2006; Dubovik et al., 2002). Especially in March, the SSA at 440nm in Hangzhou, LinAn, Jiande and ChunAn was obviously lower at short wavelength than that in the longer wavelength. This result has shown a strong absorption of dust in the short wavelength in the YRD region over eastern China. It's worth noting that there is an obvious and strong decreasing of SSA in the longer wavelength of aerosol from biomass burning or industrial emissions in August. The wavelength dependence of SSA in YRD could be used to simply describe the aerosol types including dust or the biomass burning smoke.".

[Figure]

*Fig.10. Variation in the SSA at 440nm, 670nm 870nm and 1020nmover (a) Hangzhou, (b) Xiaoshan, (c) Fuyang, (d) LinAn, (e) Tonglu, (f) Jiande and (g) ChunAn. The boxes represent the 25th to 75th percentile distribution, while the dots and solid lines within each box represent the mean and median, respectively.*

22. L349-351: The real part of refractive index is related to scattering, while the imaginary part to absorption. Does the author mean to say spring dust absorption increases the imaginary part (not real part)?

Response: Yes. The sentences of "*High imaginary parts of the refractive index occurred in August at all urban, suburban and rural sites in the YRD, which may be due to the higher emission of absorptive particles by the post-harvest burning of crop residues with more spectral dependence. The burning of crop residues may cause a large deterioration in the regional air quality in the YRD region. A higher level of spring dust aerosols with absorption could contribute to a higher value of the imaginary part of the refractive index.*" has been modified in section 3.3 line 597-602.

23. L377-378: AAOD values have very large uncertainties (standard deviation is comparable to mean). Why? Given these uncertain estimates, it would be difficult to make a robust comparison between sites.

Response: Thanks for the important suggestions. AAOD values have very large uncertainties has been explained in line 630-634 as follows "*The AAOD values may have very large an uncertainty because of the dataset is including all the values in one month. Nevertheless, there is also some varies in AAOD according to the changes of the SSA in section 3.2. These differences in the AAOD were mostly dependent on the type of aerosol and the ratio of absorbing and non-absorbing components in the aerosols.*"

24. L392, this paragraph seems redundant as the later part from L416 discusses EAE and AAE links to aerosol types.

Response: According to the reviewer's suggestion, the paragraph in line 392 and the later part from L416 about the EAE and AAE links to aerosol types has been re-discussed in the revised manuscript as follows "*As fig.12 shown, the AAE was<1.00 in June and August at all urban, suburban and rural sites of the YRD, which suggested the presence aerosols coated with absorbing or non-absorbing material in summer season. This process is favored by high temperatures and high humidity under conditions of strong solar radiation (Shen et al., 2015, Zhang et al., 2015). The particles coagulate and grow rapidly in the presence of sufficient water vapor (Li et al., 2016). The AAE became increasingly close to, or larger than, 1.00 at all*

*seven sites from September, which is consistent with decreasing amounts of precipitation. This increase in the AAE was related to the emission of black carbon from biomass burning (Soni et al., 2010; Russell et al., 2010). According to the corresponding annual mean values for the AAE at Hangzhou, Xiaoshan, Fuyang, LinAn, Tonglu, Jiande and ChunAn (1.13±0.46, 0.88±0.42, 0.85±0.43, 0.98±0.35, 1.11±0.49, 1.16±0.44 and 0.93±0.31) in Fig. 12, the seven sites has been attributed to three categories with AAE levels. The mean values of the AAE at Xiaoshan and Fuyang were <1.00, suggesting the presence of absorbing or non-absorbing materials coating black carbon at these suburban and rural sites (Bergstrom et al., 2007; Lack and Cappa et al., 2010; Gyawali et al., 2009). The AAE values were close to 1.00 at LinAn and ChunAn, indicating that the absorptive aerosols were dominated by particles of black carbon (Zhang et al., 2012; Li et al., 2016). By contrast, the AAE values at Hangzhou, Tonglu and Jiande were >1.00, indicating the presence of absorptive aerosols from the burning of biomass. This difference in the AAE distribution indicates the absorbing aerosols have different characteristics resulting from the different emission sources at urban, suburban and rural sites in the YRD.*" in line 641-661.

25. L430-432: explain why the YRD region is "completely different from north/northeast China"

Response: The detailed explanation has been added according to the reviewer's suggestion. The north/northeast China region is with lower precipitation and less vegetation coverage. And large dust particles from regional dust events transportation or local fugitive dust emission could have substantial contribution of aerosol loading in this region. While in the YRD region, there are large precipitation and dense vegetation coverage, resulting in less dust particle contribution to aerosol loading in YRD. The sentence has been changed as "*This indicates the YRD region is completely different from other north/northeast region in China where the dust particles could contribute to the aerosol loading substantially.*" In line 724-727.

26. Figure 10: really bad visualization. Consider other options, like histograms.

Response: The Figure 10 has been redrawn as Figure 14 in the manuscript.

27. L483: do you intend to say "large surface reflectance (or albedo)"

Response: Yes. The word has been changed as "large surface reflectance" in line 790.

28. Section 4 should be shortened. There is no need to repeat all results. There should be one or two paragraphs discussing the implications of the most important findings.

Response: Thank for the constructive suggestions. Section 4 Summary and discussion has been rewritten according to two reviewers' suggestions. The discussions part has been added and the summary part has been shortened and concise.

**Aerosol optical properties and  radiative forcing based on synchronous measurements of China Aerosol Remote Sensing Network (CARSNET) over eastern China**

Huizheng Che[1*], Bing Qi[2], Hujia Zhao[1], Xiangao Xia[3,4], Philippe Goloub[5], Oleg Dubovik[5],

Victor Estelles[6] , Emilio Cuevas-Agulló[7], Luc Blarel[3], Yunfei Wu[8], Jun Zhu[9], Rongguang Du[2],

Yaqiang WANG[1], Hong Wang[1], Ke Gui[1], Jie Yu[1], Yu Zheng[9], Tianze Sun[1], Quanliang Chen[10],

Guangyu Shi[11], Xiaoye Zhang[1*]

State Key Laboratory of Severe Weather (LASW) and Institute of Atmospheric Composition, Chinese Academy of Meteorological Sciences, CMA, Beijing, 100081, China

Hangzhou Meteorological Bureau, Hangzhou, 310051, China

Laboratory for Middle Atmosphere and Global Environment Observation (LAGEO), Institute of Atmospheric Physics, Chinese Academy of Sciences, Beijing, 100029, China

School of Geoscience University of Chinese Academy of Science, Beijing, 100049, China

Laboratoire d'Optique Amosphérique, Université des Sciences et Technologies de Lille, 59655, Villeneuve d'Ascq, France

Dept. Fisica de la Terra i Termodinamica, Universitat de Valencia, C/ Dr. Moliner 50, 46100 Burjassot, Spain

Centro de Investigación Atmosférica de Izaña, AEMET, 38001 Santa Cruz de Tenerife , Spain

Key Laboratory of Regional Climate-Environment for Temperate East Asia, Institute of Atmospheric Physics, Chinese Academy of Sciences, Beijing 100029, China

Collaborative Innovation Center on Forecast and Evaluation of Meteorological Disasters, Nanjing University of Information Science & Technology, Nanjing 210044, China

Plateau Atmospheric and Environment Key Laboratory of Sichuan Province, College of Atmospheric Sciences, Chengdu University of Information Technology, Chengdu, 610225, China

State Key Laboratory of Numerical Modeling for Atmospheric Sciences and Geophysical Fluid Dynamics (LASG), Institute of Atmospheric Physics, Chinese Academy of Sciences, Beijing, 100029, China

Corresponding author: chehz@camscma.cn & xiaoye@camscma.cn

**Abstract**

Variations in the optical properties of aerosols and their radiative forcing were investigated based on long-term synchronous observations made at three-minute intervals from 2011 to 2015 over seven adjacent CARSNET (China Aerosol Remote Sensing NETwork) urban (Hangzhou), suburban (Xiaoshan, Fuyang, LinAn, Tonglu, Jiande) and rural (ChunAn) stations in the Yangtze River Delta region, eastern China. The fine-mode radii in the Yangtze River Delta region were ~0.2–0.3 µm with a volume fraction of 0.10–0.12 µm$^3$ and the coarse-mode radii were ~2.0 µm with a volume fraction close to 0.07 µm$^3$.~~The radii of fine volume fraction in the Yangtze River Delta region were ~0.2–0.3 µm with a volume of 0.10–0.12 µm$^3$ and the radii of coarse volume fraction were ~2.0 µm with a volume close to 0.07 µm$^3$.ratio of the AODof fine-mode particlesagainstThe Moderate Resolution Imaging Spectroradiometer (MODIS) C6 retrieval AOD was validated by comparisonwith ground-based observations.The correlation coefficients ($R^2$R) between the MODIS C6 AODdata and the values measuredon the ground were ~0.73–0.89.
[revised manuscript text omitted]

| Longitude (°E) | 120.19 | 120.25 | 119.95 | 119.72 | 119.64 | 119.27 | 1 |
| Latitude (°N) | 30.26 | 30.16 | 30.07 | 30.23 | 29.80 | 29.49 | 2 |
| Altitude (m) | 41.9 | 14.0 | 17.0 | 139 | 46.1 | 88.9 | 1 |
| $^aN_{day}$ | 485 | 180 | 217 | 562 | 498 | 480 | 4 |
| $^bN_{inst.}$ | 2052 | 752 | 906 | 2410 | 2255 | 1952 | 1 |
| $^cAOD_{500nm}$ | 0.68±0.46 | 0.67±0.43 | 0.66±0.43 | 0.60±0.42 | 0.60±0.41 | 0.63±0.38 | 0 |
| $^eAOD$$^dAOD_{440nm}$ | 0.76±0.42 | 0.76±0.43 | 0.76±0.45 | 0.73±0.44 | 0.71±0.41 | 0.73±0.40 | 0 |
| $^eAOD_{fine}$$^dAOD_{fine(440nm)}$ | 0.68±0.42 | 0.69±0.41 | 0.69±0.44 | 0.66±0.43 | 0.64±0.41 | 0.66±0.40 | 0 |
| $^eAOD_{coarse}$$^dAOD_{coarse(440nm)}$ | 0.08±0.06 | 0.07±0.06 | 0.07±0.06 | 0.07±0.07 | 0.07±0.06 | 0.07±0.07 | 0 |
| $^dEAE$$^eEAE$ | 1.29±0.26 | 1.37±0.24 | 1.32±0.24 | 1.29±0.27 | 1.30±0.26 | 1.32±0.28 | 1 |
| $^eSSA$$^dSSA_{440nm}$ | 0.91±0.06 | 0.93±0.04 | 0.94±0.04 | 0.93±0.05 | 0.92±0.04 | 0.92±0.05 | 0 |
| $^eSSA_{fine}$$^{df}SSA_{670nmfine}$ | 0.923±0.065 | 0.915±0.064 | 0.935±0.064 | 0.924±0.054 | 0.934±0.054 | 0.924±0.075 | 0 |
| $^eSSA_{coarse}$$^{dg}SSA_{870nmcoarse}$ | 0.9082±0.079 | 0.9083±0.078 | 0.9184±0.08 | 0.981±0.068 | 0.981±0.068 | 0.9082±0.089 | 0 |
| $^hSSA_{1020nm}$ | 0.89±0.08 | 0.89±0.08 | 0.89±0.09 | 0.90±0.07 | 0.90±0.07 | 0.90±0.09 | 0 |
| $^eReal$$^dReal$ | 1.43±0.07 | 1.41±0.06 | 1.41±0.06 | 1.42±0.06 | 1.43±0.06 | 1.41±0.05 | 1 |
| $^eImaginary$$^dImaginary$ | 0.011±0.010 | 0.008±0.006 | 0.007±0.006 | 0.009±0.007 | 0.009±0.007 | 0.010±0.009 | 0 |
| $^eAAOD$$^dAAOD$ | 0.06±0.05 | 0.05±0.04 | 0.04±0.04 | 0.05±0.04 | 0.05±0.04 | 0.06±0.04 | 0 |
| $^dAAE$$^eAAE$ | 1.13±0.46 | 0.88±0.42 | 0.85±0.43 | 0.98±0.35 | 1.11±0.49 | 1.16±0.44 | 0 |
| $^eRmeas_t$$^dRmeas_t(μm)$ | 0.70±0.34 | 0.65±0.31 | 0.66±0.33 | 0.66±0.33 | 0.65±0.33 | 0.62±0.24 | 0 |
| $^eRmea_{fine}$$^dRmea_{fine}(μm)$ | 0.18±0.05 | 0.18±0.04 | 0.19±0.05 | 0.19±0.05 | 0.19±0.05 | 0.19±0.05 | 0 |
| $^eRmea_{coarse}$$^dRmea_{coarse}(μm)$ | 2.67±0.47 | 2.73±0.42 | 2.75±0.45 | 2.71±0.52 | 2.66±0.48 | 2.63±0.47 | 2 |
| $^eReff$$^dReff(μm)$ | 0.30±0.10 | 0.29±0.09 | 0.30±0.09 | 0.29±0.10 | 0.29±0.10 | 0.29±0.09 | 0 |

| | | | | | | | | |
|---|---|---|---|---|---|---|---|---|
|  dReff_fine(μm) | 0.16±0.04 | 0.16±0.03 | 0.17±0.04 | 0.16±0.04 | 0.16±0.04 | 0.17±0.04 | | |
|  dReff_coarse(μm) | 2.21±0.40 | 2.26±0.35 | 2.30±0.39 | 2.24±0.44 | 2.19±0.41 | 2.16±0.39 | | |
|  dVolume(μm$^3$) | 0.19±0.09 | 0.19±0.09 | 0.19±0.09 | 0.18±0.09 | 0.17±0.09 | 0.18±0.09 | | |
|  dVolume_fine(μm$^3$) | 0.10±0.06 | 0.11±0.06 | 0.11±0.07 | 0.10±0.06 | 0.10±0.06 | 0.10±0.06 | | |
|  dVolume_coarse(μm$^3$) | 0.09±0.06 | 0.08±0.05 | 0.08±0.06 | 0.08±0.05 | 0.08±0.06 | 0.08±0.07 | | |
|  dARF-BOT(W/m$^2$) | −93±44 | −84±41 | −80±40 | −81±39 | −79±39 | −82±40 | | |
|  dARF-TOA(W/m$^2$) | −35±20 | −36±21 | −37±21 | −36±21 | −35±20 | −35±21 | | |

[a] Number of available observation days.

[b] Number of instantaneous observations.

[c] Optical parameters at a wavelength of  500nm.

[d] Optical parameters at a wavelength of 440 nm.

[d]  [e] Angström exponents between 440 and 870 nm.

[f] Optical parameters at a wavelength of 670 nm.

[g] Optical parameters at a wavelength of 870 nm.

[h] Optical parameters at a wavelength of 1020 nm.

Ding et al. (2013a,b) showed that plumes from agricultural burning in June may significantly and seriously affect the radiation balance and air quality of the YRD region. In this study, the monthly averaged AODs at most sites showed two peaks in June and September (Fig.23) with values of ~1.26±0.50 and ~1.03±0.57, respectively. This may be attributed to the accumulation of fine-mode particles via hygroscopic growth in the summer season and the burning of crop residue biomass under a continental high-pressure system with good atmospheric stability and frequent temperature inversions. These conditions lead to the poor diffusion of pollutants (Xia et al., 2007). As Fig.3 shown, the monthly average value of the extinction Angström exponent (EAE, -dln[EAOD(λ)]/dln(λ))  in Hangzhou was higher in

January (~1.40±0.23) and September (~1.43±0.24). This conclusion is also indicated the dominance of small particles from anthropogenic emissions and agricultural activity in autumn and winter (Tan et al., 2009).

The annual fine-mode AOD values at Hangzhou, Xiaoshan, Fuyang, LinAn, Tonglu,

Jiande and ChunAn were about 0.68±0.42, 0.69±0.41, 0.69±0.44, 0.66±0.43, 0.64±0.41,

0.66±0.40 and 0.61±0.38, respectively (Fig.3). The seasonal variation in the AOD was similar to the total AOD at these urban, suburban and rural sites. The fine-mode fraction of

AOD consistently exceeded 0.90 which indicates a major contribution of fine mode fraction to the total AOD in the YRD.

Moreover, the fFigure 3 shows that the annualextinction Angström exponent(EAE)EAE at Hangzhou, Xiaoshan, Fuyang, LinAn, Tonglu, Jiande and ChunAn was about 1.29±0.26, 1.37±0.24, 1.32±0.24, 1.29±0.27, 1.30±0.26, 1.32±0.28 and 1.22±0.25, respectively. Values of EAE >1.20 were found in all months throughout the year, indicating that small particle size distributions were favored in the YRD region. The annual coarse-mode AOD values at Hangzhou, Xiaoshan, Fuyang, LinAn, Tonglu, Jiande and ChunAn were between about 0.06 and 0.08.The with the ratio coarse mode fraction of AODAOD_c/AOD_t was about 0.10, which indicates that about 10% of the 10% contribution of coarse mode fraction to the AOD in the YRD. regionis from coarse particles .The variation in the coarse-mode AOD (Fig. 2) also showed a significant increase in Marchatallseven sites of about 0.14±0.08, 0.08±0.04, 0.09±0.09, 0.13±0.11, 0.13±0.11, 0.14±0.08 and0.11±0.07at Hangzhou, Xiaoshan, Fuyang, LinAn, Tonglu, Jiande and ChunAn, respectively.The monthly average value of the EAEin Hangzhou was higher in January (~1.40±0.23) and September (~1.43±0.24). This indicated the dominance of small particles from anthropogenic emissions and agricultural activity in autumn and winter (Tan et al., 2009).The lower EAE was lower in March (~1.16±0.24) and April (~1.13±0.22)ThoughThe less coarse mode fraction indicated that there is no obvious effect of the coarse particles in the YRD region than that contributed to the higher aerosol loading in other north/northeast Chinathat contributedtothe higher aerosol loading (Zhang et al., 2012),s. Some dustdusts cases has alsocan been observed foundin YRD region that transported from north/northwest China during 2012-2015 reflect the effect of mineral dust aerosols (Gong et al., 2003). ,I suspect thatThe fugitive dust from road traffic orand construction activity is another more persistent and significant source for China's cities as well as these eastern megacities., which reflectsthe effect ofmineral dust aerosols (Gong et al., 2003).However, this effect is not as obvious in the YRD region as other regions in north or northeast China whichcontributedtothe opticalpropertiesof aerosolsin this region(Zhang et al., 2012)..

This was mainly caused by dust episodes fromnorth/northwest China, whichcontributedtothe opticalpropertiesof aerosolsin this region(Zhang et al., 2012).

[Figure]

[Figure]

Fig. 23. Variation in the total, fine- and coarse-mode AOD$_{440\ nm}$ over (a) Hangzhou, (b)
Xiaoshan, (c) Fuyang, (d) LinAn, (e) Tonglu, (f) Jiande and (g) ChunAn. The boxes represent
the 25th to 75th percentile distribution, while the dots and solid lines within each box represent
the mean and median, respectively.

Figure3shows that the annual extinction Angström exponent(EAE) atHangzhou,

~~Hangzhou was higher in January (~1.40±0.23) and September (~1.43±0.24).This indicatedthe~~

~~and winter (Tan et al.,2009).The EAE was lower in March (~1.16±0.24) and April (~1.13±0.22),~~

The monthly and diurnal cycle of AOD at 500nm has also been discussed in Fig.4 and Fig.5. The annual values of AOD500nm over the seven urban, suburban and rural sites in this study varied from 0.53 (ChunAn) to 0.76 (Hangzhou).The results show that two peaks of AOD at 500nm occurs in June and September with values of  and  in the urban site of Hangzhou, respectively which has the similar pattern as the other sites. The increase of AOD at 500nm in June is not corresponding to the same increase pattern of EAE (about 1.5) which indicates the aerosols types may be relatively constant in this region. The Fig.42 depicts the diurnal patterns of AOD at 500nm in this megacity area of eastern China. We can see that there are two types of diurnal patterns in this region. The daily AOD has been found increased in early morning (08:00 hr to 09:00 hr)  and afternoon (12:00 hr to 14:00 hr) about the value of 0.60 to 0.70 in Hangzhou, Xiaoshan, Fuyang and Linan, while the decreasing of daily AOD has been observed from 0.70 to 0.50 during the daytime (from 07:00 hr to 16:00 hr) in Tonglu, Jiande and ChunAn. The high AOD during 07:00～09:00 in the urban area may be due to the anthropogenic activities and aerosol emissions from the morning rush hour. The decreased AOD with the value of 0.37±0.36 occurred in the suburban cities of Tonglu, Jiande and ChunAn may be due to the meteorological conditions more than anthropogenic effects. During the day, the aerosols in the near-surface may spread into vertical as a result of turbulence due to the more and more unstable atmosphere by the continuous strengthening of solar radiation.

[Figure]

[Figure]

Fig.. Variation in the AOD at 500nm & EAE

at 440–870 nm over (a) Hangzhou, (b) Xiaoshan, (c) Fuyang, (d) LinAn, (e) Tonglu, (f)

Jiande and (g) ChunAn. The boxes represent the 25th to 75th percentile distribution, while the dots and solid lines within each box represent the mean and median, respectively.

[Figure]

Fig.5. Variation of diurnal cycle in the AOD at 500 nm over (a) Hangzhou, (b) Xiaoshan, (c)

Fuyang, (d) LinAn, (e) Tonglu, (f) Jiande and (g) ChunAn. The boxes represent the 25th to 75th percentile distribution, while the dots and solid lines within each box represent the mean and median, respectively.

Validation of the MODIS C6 retrieval AOD values was carried out by comparisonwith ground-based observations (Figure 4).The product of Terra MODIS/Terra and

Aqua MODIS/Aqua with Deep Blue (at 10km) and Dark Target (at 3km and 10km) methods at

3km and 10km has been evaluated against by ground-based observations separately in Figure.

802. We use the better estimated data of Quality flag = 3 and Quality flag=2, 3 for DT and

TB methods, respectively. The systematic performance of the Terra MODIS/TerraMODIS C6

retrieval AOD values was generally more stable in the YRD region compared with the MODIS/Aqua product with the two Deep Blue and Dark Target methods,  which most of the plots scattered around the 1:1 regression line. fitting

The correlation coefficients (R)  between the MODIS/Aqua and MODIS/Terra between by the Dark Target methods at 3km and sun photometer AOD (550 nm) values  were about 0.84 to 0.92 and 0.85 to 0.94 in the YRD region, respectively. The linear regression fitting performed better at the suburban sites of LinAn and Jiande according to the product of MODIS/Terra by the Dark Target methods at 3km. The fitting curve was almost consistent with the 1:1 reference line, which suggests that the aerosol properties were well defined for the MODIS C6 products. A large part of the MODIS retrieval AOD value was outside the expected error envelope of ± (0.05 + 20%$\tau_{CARSNET}$), especially for AOD values<0.80 in Hangzhou and Xiaoshan. This indicates that the MODIS retrieval algorithm could still be improved, especially in urban areas. The MODIS retrieval AOD performed better at the other five sites (Fuyang, LinAn, Tonglu, Jiande and ChunAn) in the YRD; most of the retrieved AOD values for these sites fell within the expected error envelope. The MODIS/Aqua retrievals with Dark Target methods at 3km were  underestimated while the MODIS/Terra retrievals with Dark Target methods at 3km were overestimated except  Hangzhou,  Tonglu and Jiande.  The small deviation at the suburban sites suggested that the MODIS C6 retrieval using the DT method was suitable for capturing the optical properties of aerosols in suburban areas with dense vegetation coverage of the YRD. However, this method may have larger difference in the urban areas with less vegetation such as Hangzhou. The correlation coefficients (R) of the MODIS/Aqua and MODIS/Terra between sun photometer AOD (550 nm) values by the Deep Blue and Dark Target methods at 10km were about 0.81 to

0.9081, 0.8573 to 0.9081, 0.6948 to 0.9182 and 0.8572 to 0.9386 in the YRD region, respectively. The MODIS/Aqua and MODIS/Terra retrievals with Deep Blue and Dark Target methods at 10km were underestimated except Hangzhou and Xiaoshan. In particular, the biases of the correlation coefficients (R) occurred in LinAn and Jiande has decreased from 0.94 and 0.90 to 0.87 and 0.88. The validation results correlation indicates is not as better as the MODIS product at 3km which indicate a good MODIS/TerraMODIS matching with better fitting correlation at 3km rather than 10km products.

The AOD overestimation retrieved using Dark Target (DT) and Deep Blue (DB) methods are more influenced by the SSA and the phase function of aerosol in eastern China with AOD >0.4 (Tao et al. 2015). Therefore, the detailed ground-based observation in this work is more helpful to the calibration of MODIS retrievals in eastern China.

[Figure]

[Figure]

Fig.564. Comparison of C6 MODIS/Aqua Dark Target (DT)MODIS AOD at 550 nm with the CARSNET AOD by the Dark Target methods at 3km in (a-1) Hangzhou, (a-2) Xiaoshan, (a-3) Fuyang, (a-4) LinAn, (a-5) Tonglu, (a-6) Jiande, (a-7) ChunAn and MODIS/TerraTerra MODIS DT AOD at 550 nm with the CARSNET AOD by the Dark Target methods at 3km in (b-1) Hangzhou, (b-2) Xiaoshan, (b-3) Fuyang, (b-4) LinAn, (b-5) Tonglu, (b-6) Jiande, (b-7) ChunAn. The red solid line represents the linear regression. The two black dotted lines represent the expected errors in the MODIS retrievals.

6Terra at 3km

[Figure]

Fig. 7. Comparison of  MODIS/AquaS Deep Blue (DB) AOD at 550 nm with the CARSNET AOD  at 10km in (a-1) Hangzhou, (a-2b) Xiaoshan, (a-3e) Fuyang, (a-4d) LinAn, (a-5e) Tonglu, (a-6f) Jiande  (a-7g) 
[revised manuscript text omitted]

~~However, the monthly SSA values at therural site of ChunAnonly varied from 0.92 to 0.95. We concluded thatthe type of aerosol at urban/suburban sites wasmore complex than at rural sites.Fig.6shows a significant decrease in the fine-mode SSA in July/Augustand in the coarse-mode SSA in March/April.At Hangzhou, the lower fine-mode SSA values in July/August(~0.92±0.08/~0.90±0.08) were probably a result of aerosols from biomass burning and thelower coarse-mode SSA values in March/April (~0.79±0.08/~0.81±0.07) may reflect the existence of light-absorbing dust aerosols (Yang et al., 2009). The SSA depends on the wavelength and dust particles absorbstronglyat short wavelengths,resulting in a lower SSA at 440nm (Eck et al., 2010).~~

~~The range of variation inthe SSAof fine particles (SSA$_f$)was0.93~0.05, whereasthe SSA for coarse-mode particles (SSA$_c$)was 0.81~0.84at the seven sites (Fig. 6).Thefine- and coarse-mode particles displayedsignificant scattering and absorptionabilities in the urban,~~

suburban and rural areas of the YRD region.Fig.6shows a significant decrease in the
fine-mode SSA in July/Augustand in the coarse-mode SSA in March/April.At Hangzhou, the
lower fine-mode SSA values in July/August(-0.92±0.08/-0.90±0.08) were probably a result of
aerosols from biomass burning and thelower coarse-mode SSA values in March/April
(-0.79±0.08/-0.81±0.07) may reflect the existence of light-absorbing dust aerosols (Yang et al.,
2009). The SSA depends on the wavelength and dust particles absorbstronglyat short
wavelengths,resulting in a lower SSA at 440nm (Eck et al., 2010).The absorption/scattering
properties of fine- and coarse-mode particles determine the total SSA in the YRD. These
differences in the SSA were mostly dependent on the type of aerosol and theratio of absorbing
and non-absorbing components in the aerosols.

[Figure]

[Figure]

Fig.6210. Variation in the  SSA at 440nm, 670nm 870nm and

1020nmover (a) Hangzhou, (b) Xiaoshan, (c) Fuyang, (d) LinAn, (e) Tonglu, (f) Jiande and (g) ChunAn. The boxes represent the 25th to 75th percentile distribution, while the dots and solid lines within each box represent the mean and median, respectively.

The real and imaginary parts of the refractive index represent the scattering and absorption capacity of particles, respectively. The refractive index is determined by the hygroscopic conditions and the chemical composition of the aerosols (Dubovikand King, 2000).

There was no significant difference between the real parts of the refractive index among the seven urban, suburban and rural sites in this study (range 1.41–1.43). The real parts of the refractive index in this study were smaller than the real parts of ammonium sulfate and ammonium nitrate (1.55), which may be due to the hygroscopic conditions or the mixture of dust particles. The real part of the refractive index was highest in March (~1.46±0.06) and

November (~1.45±0.06) and lowest in July (~1.42±0.06) and August (~1.41±0.07) at the urban sites.

The imaginary part of the refractive index was higher at the urban site of Hangzhou (~0.0112 ±

0.0104) as a result of the high loading of absorption aerosols in this region and was consistent with the lower SSA. High imaginary parts of the refractive index occurred in August at all urban, suburban and rural sites in the YRD, which may be due to the higher emission of absorptive particles by the post-harvest burning of crop residues. with more spectral dependence. The burning of crop residues may cause a large deterioration in the regional air quality in the YRD

region. A higher level of spring dust aerosols with absorption could contribute to a higher value of the imaginary part of the refractive index.

**3.3 Radius and aerosol volume size distributions**

~~aerosols over all sites. The fine-mode radii were ~0.2–0.3 µm in the YRD with a volume of~~

0.10–0.12 μm$^3$ and the coarse-mode radii were ~2.0 μm with a volume close to 0.07 μm$^3$. The amount of fine-mode aerosols was higher in June and September than in other months at almost sites, except for Xiaoshan. This could be caused by aerosol humidification (Eck et al., 2012; Li et al., 2010, 2014; Huang et al., 2016).This phenomenon is also found over Bejing and Shenyang in north/northeast China, suggesting that hygroscopic growth occurs over many regions of China (Li et al., 2011; Che et al., 2015c).

The coarse-mode radius in spring at all sites was smaller than in other cities in north and northeast China affected by frequent dust transport events in spring (Kong et al., 2011; Zhao et al., 2015). The coarse-mode particles showed a larger effective radius at all seven urban, suburban and rural sites in the summer, which may due to the adhesion of new particles onto larger particles (such as fly ash).

[Figure]

Fig.137.Variation in the annual volume size distribution over (a) Hangzhou, (b) Xiaoshan, (c) Fuyang, (d) LinAn, (e) Tonglu, (f) Jiande and (g) ChunAn.

**3.4 Aerosol optical properties of Absorption absorption aerosol optical depth and absorption Angström exponent**

The annual AAODs at Hangzhou, Xiaoshan, Fuyang, LinAn, Tonglu, Jiande and ChunAn were about 0.06±0.05, 0.05±0.04, 0.04±0.04, 0.05±0.04, 0.05±0.04, 0.06±0.04 and 0.04±0.03, respectively (Fig..81411). Thehigher annualvalues of the AAOD in Hangzhou and Jiandeindicatethat there are more absorbing aerosol particlesat 
[revised manuscript text omitted]

---

## Author Comment (AC2) · 30 Aug 2017

Reviewer #02

In this paper, the authors analyzed aerosol optical and physical properties derived from CARSNET ground-based measurements for 7 locations in the RYD region. Ground based observations are definitely welcomed in the community. However, this paper needs major revisions. Firstly, also suggested by the first reviewer, this is a poorly written paper. Also, there are several major technical issues with the paper. The first reviewer has done a great job highlighting issues and I have added several more comments below.

Response: Thanks for the reviewer's important and constructive comments. The manuscript has been revised and re-organized carefully to make the paper more concise and focused. The major overhaul has been done in some sections which need to be rewritten and reorganized according to the helpful suggestion.

1. Aerosol radiative forcing values are computed using a radiative transfer model (Global Atmospheric ModEl, GAME). However, no details are provided on their radiative transfer modeling efforts. For example, how do they define surface (broadband) characteristics? What are the required aerosol properties such as vertical distributions? The CARSNET observations are in discrete channels, how the authors perform a narrowband to broad band conversion? A detailed uncertainty analysis is also needed but is lacking.

Response: Thanks for the important and constructive suggestions. The detail on the radiative transfer modeling has been added in sention 2 line 188-211 as "*The ARF (aerosol radiative forcing) data were calculated by the radiative transfer module used by the AERONET inversion (Dubovik et al., 2006) under the assumption of cloud-free consideration. In this code, the aerosol vertical properties have been considered into a homogeneous atmosphere layers because of the weak dependent of ground radiances on the whole atmospheric column with minor uncertainties (Dubovik et al., 2000). The fluxes from 0.20 to 4.0µm were calculated according to the radiative transfer model GAME (Global Atmospheric ModEl) (Dubuisson et al., 1996, 2006; Roger et al., 2006). While the broadband radiation was calculated based on the aerosol optical depth, single scattering albedo and asymmetry factor based on those properties at four distinct wavelengths (440, 670, 870, 1020) which were linearly interpolated and extrapolated from the retrieval of the sun/sky-radiometer measurements. The*

*uncertainties have been found to about 30% including the influence of spectral and solar zenith angle in the aerosol radiative effect (Myhre et al., 2003; Zhou et al., 2005). The size distribution, complex refractive index, and spherical particles fraction has been retrieved from the almucantar plane in the measurements. The SA (surface albedo) is obtained from the MODIS albedo product (MCD43C3) with the interpolation value of 440, 670, 870, and 1020 nm. The water vapor at 940 nm has been retrieved by the sun photometer. The ozone content was obtained from NASA Total Ozone Mapping Spectrometer measurements from 1978 to 2004. And other atmospheric gaseous data came from the US standard 1976 atmosphere model. In this study, the two parameters of ARF at the surface (ARF-BOA) and at the top of the atmosphere (ARF-TOA) have been calculated to describe the aerosol direct radiation effect to account for the changes of the solar radiation by calculating the difference energy between the aerosols presentation and absentation.*".

2. The authors compared CARSNET measurements with a merged MODIS Deep Blue and Dark Target product. Given that the MODIS Deep Blue and Dark Target methods are fundamentally different in their retrieval processes, I would recommend the authors evaluating each product separately.

Response: Thanks for the helpful suggestions. According to both reviewers' comments, the Aqua product has been added to validate the MODIS AOD at 550nm according to the suggestions in section 3.2. The MODIS C6 aerosol optical thickness products refined by Levy et al. (2013) were evaluated against our ground-based observations by the Deep Blue and Dark Target methods at 3km and 10km separately in section 3.2 as follows in line 400-476:

[revised manuscript text omitted]

3. Also, in lines 248-249, it states "The EAE was lower in March ($\sim$1.16$\pm$0.24) and April ($\sim$1.13$\pm$0.22), which reflects the effect of mineral dust aerosols (Gong et al., 2003)." This seems to contradict to a later conclusion (Table 2) that the dust aerosol presence is insignificant for the region. This actually brings up an issue, as the authors try to compare mean properties of the 7 sites and try to provide explanations for the differences. Some explanations are weak with little or no supporting evidence. In addition, the differences in some of the mean properties are actually way smaller than variations (numbers after ± sign) of the data, and thus some statistical methods are needed to back up the authors' comments with consideration of data spreads.

Response: Both reviewers mentioned this question. Though the dust aerosol presence is not insignificant for this region but the particle size was larger in spring with small EAE in March (~1.16±0.24) and April (~1.13±0.22). Some dusts cases can be observed in YRD region that transported from north/northwest China during 2012-2015 reflect the effect of mineral dust aerosols as follow cases:

As can be seen in the figure below, the PM10 mass concentration is higher in March to May than the value in February and June. This pattern clearly indicates the influence of dust in spring of Hangzhou. The Eastern China has been affected by a wide range of dust event in North China. These dust events across the northwest to the northeast area of China, then continue keep going south and east China. The higher PM10 mass concentration is obviously affected by the dust come from the northwest.

[Figure]

For example, there is a dust transportation event occurred in Hangzhou on 05-09 March, 2013. A series of transported aerosol masses over Hangzhou are monitored by MPL shown through time-height cross section of extinction coefficient at 527nm. Referring to CALIPSO L2 retrieval results of vertical feature mask and aerosol subtypes illustrated in Fig. 9(b), the northwestern upwind areas of Hangzhou exists an aerosol layer mixed of "dust" and "polluted dust" about 3km thick from the surface in 5 March 2015, since when a thin external layer is detected concurrently. Therefore, the higher AOD in spring described in this study is significantly affected by dust process.

[Figure]

[Figure]

[Figure]

4. Line 108-109, I am not sure what the authors mean by "Levy et al. (2013) refined the MODIS Collection 6 (C6) aerosol retrieval process to provide better AOD retrievals". What is "better AOD retrieval"? May be the authors referring to more accurate AOD retrievals?

Response: Thank for the reviewer's suggestion. The "Levy et al. (2013) refined the MODIS Collection 6 (C6) aerosol retrieval process to provide better AOD retrievals" has been modified as "Levy et al. (2013) refined the MODIS Collection 6 (C6) aerosol retrieval process to provide more accurate AOD retrievals". in the revised manuscript on line 124-125.

5. Line 135 "Jiande, Xiaoshan Tonglu and Xiaoshan" should be "Jiande, Xiaoshan, Tonglu and

Xiaoshan"?

Response: The "Jiande, Xiaoshan Tonglu and Xiaoshan" in line 135 has been changed as "Jiande, Xiaoshan and Tonglu" in revised manuscript on line150.

6. Lines 143-144, "Instantaneous direct data for the AOD were selected at least ten times each day at temporal resolution of about three minutes" Define "direct data". Also, what are the section criteria? Details need to be provided.

Response: Thank for the useful suggestion. The direct data means AOD calculated by the direct solar measurements at each wavelength. This selection is to increase the representability of the aerosol optical characteristics. This criterion is also used in previous studies such as Che et al. (2015). The sentence is line 143-144 has been modified as "*Instantaneous AOD measurements more than ten times at each day were selected for daily average calculation and statistical analysis to increase the representability of the aerosol optical characteristics (Che et al., 2015).*" in revised manuscript in line 158-162.

7. Line 156, define EAOD.

Response: According to the reviewer's suggestion, the EAOD has been defined in line 183-184 as "*The EAOD in this study has been defined as extinction aerosol optical depth calculated by the direct solar measurements at the wavelengths of 440, 500,670, 870, 1020nm*".

8. Lines 169-171, "The AOD data from Terra-MODIS were validated by matching the CARSNET AODs within 30 minutes of the MODIS overpass within the 3$x$3 pixels surrounding the CARSNET site." Are both CARSNET and MODIS data averaged for the process? Need some details.

Response: Yes. Both the CARSNET and MODIS data are averaged in the process. The details has been descript as "*The AOD averaged data from Terra-MODIS and Aqua-MODIS were validated by matching the averaged CARSNET AODs within 30 minutes of the MODIS overpass within the 5x5 pixels surrounding the CARSNET site (Tao et al., 2015).*" in line 220-222.

9. Lines 180-182, "The AOD at the urban site of Hangzhou was the highest of all the study sites as a result of high local anthropogenic activity in this urban area compared with the other suburban and rural sites." Results do not support this comment, as the three sites, including Handzou, Xiaoshan and Fuyang, have mean AOD of 0.76. Also, as I mentioned before, variations in data are larger than differences in mean values. Some statistical analyses are needed to back up their conclusions.

Response: The authors agree with the reviewer's opinion. To decrease the misunderstanding, This sentence has been rewritten as "*The AOD440nm at Hangzhou Xiaoshan and Fuyang was higher as a result of the more industrial activity and high resident density in the eastern part metropolis region resulting in larger aerosol emissions compared with the other suburban and rural sites.*" in line 260-270.

10. Lines 189-197, the authors compared AOD values from the 7 sites to other regions in China as reported from other papers. However, the authors should also take the temporal variation into consideration (e.g., mean values change from year to year, right?).

Response: The reviewer's suggestions are right. The mean values could change from year to year. In this paper, we did not take the temporal variation into consideration because of the different observation periods. So we just compared the multi-year averaged AOD values from the 7 sites in this study to other regions.

11. Line 241, what is extinction Angström exponent?

Response: Thank for the suggestion. The extinction Angström exponent has been defined in 309-310 as "*As Fig.3 shown, the monthly average value of the extinction Angström exponent (EAE, -dln[EAOD(λ)]/dln(λ)).*"

12. Lines 311-314, "The characteristics of the SSA at these seven sites gradually increased from the east coast (0.91±0.06 at Hangzhou) inland toward the west (0.94±0.03 at ChunAn). These results indicate the emissions caused by human activity affect the absorption of aerosols in urban areas." Again, the authors need to consider other possibilities and worry about data varibility. How about meteorological conditions? What about hygroscopic growth? Again, the authors need to back up their comments with evidence.

Response: Thanks for the important suggestions. The sites in this study are in the city scale about 10 km apart from each other over adjacent urban, suburban and rural areas in the YRD region. Therefore the same weather system can be regarded as weak effect of meteorological elements. The discussion has been added in line 512-518 as "*The seven observation sites are usually controlled by the same weather system that indicates a weak effect of meteorological elements in each site to the change of aerosol optical characteristics. These results indicate the emissions caused by human activity affect the absorption of aerosols in urban areas. The SSA was higher at LinAn and ChunAn than at the other sites, which may reflect the presence of a larger number of scattering aerosols (e.g. particles from urban/industrial activities) over the clear rural sites than over urban or suburban sites.*" in section 3.3. Furthermore, we added the discussion of hygroscopic growth in line 529-531as "*The increased level of scattering aerosols with higher SSA in June may be influenced by the hygroscopic growth in favor of the interaction between aerosols from different emissions sources (Xia et al., 2007).*"

13. Figure 10, what do the colors represent?

Response: Figure 10 has been changed as Figure 14 in the revised manuscript. The blue dot represent the sphericity fraction of particles and the red dot represent the AAE ($\alpha_{abs}$) values (AAE = −dln[AAOD(λ)]/dln(λ)).

[revised manuscript text omitted]

$^{a}$ Number of available observation days.

$^{b}$ Number of instantaneous observations.

$^{c}$ Optical parameters at a wavelength of 500nm.

$^{d}$ Optical parameters at a wavelength of 440 nm.

$^{d}$ $^{e}$Angström exponents between 440 and 870 nm.

$^{f}$ Optical parameters at a wavelength of 670 nm.

$^{g}$ Optical parameters at a wavelength of 870 nm.

$^{h}$ Optical parameters at a wavelength of 1020 nm.

Ding et al. (2013a,b) showed that plumes from agricultural burning in June may significantly and seriously affect the radiation balance and air quality of the YRD region. In this study, the monthly averaged AODs at most sites showed two peaks in June and September (Fig.23) with values of ~1.26±0.50 and ~1.03±0.57, respectively. This may be attributed to the accumulation of fine-mode particles via hygroscopic growth in the summer season and the burning of crop residue biomass under a continental high-pressure system with good atmospheric stability and frequent temperature inversions. These conditions lead to the poor diffusion of pollutants (Xia et al., 2007). As Fig.3 shown, the monthly average value of the extinction Angström exponent (EAE, -$d$ln[EAOD($\lambda$)]/$d$ln($\lambda$))  in Hangzhou was higher in

January (~1.40±0.23) and September (~1.43±0.24). This conclusion is also indicated the dominance of small particles from anthropogenic emissions and agricultural activity in autumn and winter (Tan et al., 2009).

The annual fine-mode AOD values at Hangzhou, Xiaoshan, Fuyang, LinAn, Tonglu,

Jiande and ChunAn were about 0.68±0.42, 0.69±0.41, 0.69±0.44, 0.66±0.43, 0.64±0.41,

0.66±0.40 and 0.61±0.38, respectively (Fig.3). The seasonal variation in the AOD was similar to the total AOD at these urban, suburban and rural sites. The fine-mode fraction of

AOD consistently exceeded 0.90 which indicates a major contribution of fine mode fraction to the total AOD in the YRD.

Moreover, the fFigure 3 shows that the annualextinction Angström exponent(EAE)EAE at Hangzhou, Xiaoshan, Fuyang, LinAn, Tonglu, Jiande and ChunAn was about 1.29±0.26, 1.37±0.24, 1.32±0.24, 1.29±0.27, 1.30±0.26, 1.32±0.28 and 1.22±0.25, respectively. Values of EAE >1.20 were found in all months throughout the year, indicating that small particle size distributions were favored in the YRD region. The annual coarse-mode AOD values at Hangzhou, Xiaoshan, Fuyang, LinAn, Tonglu, Jiande and ChunAn were between about 0.06 and 0.08.The with the ratio coarse mode fraction of AODAOD$_c$/AOD$_t$ was about 0.10, which indicates that about 10% of the 10% contribution of coarse mode fraction to the AOD in the YRD. regionis from coarse particles .The variation in the coarse-mode AOD (Fig. 2) also showed a significant increase in Marchatallseven sites of about 0.14±0.08, 0.08±0.04, 0.09±0.09, 0.13±0.11, 0.13±0.11, 0.14±0.08 and0.11±0.07at Hangzhou, Xiaoshan, Fuyang, LinAn, Tonglu, Jiande and ChunAn, respectively.The monthly average value of the EAEin Hangzhou was higher in January (~1.40±0.23) and September (~1.43±0.24). This indicated the dominance of small particles from anthropogenic emissions and agricultural activity in autumn and winter (Tan et al., 2009).The lower EAE was lower in March (~1.16±0.24) and April (~1.13±0.22)ThoughThe less coarse mode fraction indicated that there is no obvious effect of the coarse particles in the YRD region than that contributed to the higher aerosol loading in other north/northeast Chinathat contributedtothe higher aerosol loading (Zhang et al., 2012),s. Some dustdusts cases has alsocan been observed foundin YRD region that transported from north/northwest China during 2012-2015 reflect the effect of mineral dust aerosols (Gong et al., 2003). ,I suspect thatThe fugitive dust from road traffic orand construction activity is another more persistent and significant source for China's cities as well as these eastern megacities., which reflectsthe effect ofmineral dust aerosols (Gong et al., 2003).However, this effect is not as obvious in the YRD region as other regions in north or northeast China whichcontributedtothe opticalpropertiesof aerosolsin this region(Zhang et al., 2012)..

This was mainly caused by dust episodes fromnorth/northwest China, whichcontributedtothe opticalpropertiesof aerosolsin this region(Zhang et al., 2012).

[Figure]

[Figure]

Fig. 2̶3. Variation in the total, fine- and coarse-mode $AOD_{440\ nm}$ over (a) Hangzhou, (b) Xiaoshan, (c) Fuyang, (d) LinAn, (e) Tonglu, (f) Jiande and (g) ChunAn. The boxes represent the 25th to 75th percentile distribution, while the dots and solid lines within each box represent the mean and median, respectively.

Figure3shows that the annual extinction Angström exponent(EAE) atHangzhou,

Xiaoshan, Fuyang, LinAn, Tonglu, Jiande and ChunAnwasabout 1.29±0.26, 1.37±0.24, 1.32±0.24, 1.29±0.27, 1.30±0.26, 1.32±0.28 and 1.22±0.25, respectively.Values of EAE >1.20were found in all monthsthroughoutthe year,indicatingthat smallparticlesize distributions were favored in the YRD region.The monthly average valueof the EAEin Hangzhou was higher in January (~1.40±0.23) and September (~1.43±0.24).This indicatedthe dominance of small particles fromanthropogenic emissions and agricultural activity in autumn and winter (Tan et al.,2009).The EAE was lower in March (~1.16±0.24) and April (~1.13±0.22), which reflectstheeffect ofmineraldustaerosols (Gong et al., 2003). However, this effectis not as obvious in the YRD region as other regions in north or northeast China.

[revised manuscript text omitted]

**3.5 Aerosol optical properties of Aerosol aerosol radiative forcing at the Earth's surface**

**and top of the atmosphere**

[revised manuscript text omitted]

from anthropogenicactivity. Hygroscopic growth and the burning of biomassfrom crop residuesin the summer season could cause this obvious increase in the AOD. The ratios of AOD/AOD fine mode fraction of AOD was(>0.90) and coarse mode fraction of AOD (~0.10) consistently >0.90, indicating that fine-mode particles made a major contribution to the total AOD in the YRD.Theas well as the relationship between the EAE and the spectral difference in the EAE suggested thatthe dominance of fine mode fraction to the AOD and the subordinate position of coarse mode fraction in the YRD. dustis not importantin eastern China. The validation results indicates a good Terra-MODIS matching with better fitting correlation at 3km rather than 10km products with the The MODIS C6 AOD retrievals performed better in suburban than in urban and rural areas, but were systematically over estimated in rural and urban areas and their immediate surroundings.A large part of the MODIS retrieval AOD was outside the expected error,especially atAOD values <0.80 in urban areasand their immediate surroundings.

The range of variationof the total, fine- and coarse-mode SSA at 440nmvalues was about 0.91–0.94, 0.93–0.95and 0.81–0.84, respectively, 
[revised manuscript text omitted]

---

## Author Comment (AC3) · 30 Aug 2017

Reviewer #03

This paper presented the aerosol optical properties as observed over seven CARSNET sites over eastern China. Aerosol loading, together with aerosol SSA, refractive index, and particle size were analyzed in a monthly scale. Authors also evaluated MODIS AOD retrievals, analyzed aerosol type, and calculated radiative forcing using those ground-based measurements. Their study covered many aspects of aerosol properties over the studied domain, providing a comprehensive analysis. However, the paper lacks focus, by simply redundantly piling data analysis variable by variable. As indicated by the other two reviewers, the paper is poorly structured. Substantial revisions are needed to improve the organization of data analysis on various parameters and make the paper more concise and focused. Some aerosol variables (i.e, AE and aerosol size, and SSA and refractive index) are highly related and should to be presented interactively. The paper also needs to be re-organzied in three different section, i.e., (1) analysis of aerosol properties; (2) validation of MODIS (this part indeed does not sever the objectives of this paper, should consider to remove); (3) radiative forcing estimate. In addition, a discussion section may be added to discuss the results and how they can be interpreted in perspective of previous studies, as well as the strength and limitation of the present study (see my below comments).

Response: Thanks for the reviewer's important and constructive comments and suggestions. According to the reviewers suggestion, the manuscript has been revised and re-organzied carefully to make the paper more concise and focused. The aerosol variables including aerosol volume size distribution, AE, SSA and refractive index has been presented interactively. The revised manuscript also re-organized the section of aerosol properties analyzing, MODIS validation and radiative forcing estimating. In addition, the discussion section has been added and re-discussed.

Specific comments:

1. Page 3, L18-19: Many networks are listed here. But is not clear "which" network "includes several automated sites in China". Please revise this in a more accurate way.

Response: Thank for the suggestions. This sentence "……*which includes several automated sites in China.*" has been revised as "*The above networks exclude EARLINET include several*

*automated sites in China.*" in Introduction line 88.

2. Page 6, L14: The SSA was retrieved using only -> The retrieved SSA was used only when

Also, please provide reference for selecting AOD440 of 0.4 as the threshold.

Response: Thank for the important suggestions. The reference of $AOD_{440}$nm about 0.40 as the threshold has been added in line 175-177 "*The SSA was retrieved using only $AOD_{440nm}>0.40$ measurements to avoid the large uncertainties inherent in a low AOD (Dubovik et al. 2002, 2006).*".

3. Page 6, L15-19: These two sentence are very confusing and need rewords like: Real and imaginary parts of refractive index at 4 wavelengths (440, 675, 870, and 1020 nm) were retrieved from sky radiance and were confined in the range of : : :, respectively. Also retrieved were aerosol volumes of xx size bins within the 0.05 - 15 um radius range.

Response: According to the reviewer's helpful suggestions, the two sentence in line 15-19 "*The real and imaginary parts of the complex refractive index were retrieved for the wavelengths corresponding to sky radiance measurements in the ranges 1.33–1.6 and 0.0005–0.5, respectively (Dubovik and King, 2000; Yu et al., 2009; Che et al., 2009a)*" has been modified as "*Real and imaginary parts of refractive index at 4 wavelengths (440, 675, 870, and 1020 nm) were retrieved from sky radiance and were confined in the range of 1.33– 1.60 and 0.0005–0.50, respectively (Dubovik and King, 2000; Che et al., 2015b). Also retrieved were aerosol volumes of 22 size bins within the 0.05 - 15 µm radius range.*" In revised manuscript on line 177-182.

4. Page 7, L10: Justification is needed for using 3x3 pixel averaging.

Response: Thank for the suggestions. We rechecked the pixel value and the "*The aerosol data from Terra-MODIS were validated by matching the CARSNET AODs within 30 minutes of the MODIS overpass within the 3 × 3 pixels surrounding CARSNET site*" in line 10 has been modified as "*Terra-MODIS and Aqua-MODIS were validated by matching the averaged CARSNET AODs within 30 minutes of the MODIS overpass within the 5×5 pixels surrounding the CARSNET site (Tao et al., 2015).*" in revised version manuscript in line 220-222.

5. In the first paragraph of the result section (and in many places in the following sections), authors included a lot of comparisons of AOD between YRD area to other regions of China. Such comparisons may be interested but would distract readers. The result section should focus on presenting the findings, and such extensive discussion should be placed in a discussion section.

Response: Thanks for the reviewer's important suggestion. The comparison of AOD between YRD and other regions of China in the first paragraph has been replaced in the section 4 of Discussion and Summary.

6. Figure 2: The font size of the figure labels and legends should be increased.

Response: Thank for the suggestions. The font size of the figure labels and legends has been increased in the revised manuscript as Figure 3.

7. Page 14, L2-3: Could authors explain in more detail on the reasoning of "method for estimating the surface reflectance was suitable for this region"? What about surface reflectance estimation in Hangzhou site?

Response: Thanks for the reviewer's suggestion. In the revised paper, the incorrect description has ben corrected. Actually, the two sites of LinAn and Jiande are with dense vegetation coverage, the surface reflectance was suitable in the MODIS C6 retrieval DT method and the optical properties of aerosols could be captured better than the urban site of Hangzhou with less vegetation inducing the large difference between satellite and ground-based AOD values. The correction in the revised manuscript is as following "*The small deviation at the suburban sites suggested that the MODIS C6 retrieval using the DT method was suitable for capturing the optical properties of aerosols in suburban areas with dense vegetation coverage of the YRD. However, this method may have larger difference in the urban areas with less vegetation such as Hangzhou.*" in line 429-432.

8. Section 3.2: Please note that AERONET inversion algorithm assume refractive index does not vary with aerosol particle size [Dubovik et al. 2000]. In other words, refractive indices are same for retrieved fine mode and coarse mode. As a results, mode-specific SSA were not recommend to use due to large uncertainty. Furthermore, the coast-mode aerosol loading is too small to offer sufficient information on absorption of the coarse-mode particles. Therefore, only total SSA should be used for the analysis to avoid misleading (even AERONET total SSA has error of 0.03). In addition, SSA on a longer wavelength could be included to examine the absorbing aerosol type, as different absorbing particles (dust and smoke) appear different spectral contrast of SSA.

Response: The reviewer's comments are very constructive. According to the reviewer's suggestion, the SSA of fine and coarse mode has been removed and the total SSA at 440, 670, 870, 1020nm have been re-figured in Figure 10. Also the section of SSA has been rewritten in the revised manuscript as follows in section 3.3 "*The distribution of the SSA at the wavelengths of 440nm, 670nm, 870nm and 1020nm over the seven sites in the YRD are shown in Fig.10. The SSA varied from 0.91 to 0.94, which is similar to the range seen in other regions of China, such as Wuhan (0.92), Beijing (0.89) and Xinglong (0.92) (Wang et al., 2015; Xin et al., 2014; Zhu et al., 2014). This indicated that scattering aerosol particles in eastern China resulting from high levels of industrial and anthropogenic activity were dominant. The characteristics of the SSA at these seven sites gradually increased from the east coast (0.91±0.06 at Hangzhou) inland toward the west (0.94±0.03 at ChunAn). The seven observation sites may always controlled by the same weather system that indicates a weak effect of meteorological elements in each site to the change of aerosol optical characteristics. These results indicate the emissions caused by human activity affect the absorption of aerosols in urban areas. The SSA was higher at LinAn and ChunAn than at the other sites, which may reflect the presence of a larger number of scattering aerosols (e.g. particles from urban/industrial activities) over the clean rural sites than over urban or suburban sites.*

*The SSA over urban and suburban sites showed the largest monthly variation. The monthly average values of SSAT were high in February (~0.94±0.05) and June (~0.92±0.06), but low in March (~0.90±0.06) and August (~0.89±0.09) in Hangzhou. However, the monthly SSA values at the rural site of ChunAn only varied from 0.92 to 0.95. We concluded that the type of aerosol at urban/suburban sites was more complex than at rural sites. The increased level of scattering aerosols with higher SSA in June may be influenced by the hygroscopic growth in*

*favor of the interaction between aerosols from different emissions sources (Xia et al., 2007). The existence of light-absorbing dust aerosols may contribute to the weaker lower SSA in spring while the aerosols from biomass burning were probably due to the strong decreased in SSA values in August (Yang et al., 2009).*

*The wavelength dependence of SSA present specific absorption/scattering properties of different type aerosol (Sokolik and Toon, 1999; Eck et al., 2010). The SSA of dust in spring shown a dependence on the spectrum from 440nm to 1020nm in general (Cheng et al., 2006; Dubovik et al., 2002). Especially in March, the SSA at 440nm in Hangzhou, LinAn, Jiande and ChunAn was obviously lower at short wavelength than that in the longer wavelength. This result has shown a strong absorption of dust in the short wavelength in the YRD region over eastern China. It's worth noting that there is an obvious and strong decreasing of SSA in the longer wavelength of aerosol from biomass burning or industrial emissions in August. The wavelength dependence of SSA in YRD could be used to simply describe the aerosol types including dust or the biomass burning smoke."*

.

[Figure]

Fig.10. Variation in the SSA at 440nm, 670nm 870nm and 1020nmover (a) Hangzhou, (b) Xiaoshan, (c) Fuyang, (d) LinAn, (e) Tonglu, (f) Jiande and (g) ChunAn. The boxes represent the 25th to 75th percentile distribution, while the dots and solid lines within each box represent the mean and median, respectively.

9. Figure 5: As in comments, please consider removing SSA of each mode and retaining the total SSA. Font size of labels should be increased.

Response: Thanks for the reviewer's suggestion. The SSA of fine and coarse mode has been removed and the total SSA at 440, 670, 870, 1020nm has been re-discussed. The Font size of labels has also been increased. Please see the above responses.

10. Section 3.3: Again, AERONET refractive index retrievals are not size dependent.

Response: The reviewer's suggestion is very important. The authors agree to the reviewer's suggestion. According to the reviewer's suggestion, the incorrect description of refractive index has been corrected in the revised paper as following *"The real and imaginary parts of the refractive index represent the scattering and absorption capacity of particles, respectively. The refractive index is determined by the hygroscopic conditions and the chemical composition of the aerosols (Dubovikand King, 2000). There was no significant difference between the real parts of the refractive index among the seven urban, suburban and rural sites in this study (range 1.41–1.43). The real parts of the refractive index in this study were smaller than the real parts of ammonium sulfate and ammonium nitrate (1.55), which may be due to the hygroscopic conditions or the mixture of dust particles. The real part of the refractive index was highest in March (~1.46±0.06) and November (~1.45±0.06) and lowest in July (~1.42±0.06) and August (~1.41±0.07) at the urban sites.*

*The imaginary part of the refractive index was higher at the urban site of Hangzhou (~0.0112 ± 0.0104) as a result of the high loading of absorption aerosols in this region and was consistent with the lower SSA. High imaginary parts of the refractive index occurred in August at all urban, suburban and rural sites in the YRD, which may be due to the higher emission of absorptive particles by the post-harvest burning of crop residues with more spectral dependence. The burning of crop residues may cause a large deterioration in the regional air quality in the YRD region. A higher level of spring dust aerosols with absorption could contribute to a higher value of the imaginary part of the refractive index."* in line 449-466.

11. Section 3.6: Is Figure 13 based on monthly averaged variables? If true, monthly data may cause problem in classifying aerosol type. Those would simply represent the mean values of those parameters rather the mean states of aerosol types. The information of actual aerosol types may fade out during averaging process.

Response: The reviewer's comments are very important. The Figure 13 has been Figure 9 in the revised paper. This Figure is not based on monthly averaged variables but based on instantaneous data to classifying aerosol type clearly.

12. Table 2: Pie chart may be a better option to present the aerosol type category.

Response: Thanks for the reviewer's suggestion. The Pie chart has been added as Figure 13 instead of Table 2 to present the aerosol type category as following: "*The non-absorption particles are account for ~50 to 80% in the YRD region. There is higher contribution of non-absorption particles about 78.17% in Fuyang and less non-absorption particles about 50.01% in Jiande. The result is consistent with the level of total SSA at 440nm of Fuyang (0.94) with more scattering particles than Jiande (0.92).*" in line 735-739.

[Figure]

*Fig.13.Types of aerosol over (a) Hangzhou, (b) Xiaoshan, (c) Fuyang, (d), LinAn, (e) Tonglu, (f) Jiande and (g) ChunAn.*

13. Page 3, L25: aerosols present -> aerosols

Response: According to the reviewer's suggestion. The "aerosols present" has been changed as "aerosols" in line 96 the revised manuscript.

14. Page 6, L 21: Do you intend to say "from 0.2 to 4.0 um"?

Response: Yes. It should be "The broadband fluxes from 0.2 to 4.0 μm were calculated ……." in line 193 the revised manuscript.

15. Page 6, L26-27: I believe the reference for AERONET inversion algorithm (sphericity fraction) should be Dubovik et al. [2006], please verify.

Response: Thank for the suggestion. The reference "*The aerosol sphericity fractions were retrieved from measurements in the almucantar plane according to the inversion algorithms in Holben et al. (2006) and Eck et al. (2008).*" has been deleted the revised manuscript.

16. Page 7, L1: "were used to provide an evaluation of MODIS AOD retrieval with" -> were evaluated against

Response: Thank for the suggestion. The sentence "*……were used to provide an evaluation of MODIS AOD retrieval with……*" in line 1 has been modified as "*The MODIS C6 aerosol optical thickness products refined by Levy et al. (2013) were evaluated against our ground-based observations by the Deep Blue (at 10km) and Dark Target (at 3km and 10km) methods separately.*" in section 2 line 212-215.

17. Page 7, L16: the variation range of AOD at 440 nm is -> the annual mean of AOD at 440 nm ranges

Response: Thank for the suggestion. The sentence "……the variation range of AOD at 440 nm is……" has been modified as "The annual mean of AOD at 440nm over the seven urban, suburban and rural sites in this study ranges from 0.68 to 0.76 (Table 1)." in section 3.2 of the revised manuscript in line 245-246.

18. Page 7, L24: The word "trend" often refers to change with time. "pattern" could be better option.

Response: According to the suggestion, the word "trend" has been changed as "pattern" all through the text.

19. Page 8, L3: a little bit higher -> slightly higher

Response: According to the suggestion, the "a little bit higher" has been changed as "slightly higher" in Discussion part of line 806 in the revised manuscript.

20. Page 9, L16: Please use "fine-mode fraction of AOD"; consistent higher -> consistently higher

Response: Thank for the suggestions of reviewers. The "$AOD_f/AOD_t$" has been changed as "fine-mode fraction of AOD" all through the revised manuscript. The "consistent higher" has been changed to "consistently exceeded …." in line 318.

21. Page 9, L18: variation for -> variation of

Response: Thank for the suggestion. The "variation for" has been corrected in the revised paper.

**Aerosol optical properties and  radiative forcing based on synchronous measurements of China Aerosol Remote Sensing Network (CARSNET) over eastern China**

Huizheng Che[1*], Bing Qi[2], Hujia Zhao[1], Xiangao Xia[3,4], Philippe Goloub[5], Oleg Dubovik[5],

Victor Estelles[6] , Emilio Cuevas-Agulló[7], Luc Blarel[3], Yunfei Wu[8], Jun Zhu[9], Rongguang Du[2],

Yaqiang WANG[1], Hong Wang[1], Ke Gui[1], Jie Yu[1], Yu Zheng[9], Tianze Sun[1], Quanliang Chen[10],

Guangyu Shi[11], Xiaoye Zhang[1*]

State Key Laboratory of Severe Weather (LASW) and Institute of Atmospheric Composition, Chinese Academy of Meteorological Sciences, CMA, Beijing, 100081, China

Hangzhou Meteorological Bureau, Hangzhou, 310051, China

Laboratory for Middle Atmosphere and Global Environment Observation (LAGEO), Institute of Atmospheric Physics, Chinese Academy of Sciences, Beijing, 100029, China

School of Geoscience University of Chinese Academy of Science, Beijing, 100049, China

Laboratoire d'Optique Amosphérique, Université des Sciences et Technologies de Lille, 59655, Villeneuve d'Ascq, France

Dept. Fisica de la Terra i Termodinamica, Universitat de Valencia, C/ Dr. Moliner 50, 46100 Burjassot, Spain

Centro de Investigación Atmosférica de Izaña, AEMET, 38001 Santa Cruz de Tenerife , Spain

Key Laboratory of Regional Climate-Environment for Temperate East Asia, Institute of Atmospheric Physics, Chinese Academy of Sciences, Beijing 100029, China

Collaborative Innovation Center on Forecast and Evaluation of Meteorological Disasters, Nanjing University of Information Science & Technology, Nanjing 210044, China

Plateau Atmospheric and Environment Key Laboratory of Sichuan Province, College of Atmospheric Sciences, Chengdu University of Information Technology, Chengdu, 610225, China

State Key Laboratory of Numerical Modeling for Atmospheric Sciences and Geophysical Fluid Dynamics (LASG), Institute of Atmospheric Physics, Chinese Academy of Sciences, Beijing, 100029, China

Corresponding author: chehz@camscma.cn & xiaoye@camscma.cn

**Abstract**

Variations in the optical properties of aerosols and their radiative forcing were investigated based on long-term synchronous observations made at three-minute intervals from 2011 to 2015 over seven adjacent CARSNET (China Aerosol Remote Sensing NETwork) urban (Hangzhou), suburban (Xiaoshan, Fuyang, LinAn, Tonglu, Jiande) and rural (ChunAn) stations in the Yangtze River Delta region, eastern China. The fine-mode radii in the Yangtze River Delta region were ~0.2–0.3 μm with a volume fraction of 0.10–0.12 μm$^3$ and the coarse-mode radii were ~2.0 μm with a volume fraction close to 0.07 μm$^3$. The radii of fine volume fraction in the Yangtze River Delta region were ~0.2–0.3 μm with a volume of 0.10–0.12 μm$^3$ and the radii of coarse volume fraction were ~2.0 μm with a volume close to 0.07 μm$^3$. The fine-mode aerosols were obviously larger in June and September than in other months at almost the sites. The aerosol optical depth (AOD at 440nm) varied from 0.68 to 0.76, with two peaks in June and September, and decreased from the eastern coast to western inland areas. The ratio of the AODof fine-mode particlesfine mode fraction to the total AOD was >0.90 and the extinction Angström exponent was >1.20 throughout the year at all seven sites. The AOD at 500nm has also been studied because of the wavelength dependent of optical properties to show the monthly and diurnal cycle. againstThe Moderate Resolution Imaging Spectroradiometer (MODIS) C6 retrieval AOD was validated by comparisonwith ground-based observations.The correlation coefficients ($R^2$R) between the MODIS C6 AODdata and the values measuredon the ground were ~0.73–0.89. 
[revised manuscript text omitted]

| [e]AOD[d]$AOD_{440nm}$ | 0.76±0.42 | 0.76±0.43 | 0.76±0.45 | 0.73±0.44 | 0.71±0.41 | 0.73±0.40 | |
| [e]AOD_{fine}[d]$AOD_{fine(440nm)}$ | 0.68±0.42 | 0.69±0.41 | 0.69±0.44 | 0.66±0.43 | 0.64±0.41 | 0.66±0.40 | |
| [e]AOD_{coarse}[d]$AOD_{coarse(440nm)}$ | 0.08±0.06 | 0.07±0.06 | 0.07±0.06 | 0.07±0.07 | 0.07±0.06 | 0.07±0.07 | |
| [d]EAE[e]EAE | 1.29±0.26 | 1.37±0.24 | 1.32±0.24 | 1.29±0.27 | 1.30±0.26 | 1.32±0.28 | 1 |
| [e]SSA[d]$SSA_{440nm}$ | 0.91±0.06 | 0.93±0.04 | 0.94±0.04 | 0.93±0.05 | 0.92±0.04 | 0.92±0.05 | |
| [e]SSA_{fine}[df]$SSA_{670nmfine}$ | 0.923±0.065 | 0.915±0.064 | 0.935±0.064 | 0.924±0.054 | 0.934±0.054 | 0.924±0.075 | |
| [e]SSA_{coarse}[dg]$SSA_{870nmcoarse}$ | 0.9082±0.079 | 0.9083±0.078 | 0.9184±0.08 | 0.981±0.068 | 0.981±0.068 | 0.9082±0.089 | |
| [h]$SSA_{1020nm}$ | 0.89±0.08 | 0.89±0.08 | 0.89±0.09 | 0.90±0.07 | 0.90±0.07 | 0.90±0.09 | |
| [e]Real[d]Real | 1.43±0.07 | 1.41±0.06 | 1.41±0.06 | 1.42±0.06 | 1.43±0.06 | 1.41±0.05 | 1 |
| [e]Imaginary[d]Imaginary | 0.011±0.010 | 0.008±0.006 | 0.007±0.006 | 0.009±0.007 | 0.009±0.007 | 0.010±0.009 | |
| [e]AAOD[d]AAOD | 0.06±0.05 | 0.05±0.04 | 0.04±0.04 | 0.05±0.04 | 0.05±0.04 | 0.06±0.04 | |
| [d]AAE[e]AAE | 1.13±0.46 | 0.88±0.42 | 0.85±0.43 | 0.98±0.35 | 1.11±0.49 | 1.16±0.44 | |
| [e]Rmeas_{t}[d]$Rmeas_{t}$(µm) | 0.70±0.34 | 0.65±0.31 | 0.66±0.33 | 0.66±0.33 | 0.65±0.33 | 0.62±0.24 | |
| [e]Rmea_{fine}[d]$Rmea_{fine}$(µm) | 0.18±0.05 | 0.18±0.04 | 0.19±0.05 | 0.19±0.05 | 0.19±0.05 | 0.19±0.05 | |
| [e]Rmea_{coarse}[d]$Rmea_{coarse}$(µm) | 2.67±0.47 | 2.73±0.42 | 2.75±0.45 | 2.71±0.52 | 2.66±0.48 | 2.63±0.47 | 2 |
| [e]Reff[d]Reff(µm) | 0.30±0.10 | 0.29±0.09 | 0.30±0.09 | 0.29±0.10 | 0.29±0.10 | 0.29±0.09 | |

| | | | | | | | |
|---|---|---|---|---|---|---|---|
| [e] [d]Reff$_{fine}$(µm) | 0.16±0.04 | 0.16±0.03 | 0.17±0.04 | 0.16±0.04 | 0.16±0.04 | 0.17±0.04 | |
| [e] [d]Reff$_{coarse}$(µm) | 2.21±0.40 | 2.26±0.35 | 2.30±0.39 | 2.24±0.44 | 2.19±0.41 | 2.16±0.39 | |
| [e] [d]Volume(µm$^3$) | 0.19±0.09 | 0.19±0.09 | 0.19±0.09 | 0.18±0.09 | 0.17±0.09 | 0.18±0.09 | |
| [e] [d]Volume$_{fine}$(µm$^3$) | 0.10±0.06 | 0.11±0.06 | 0.11±0.07 | 0.10±0.06 | 0.10±0.06 | 0.10±0.06 | |
| [e] [d]Volume$_{coarse}$(µm$^3$) | 0.09±0.06 | 0.08±0.05 | 0.08±0.06 | 0.08±0.05 | 0.08±0.06 | 0.08±0.07 | |
| [e] [d]ARF-BOT(W/m$^2$) | −93±44 | −84±41 | −80±40 | −81±39 | −79±39 | −82±40 | |

[revised manuscript text omitted]
 included to examine the aerosol types absorbing aerosol type, as different absorbing particles (including dust or the andbiomass burning smoke) appear different spectral contrast of SSA.

.

However, the monthly SSA values at therural site of ChunAnonly varied from 0.92 to 0.95. We concluded thatthe type of aerosol at urban/suburban sites wasmore complex than at rural sites.Fig.6shows a significant decrease in the fine-mode SSA in July/Augustand in the coarse-mode SSA in March/April.At Hangzhou, the lower fine-mode SSA values in July/August(~0.92±0.08/~0.90±0.08) were probably a result of aerosols from biomass burning and thelower coarse-mode SSA values in March/April (~0.79±0.08/~0.81±0.07) may reflect the existence of light-absorbing dust aerosols (Yang et al., 2009). The SSA depends on the wavelength and dust particles absorbstronglyat short wavelengths,resulting in a lower SSA at 440nm (Eck et al., 2010).

The range of variation inthe SSAof fine particles (SSA$_f$)was0.03–0.05, whereasthe SSA for coarse-mode particles (SSA$_c$)was 0.81–0.84at the seven sites (Fig. 6).Thefine- and coarse-mode particles displayedsignificant scattering and absorptionabilities in the urban, suburban and rural areas of the YRD region.Fig.6shows a significant decrease in the
fine-mode SSA in July/Augustand in the coarse-mode SSA in March/April.At Hangzhou, the
lower fine-mode SSA values in July/August(-0.92±0.08/-0.90±0.08) were probably a result of
aerosols from biomass burning and thelower coarse-mode SSA values in March/April
(-0.79±0.08/-0.81±0.07) may reflect the existence of light-absorbing dust aerosols (Yang et al.,
2009). The SSA depends on the wavelength and dust particles absorbstronglyat short
wavelengths,resulting in a lower SSA at 440nm (Eck et al., 2010).The absorption/scattering
properties of fine- and coarse-mode particles determine the total SSA in the YRD. These
differences in the SSA were mostly dependent on the type of aerosol and theratio of absorbing
and non-absorbing components in the aerosols.

[Figure]

[Figure]

Fig.6̶1̲2̲10. Variation in the t̶o̶t̶a̶l̶,̶ ̶f̶i̶n̶e̶-̶ ̶a̶n̶d̶ ̶c̶o̶a̶r̶s̶e̶-̶m̶o̶d̶e̶ SSA at 440nm, 670nm 870nm and

1020nm_{440 nm}over (a) Hangzhou, (b) Xiaoshan, (c) Fuyang, (d) LinAn, (e) Tonglu, (f) Jiande and (g) ChunAn. The boxes represent the 25th to 75th percentile distribution, while the dots and solid lines within each box represent the mean and median, respectively.

The real and imaginary parts of the refractive index represent the scattering and absorption capacity of particles, respectively. The refractive index is determined by the hygroscopic conditions and the chemical composition of the aerosols (Dubovikand King, 2000). There was no significant difference between the real parts of the refractive index among the seven urban, suburban and rural sites in this study (range 1.41–1.43). The real parts of the refractive index in this study were smaller than the real parts of ammonium sulfate and ammonium nitrate (1.55), which may be due to the hygroscopic conditions or the mixture of dust particles. The real part of the refractive index was highest in March (~1.46±0.06) and November (~1.45±0.06) and lowest in July (~1.42±0.06) and August (~1.41±0.07) at the urban sites.A higher level ofdust aerosols with weak scattering in spring and autumn could contribute to a higher value of the real part of the refractive index; this was reduced or eliminated by rainfall during the summer months.

The imaginary part of the refractive index was higher at the urban site of Hangzhou (~0.0112 ± 0.0104) as a result of the high loading of absorption aerosols in this region and was consistent with the lower SSA. High imaginary parts of the refractive index occurred in August at all urban, suburban and rural sites in the YRD, which may be due to the higher emission of absorptive particles by the post-harvest burning of crop residues. with more spectral dependence. The burning of crop residues may cause a large deterioration in the regional air quality in the YRD region. A higher level of spring dust aerosols with absorption could contribute to a higher value of the imaginary part of the refractive index.

**3.3 Radius and aerosol volume size distributions**

Fig.7shows 13 shows the monthly aerosol size distribution (dV/dlnr) in the YRD for all sites. The volumes of fine-mode aerosols were obviously higher than those of coarse-mode aerosols over all sites. The fine-mode radii were ~0.2–0.3 μm in the YRD with a volume of

0.10–0.12 µm$^3$ and the coarse-mode radii were ~2.0 µm with a volume close to 0.07 µm$^3$. The amount of fine-mode aerosols was higher in June and September than in other months at almost sites, except for Xiaoshan. This could be caused by aerosol humidification (Eck et al., 2012; Li et al., 2010, 2014; Huang et al., 2016).This phenomenon is also found over Bejing and Shenyang in north/northeast China, suggesting that hygroscopic growth occurs over many regions of China (Li et al., 2011; Che et al., 2015c).

The coarse-mode radius in spring at all sites was smaller than in other cities in north and northeast China affected by frequent dust transport events in spring (Kong et al., 2011; Zhao et al., 2015). The coarse-mode particles showed a larger effective radius at all seven urban, suburban and rural sites in the summer, which may due to the adhesion of new particles onto larger particles (such as fly ash).

[Figure]

Fig.137.Variation in the annual volume size distribution over (a) Hangzhou, (b) Xiaoshan, (c) Fuyang, (d) LinAn, (e) Tonglu, (f) Jiande and (g) ChunAn.

**3.4 Aerosol optical properties of Absorption absorption aerosol optical depth and absorption Angström exponent**

The annual AAODs at Hangzhou, Xiaoshan, Fuyang, LinAn, Tonglu, Jiande and ChunAn were about 0.06±0.05, 0.05±0.04, 0.04±0.04, 0.05±0.04, 0.05±0.04, 0.06±0.04 and 0.04±0.03, respectively (Fig..81411). Thehigher annualvalues of the AAOD in Hangzhou and Jiandeindicatethat there are more absorbing aerosol particlesat 
[revised manuscript text omitted]
 ratios of AODf/AODt fine mode fraction of AOD was(>0.90) and coarse mode fraction of AOD (~0.10) consistently >0.90, indicating that fine-mode particles made a major contribution to the total AOD in the YRD.Theas well as the relationship between the EAE and the spectral difference in the EAE suggested thatthe dominance of fine mode fraction to the AOD and the subordinate position of coarse mode fraction in the YRD. dustis not importantin eastern China. The validation results indicates a good Terra-MODIS matching with better fitting correlation at 3km rather than 10km products with the The MODIS C6 AOD retrievals performed better in suburban than in urban and rural areas, but were systematically over estimated in rural and urban areas and their immediate surroundings.A large part of the MODIS retrieval AOD was outside the expected error,especially atAOD values <0.80 in urban areasand their immediate surroundings.

The range of variationof the total, fine- and coarse-mode SSA at 440nmvalues was about 0.91–0.94, 0.93–0.95and 0.81–0.84, respectively, 
[revised manuscript text omitted]

---

## Author Response (AR2)

1. After reviewing the revised manuscript by Che et al., I regrettably have to recommend not accepting it for publication at ACP. Many of the issues found during the initial review have not been properly addressed.

   Response: We thank the reviewer of his or her constructive comments and suggestions. A major overhaul of the paper has been done to address the review comments—all concerns raised in the initial reviews have been considered this time. To this end, the abstract, results analysis, discussion, and conclusion sections all have been rewritten. In the Introduction, the importance of the study has been explained, and the objectives clearly stated. In the section 2, as the reviewer suggested, additional information has been added about the uncertainties and methods of the aerosol retrieval and ARF calculations, and links are now provided to previous studies and literature. Section 3 now presents a more detailed and coherent analyses of the results, and the discussion has been revised and expanded to make the paper more compelling. In the section 4, the conclusions have been rewritten, and they now concisely summarize the most important findings of our study. The detailed issues have been highlighted (in RED) in the revised manuscript.

2. The revised manuscript has plenty of redundancy and poor logic, and fails to present a solid and interesting analysis.

   Response: The authors have thoroughly re-organized this manuscript. The sections concerning diurnal AOD cycles, aerosol complex refractive index, comparisons to MODIS results, type classification by EAE and the spectral difference have been deleted.

   The flow of the revised manuscript is more straightforward than in previous versions. In brief, we first use data from a dense network composed of seven urban and rural sites to characterize the climatology of aerosol properties (including microphysical and optical characteristics) over the Yangtze Rive Delta (YRD) region. The aerosol properties were then used to calculate direct aerosol radiative forcing (DARF) under clear conditions over the region. As the DARF is affected by aerosol absorptivity as well as the underlying surface conditions, we classified the aerosols by type (strength of absorption for particles in both fine and coarse modes) based on the optical properties. In summary, the results of this study will be of interest to those who study aerosol-climate interactions or air-pollution. The above description

summarizes what we have done to improve the flow of the paper.

A more in-depth analysis has been done in the revised manuscript to make the study substantial and interesting. Previous ground-based studies in the YRD have usually been conducted over short or discontinuous periods of time (information on seasonal but not monthly averages has been provided), and therefore information on the temporal and spatial variations in aerosol characteristics over YRD is limited. In this study, we uncovered some important information on aerosol-climate connections through careful analysis of the data.

In brief, (1) we first considered aerosol microphysical properties (particle radius and volume size distributions) and found that the aerosol populations over the YRD region differ from those in northern and northeastern China. High volumes and effective radii of fine-mode aerosols occurred in June and September, and they were consistently lower in July and August. In contrast, high volumes and effective radii of fine-mode aerosol typically occur in July over northern China and northeastern China. This difference can be attributed to the unique regional climatology of aerosols over YRD as we discuss in the revised paper.

(2) Next, the aerosol optical properties over the YRD were analyzed in detail. Aerosol extinction (AOD) also showed two peaks, one in June and the other in September, and a minimum occurred in July because of the fine-mode particle effects. This is unlike northern and northeastern China where only one peak in AOD typically occurs from June to August. Also, the AOD seasonal pattern over YRD in this study is different from the "high in summer but low in winter" pattern in urban areas of north China found in previous studies. The AOD in winter over YRD, especially in January and February, is as high as in March to May. The mean extinction Angström exponent was found higher than 1.20 throughout the year, indicating that small particles were predominant in the region, which is different from North China where coarse dust particles are found in abundance in spring due to recurrent dust storms. An obvious wavelength dependence in SSA, that is strong absorption at infrared wavelengths, was found in July and August, and this was due to biomass burning or industrial emissions. The inter-site differences in the absorbing aerosol optical properties have been discussed in the revised manuscript, and the analysis indicated a degree of spatial heterogeneity in the distributions of absorbing aerosols even though the AODs were relatively similar. The difference in aerosols was attributed to the complex emission sources that impact the urban

and rural sites in the YRD.

(3) The direct DARF-BOA values under clear conditions were also calculated, and the results showed strongly negative DARF-BOA in June, followed by March and September due to high aerosol extinction. The monthly DARF-TOA means under clear conditions varied smoothly (averaging -40 W/m$^2$) during two periods: one in January to May and the other October to December. The mean DARF-TOA values under clear conditions were about −20 W/m$^2$ at seven sites in July/August with obviously lower AOD and SSA over YRD region. In contrast, the DARF-TOA values under clear conditions at Shenyang (Northeastern China urban area), Beijing (Northern China urban area), and Xianghe (Northern China rural area) showed the negative peak during June to August due to the large aerosol extinctions in summer season. The calculations also showed that the DARF-TOA was positive from April to October when the SSA $_{440\ nm}$ was < 0.80, and there was strong wavelength dependence, probably as a result of the burning of crop residues.

(4) As for the aerosol type classifications, we used the SSA, FMF, and EAE values to classify the absorbing weakly-absorbing particles in the fine and coarse modes. The results showed that the aerosol absorption is weak to moderate in the YRD region, and the fine-mode particles has an especially large contribution to the high percentage of absorbing particles at Hangzhou.

3.  Details are still lacking in how RT simulations are conducted (aerosol layer altitude, cloud conditions, RF definition, etc), and the RT code has too large error/uncertainty (L180).

Response: Following to the reviewer's suggestion, more information has been added to the paper to explain how the RT simulations are done; the paper now includes a definition of RF, and it explain the assumptions regarding the altitude of aerosol layer , gas absorption, etc. We apologize for inaccurate earlier descriptions that were confusing to the reviewer. In this study, the authors investigated the direct aerosol radiative forcing (DARF) under free cloud conditions because the calculations were made based on the aerosol microphysical and optical parameters retrieved from ground-based measurements under free cloud conditions Details concerning the DARF calculations and error/uncertainty descriptions are highlighted (in RED) in Section 2 of the revised manuscript. Furthermore, the description of RT code error/uncertainty has been corrected as following in the present revision as follows:

"*The flux calculations were performed for a multi-layered atmosphere with the US standard 1976 atmosphere model for gaseous distributions and single fixed aerosol vertical distribution (exponential with an aerosol height of 1 km) (Gacia et al., 2008). As these authors have pointed out, solar fluxes calculated using the module described above show excellent agreement with ground-based measurements of solar radiation (slope of 0.98 ± 0.00 and bias of -5.32 ± 1.00 W/m2) with a correlation of 99%. There is a small overestimation of +9 ± 12 Wm-2 of the observed solar radiation at the surface in global terms, and this corresponds to a relative error of +2.1 ± 3.0%. The differences range from +14 ± 10 Wm-2 to +6 ± 13 Wm-2 for urban-industrial and biomass burning aerosols, respectively. The errors are expected to be of the same magnitude at the TOA, since the same methodology and inputs are used at both levels (gaseous and aerosol distribution, radiative model, etc).*"

4. I encourage the authors to formulate the most interesting and important findings from their data, rather than indulge themselves in the plain report of everything as what it is. It seems to me, the most important finding is that from all aspects, the seven sites show very little differences and indicate the presence of very similar aerosol and climate conditions.

Response: The authors agree with the reviewer's suggestions. In previous version, we simply presented the data without an in-depth analysis of the results. In the revised version, the most interesting and important findings from our data analysis have been highlighted and discussed in detail (Please see Response 2 above and also the revised manuscript).

Following up on the reviewer's important comments, we initially consider the data for the seven sites to investigate climatology of aerosol microphysical and optical characteristics over the YRD region. The seven sampling sites in this study are located in an area of 170 x 40km, and they include one densely populated urban site (Hangzhou), five urban center sites in smaller cities (LinAn, Fuyang, Jiande, Xiaoshan and Tonglu), and one rural site (ChunAn). Although there were generally small differences among sites, the results showing very similar aerosol and climate conditions provide a broadly representative picture of the aerosol populations over the whole YRD region. This is particularly true given that most of the previous studies were conducted at a single site and not for an extended time. Furthermore, even though the aerosols were relatively similar among seven sites, there are also some subtle

differences in the aerosol optical properties, DARF, and aerosol type classification between the urban and rural areas. These differences reflect the impacts of local anthropogenic effects over specific parts of the YRD.

5. Comparison to MODIS should not be a goal of this study, as pointed out in the first round review, and should be reformulated/reduced.

Response: According to the first round review suggestion, the comparison to MODIS retrievals has been deleted but likely will be discussed in future work.

6. English needs improvement in quite a few places (misspells, typos, grammar, etc), a sign of haste and carelessness.

Response: The English has been polished by a native speaker carefully.

I found authors have addressed most of the reviewers's comments to improve the quality of the article. I only have a few minor and technical comments for the authors to consider.

Response: Thanks for the reviewer's help to improve the quality. We added "The authors would like to thank the three anonymous reviewers and the editor for their constructive suggestions and comments in the revised manuscript.

1. In terms of the paper structure, I still think it is not appropriate to present the validation of MODIS AOD in section 3.2. This section presents the aerosol optical properties from Sun Photometers, and satellite of validation belongs to one of the applications of the CARSNET products. So, please consider to use a standalone section for MODIS validation.

Response: Following both reviewers' suggestions, the comparison to MODIS retrievals has been deleted in the revised paper.

2. Line 39-40: "volume fraction" means the fraction of volume, which is not appropriate to be used as volume. I recommend to replace it with "modal volume" or "fractional volume". And the units of volume here should um^3/um^2, volume of aerosol particle per um^2 area. This correction should be applied across the entire article.

Response: The abstract has been rewritten according to the reviewers' suggestions. In the revised abstract and elsewhere, the term of "volume fraction" has been represented as "volume" or "fractional volume". The units of volume in the revised manuscript have been corrected as "$\mu m^3/\mu m^2$" across the entire paper.

3. Line 42: 440nm -> 440 nm. This also applies elsewhere in the article.

Response: The term of "440nm" has been correct into "440 nm" throughout the revised manuscript.

4. Line 48: 10km -> 10 km; 3km -> 3 km. Again, elsewhere in the article.

Response: Because the part about the comparison to MODIS retrievals has been deleted, the terms of "10km" and "3km" have been removed in the revised paper.

5. Line 54-57: First, change "radiative forcing" into "direct radiative forcing", because indirect radiative forcing was not considered. Second, it lacks innovation to simply say "aerosols

causes negative forcing ...", because negative radiative effect of aerosol over low-reflectivity surface has been well known. So, present the numbers. Third, it reads "… the lower surface albedo in a unique geographical climate condition of better vegetation in the YRD region than in north/northeast China." So, does it means that aerosols in north/northeast China exert a positive radiative effects? Please make it clear.

Response: According to the reviewer's suggestion (1) the term of "radiative forcing" has changed into "direct radiative forcing"; (2) the detailed aerosol direct radiative forcing numbers have been added in the revised paper; (3) to avoid confusion to readers, this sentence in question was removed from the revised paper.

6. Figure 1: I don't see any description of the two red river lines on the map. Description maybe needed in the figure caption, otherwise, consider to remove them.

Response: According to the reviewer's suggestion, the two red river lines representing the Yellow River and Yangtze River on the map in Figure 1 have been removed in the revised paper.

7. Lines 222, 406, 464, and 557: Redundant section titles. Consider to remove "Aerosol optical properties of"

Response: The section titles have been shortened in the revised paper.

---

## Author Response (AR3)

Reviewer 1

1. The revised manuscript has been greatly improved in presentation quality and science merit. I thereby would recommend accepting it for publication after a few comments are addressed. I notice that there is a change in the author list and would recommend other proper changes are made if any.

   Response: Thanks for the reviewer's comments. Dr. T. F. Eck gave many constructive suggestions and comments, so we added him as one of co-authors. Before re-submitting the revised manuscript, we have clarified this with the Co-Editor.

2. L55, clarify the time-scale for DARF, is it monthly or annual mean? Remove " which indicates cooling at the surface and top of the atmosphere". This is obvious from the DRAF values, and more importantly, could be misleading because the DRAF is calculated assuming cloud-free conditions.

   Response: According to the reviewer's suggestion, the incorrect descriptions have been modified.

   (1) "The annual mean DARF was ……"

   (2) The sentence of "… which indicates cooling at the surface and top of the atmosphere" has been removed in the revised manuscript.

3. L188, DARF is defined based on the net flux change at TOA or surface, however, equations 4 and 5 do not speak so. In both equations only upward or downward fluxes are used, which are inconsistent with the definitions. You may argue solar down flux could be dropped in Eq. 4; but Eq.5 is simply wrong. Just stay consistent in your definitions and equations. Also, by convention, dF (delta_F) is more often used to represent net flux, not DARF.

   Response: Thanks for the reviewer's suggestions. The incorrect description has been modified to keep the definitions and equations consistently in the revised manuscript. We also changed "$\Delta F$" to "$DARF$" to avoid misunderstanding.

   "The DARF is defined as the difference in the shortwave radiative fluxes between the two energy levels including and excluding aerosol effects at the Earth's surface (bottom of the atmosphere, BOA) and the top of the atmosphere (TOA) in equations (4) and (5) as follows:"

$$\text{DARF}_{TOA} = F_{TOA}^{\uparrow 0} - F_{TOA}^{\uparrow}$$
$$\text{DARF}_{BOA} = F_{BOA}^{\downarrow} - F_{BOA}^{\downarrow 0}$$

redReviewer 3

I found the revised manuscript is significantly improved. I have a few minor comments in below that author should consider to address.

Response: Thanks for the reviewer's comment.

Specific comments:

1.  It maybe necessary to include one additional important assumption in AERONET inversion products. That is, the complex refractive index is assumed independent of particle size. In other words, the inversion algorithm [Dubovik et al., 2000, 2006] gives the same refractive for both fine and coarse modes. This assumption valid for fine or coarse-dominated cases, however, could cause error in SSA and even particle size for mixed aerosol scenarios. This issue has been discussed by Xu and Wang [2015] and Xu et al [2015]. The same studies also made efforts to retrieve mode-dependent aerosol complex refractive indices and SSA.

    Response: Thanks for the reviewer's suggestion. The reviewer's comments have been considered in the revised manuscript. "The complex refractive index is assumed independent of particle size. This assumption is valid for fine or coarse-dominated cases, however, could cause some errors in SSA and particle size retrievals for mixed aerosol scenarios (Xu and Wang, 2015; Xu et al 2015)."

2.  The quality of most figures needs improvements. In particular, the font size of plots and legends should be enlarged.

    Response: The quality of most figures has been improved. And the font size of plots and legends has been enlarged.

3.  Line 161-162: The description of fine and coarse mode separation may be incorrect here. Please verify. According to the official description of AERONET inversion Products, "the inversion code finds the minimum withinthe size interval from 0.439 to 0.992 μm. This minimum is used as a separation point between fine and coarse mode particles. Using that separation, the code simulates optical thickness, phase function and single scattering albedo of fine and coarse mode separately. Furthermore, the retrieval provides estimates of Effective Radius reff, Volume Median Radius reff, Standard Deviation σ and Volume concentrationsCv (μm3/μm2) for both fine and coarse modes of the retrieved size distribution." Source: https://aeronet.gsfc.nasa.gov/new_web/Documents/Inversion_products_V2.pdf

    Response: The incorrect description has been modified in the revised manuscript. "Following the procedures of Dubovik et al. (2002, 2006), all particles smaller than 0.992 μm were considered fine mode particles while those larger than 0.992 μm were considered coarse mode."

4.  Line 242-244: ".., but the range of effective radii was greater, possibly due in part to the lower retrieval accuracy for coarse particles compared with the fine-mode." This confuses me, as coarse-mode effective radius seems not quite dispersed among different sites (according to Table 1). Please double check.

Response: The description has been corrected as "The coarse-mode aerosol volumes also showed small differences among site, but the range of effective radii varied 2.16–2.30 μm.".

5. Line 322: The abbreviation "FMF" needs to be defined. And, is the FMF in terms of particle volume?
Response: The abbreviation "FMF" has been defined as fine mode particle AOD fraction ($AOD_{fine(440nm)}/AOD_{440nm}$) in the revised paper.

6. References:
Xu, X., and J. Wang (2015), Retrieval of aerosol microphysical properties from AERONET photopolarimetric measurements: 1. Information content analysis, Journal of Geophysical Research: Atmospheres, 120, 7059–7078, doi:10.1002/2015JD023108.
Xu, X., et al. (2015), Retrieval of aerosol microphysical properties from AERONET photopolarimetric measurements: 2. A new research algorithm and case demonstration, Journal of Geophysical Research: Atmospheres, 120, 7079–7098, doi:10.1002/2015JD023113.
Response: The above two references have been considered in the revised manuscript.